# Functionally integrating nanoparticles alleviate deep vein thrombosis in pregnancy and rescue intrauterine growth restriction

Juan Cheng[1,2,3,4,7], Siqi Zhang[1,3,7], Chenwen Li[4], Ke Li[1,3], Xiaoyan Jia[1,3], Quanfang Wei[5], Hongbo Qi ®[1,2] ✉ & Jianxiang Zhang ®[4,6] ✉

There is still unmet demand for effective, safe, and patient-friendly anti-thrombotics to treat deep vein thrombosis (DVT) during pregnancy. Here we first engineer a bioactive amphiphile (TLH) by simultaneously conjugating Tempol and linoleic acid onto low molecular weight heparin (LMWH), which can assemble into multifunctional nanoparticles (TLH NP). In pregnant rats with DVT, TLH NP can target and dissolve thrombi, recanalize vessel occlusion, and eradicate the recurrence of thromboembolism, thereby reversing DVT-mediated intrauterine growth restriction and delayed development of fetuses. Mechanistically, therapeutic effects of TLH NP are realized by inhibiting platelet aggregation, facilitating thrombolysis, reducing local inflammation, attenuating oxidative stress, promoting endothelial repair, and increasing bioavailability. By decorating with a fibrin-binding peptide, targeting efficiency and therapeutic benefits of TLH NP are considerably improved. Importantly, LMWH nanotherapies show no toxicities to the mother and fetus at the dose 10-time higher than the examined therapeutic dosage.

Deep vein thrombosis (DVT) during pregnancy can lead to pulmonary embolism, recurrent abortion, fetal growth restriction, preeclampsia, and placental abruption, which are associated with high morbidity, mortality, and costs, therefore representing a life-threatening disease to the mother and fetus[1–3]. In particular, pulmonary embolism caused by DVT is the main cause of maternal death[3,4]. Extensive clinical studies indicated that DVT during pregnancy generally occurs in the iliofemoral vein of the left lower limb[5]. During pregnancy, especially in the middle and late trimesters, blood hypercoagulation leads to dysfunction of the coagulation system, which in turn causes the imbalance between the anticoagulant and fibrinolytic system, thereby resulting in DVT in pregnancy[6,7]. The pathogenesis of DVT is closely associated with elevated levels of procoagulant factors/proteins[8], endothelial cell injury and apoptosis[9], platelet activation and aggregation[10], over-produced reactive oxygen species (ROS)[11], and infiltration of inflammatory cells[12]. DVT-related symptoms, such as blood flow block[13], pulmonary embolism[4], placental microthrombosis[4], and placental ischemia/hypoxia[14], not only threaten the maternal life, but also potentially affect the growth and development of the fetus[15]. Therefore, clinical interventions are required to reduce the occurrence and development of DVT in pregnancy.

Considering the physiological state of pregnancy, mechanical intervention such as implantation of vena cava filters is not suitable, since it causes severe side effects on both mothers and fetuses[16]. Whereas anticoagulants, antiplatelet drugs, and thrombolytic agents can prevent and dissolve DVT, most of these drugs show the risk of

[1]Chongqing Key Laboratory of Maternal and Fetal Medicine, The First Affiliated Hospital of Chongqing Medical University, 400016 Chongqing, China. [2]Women and Children's Hospital of Chongqing Medical University, 401147 Chongqing, China. [3]Department of Obstetrics, The First Affiliated Hospital of Chongqing Medical University, 400016 Chongqing, China. [4]Department of Pharmaceutics, College of Pharmacy, Third Military Medical University (Army Medical University), 400038 Chongqing, China. [5]Biomedical Analysis Center, College of Basic Medical Sciences, Third Military Medical University (Army Medical University), 400038 Chongqing, China. [6]State Key Lab of Trauma, Burn and Combined Injury, Third Military Medical University (Army Medical University), 400038 Chongqing, China. [7]These authors contributed equally: Juan Cheng, Siqi Zhang. ✉e-mail: qihongbo728@163.com; jxzhang@tmmu.edu.cn

major bleeding and thrombocytopenia in mothers[17,18], as well as increase the incidence of fetal congenital heart disease (for example, indomethacin, a clinically used anti-platelet drug, can cause premature closure of the ductus arteriosus) and skeletal malformations (such as warfarin) by crossing through the placental barrier[19,20], thereby limiting their clinical applications. Currently, only aspirin and low molecular weight heparin (LMWH) are recommended to prevent and treat DVT during pregnancy. LMWH is a clinically well-recognized antithrombotic agent that does not cross the placental barrier[21]. Although LMWH generally shows a relatively low incidence of hemorrhage, the related risk still exists. Once this happens, most hemostatic drugs cannot be given to pregnant women, which is likely to cause adverse pregnancy outcomes. Besides, LMWH therapy can lead to other side effects, such as local bruising, skin reactions, and the development of osteoporotic fractures[22]. These limitations are closely related to the nonspecific distribution of LMWH[23]. In addition, LMWH shows poor patient adherence and compliance, mainly caused by twice-a-day injection due to its short half-life.

Recently, either synthetic or biomimetic nanoparticles (NPs) with various biophysicochemical properties have been examined as thrombus-targeting delivery vehicles of different anti-thrombotic agents for the treatment of DVT[24–28]. Whereas anticoagulant and/or

thrombolytic strategies mainly used in these cases can achieve effective antithrombotic effects, the related therapies cannot fully ameliorate thrombotic disorders resulting from local endothelial injury due to inflammation and oxidative stress. As well documented, inflammation and oxidative stress can induce thrombosis, while coagulation also augments inflammation/oxidative stress, resulting in a vicious cycle[29,30]. Moreover, thus far no any anti-thrombotic nanotherapies have been examined in pregnant animals to demonstrate their therapeutic advantages and safety profiles. Of note, some NPs caused pregnancy complications and neurotoxicity in the offspring in mice due to their transplacental transfer[31–33]. Consequently, innovative treatment strategies as well as effective and safe therapies remain to be developed for DVT in pregnancy.

In view of the pivotal role of inflammation and oxidative stress in the formation of thrombosis[29,30], we hypothesize that NPs integrated with antithrombotic, anti-inflammatory, antioxidant, and endothelial protective activities can serve as potent therapies for DVT, while nanotherapies with limited transplacental transport capacity may be used safely during pregnancy. As a proof of concept, a multifunctional amphiphile derived from LMWH was first engineered by simultaneously conjugating with Tempol and linoleic acid (LA). Tempol functions as an antioxidative component[34], while LA is an anti-

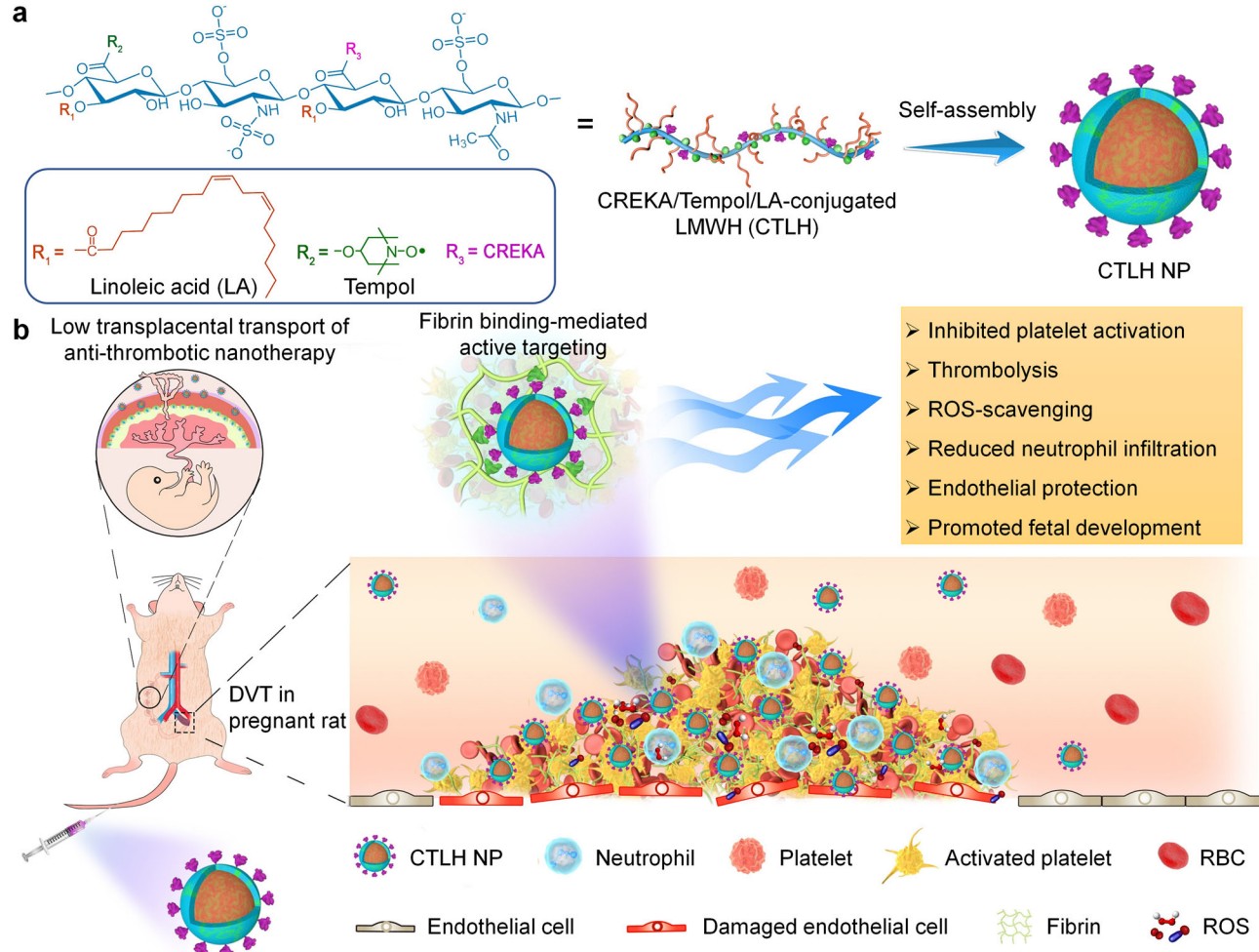

**Fig. 1 | Schematic illustration of engineering of a multifunctional anti-thrombotic nanotherapy for targeted treatment of deep vein thrombosis (DVT) during pregnancy. a** Development of a nanotherapy with multiple bioactivities by self-assembly of a multifunctional low molecular weight heparin (LMWH) derivative (i.e., CTLH). Covalent conjugation of LMWH with bioactive moieties of

linoleic acid (LA) and Tempol affords an amphiphile with multiple pharmacological activities, which is further functionalized with a thrombus-targeting peptide CREKA to obtain CTLH that can assemble into a multifunctional nanotherapy CTLH NP. **b** A sketch showing targeted treatment of DVT in pregnant rats by CTLH NP.

inflammatory agent that has been practically used as a nutrient supplement for pregnant women[35]. Thus engineered amphiphilic conjugate can assemble into NPs with multiple bioactivities, which showed desirable antithrombotic effects in pregnant rats with induced DVT and effectively alleviated fetal growth restriction caused by DVT. By decorating with a fibrin-targeting peptide, therapeutic effects of this multifunctional nanotherapy can be further improved. Of note, both passive and active targeting nanotherapies demonstrated excellent safety performance in pregnant rats.

## Results

### Design of advanced nanotherapies for the treatment of DVT during pregnancy

For targeted treatment of DVT during pregnancy, we rationally designed and developed a thrombolytic, anti-inflammatory, anti-oxidative, and fibrin-targeting nanotherapy by a hierarchical functionalization strategy (Fig. 1). LMWH, a clinically used anticoagulant, also exhibits endothelial-protective effects. LMWH was employed as a bioactive scaffold, onto which Tempol and LA were covalently conjugated. In addition to its anti-inflammatory activity, LA has been practically used as a nutrient supplement during pregnancy, demonstrating beneficial effects on neural development of the fetus[35]. Tempol is a well-recognized ROS-scavenging agent, and its derivative has been clinically investigated for the treatment of oxidative stress-related diseases[36]. To enhance thrombus-targeting capability, the LMWH-derived amphiphile was further functionalized with CREKA, a peptide ligand that can target thrombi by specifically binding to fibrin and clotted plasma proteins[37–39]. The finally obtained amphiphile can assemble into a spherical nanotherapy with multiple bioactivities including anti-coagulant, anti-inflammatory, anti-oxidative, and endothelial-protective effects.

### Preparation and characterization of nanotherapies based on amphiphilic LMWH derivatives

First, LA-conjugated LMWH (LH) was synthesized by conjugating LA onto LMWH via an ester bond (Supplementary Fig. 1a). Similarly, Tempol/LA simultaneously conjugated LMWH (TLH) was prepared by sequentially linking Tempol and LA onto LMWH via ester bonds (Supplementary Fig. 1b). $^1$H NMR and Fourier transform infrared (FTIR) spectroscopy demonstrated the presence of LA and Tempol/LA moieties in finally obtained LH and TLH, respectively (Supplementary Fig. 2a, b). For TLH, electron paramagnetic resonance (EPR) spectroscopy showed a sharp triplet signal, resulting from the interaction between $^{14}$N and the unpaired electron of oxygen (Supplementary Fig. 2c)[40]. Of note, the EPR spectrum of TLH was similar to that of Tempol, indicating that covalent conjugation did not change the free radical structure of Tempol. Calculation according to $^1$H NMR spectra revealed that the content of LA in LH was 35.5%, while the content of Tempol and LA in TLH was 19.5% and 24.4%, respectively.

NPs derived from LH or TLH were prepared by a dialysis-based self-assembly method (Fig. 2a, b), which were defined as LH NP and TLH NP, respectively. Observation by transmission electron microscopy (TEM) indicated that both NPs displayed a spherical shape, with core-shell structure (Fig. 2c, d). The spherical shape of LH NP and TLH NP was also confirmed by scanning electron microscopy (SEM; Fig. 2e, f). Measurement by dynamic light scattering revealed relatively narrow size distribution profiles for LH NP and TLH NP (Fig. 2g, h), with an average diameter of $169 \pm 1$ and $175 \pm 3$ nm, respectively. Both LH NP and TLH NP showed negative ζ-potential (Supplementary Fig. 3a). Upon incubation in PBS or serum, we observed slight changes in both mean diameter and ζ-potential for LH NP and TLH NP (Supplementary Fig. 3b and Fig. 2i), indicating both nanotherapies displayed good colloidal stability. Fluorescence spectrometry using pyrene as a probe was employed to explore the mechanism underlying self-assembly and the NPs formation by LH

and TLH[41]. Besides enhanced fluorescence intensities (Fig. 2j), a red shift appeared in the excitation spectra of pyrene with increased LH concentrations, indicating the onset of aggregation and formation of hydrophobic microdomains. The critical aggregation concentration (CAC) was obtained according to the plot of LH concentration-dependent changes in the $I_{338}/I_{333}$ ratios (Fig. 2k), defined as the intersection of two line segments separately drawn through the lowest and rapidly increasing points. Consequently, self-assembly of LH NP is mainly mediated by hydrophobic interactions between LA moieties, with CAC of ~63 μg/mL. Similarly, self-assembly of TLH in aqueous solution was examined (Fig. 2l, m), showing CAC of ~25 μg/mL.

### In vitro biological effects of LH NP and TLH NP

**Antioxidant activity.** We first examined ROS-scavenging capability of LH NP and TLH NP. It was found that TLH NP effectively eliminated superoxide anion, free radical, hydrogen peroxide ($H_2O_2$), and hypochlorite ($ClO^-$), in a dose-dependent manner (Fig. 2n–q). These results indicated that TLH NP was capable of eliminating a broad spectrum of ROS. Nevertheless, LH NP showed very limited ROS-scavenging capacity, as compared to TLH NP (Supplementary Fig. 3c–f). Consequently, ROS-scavenging capability of TLH NP was mainly contributed by Tempol. This is in line with the previous finding that Tempol is a superoxide dismutase-mimetic agent capable of eliminating ROS[34,36,42].

**Anticoagulant, thrombolytic, and thrombus-binding effects.** Quantification by the chromogenic anti-factor Xa (FXa) assay showed that anticoagulant activity of pristine LMWH, LH NP, and TLH NP was $113.5 \pm 9.3$, $60.8 \pm 7.2$, and $56.4 \pm 8.1$ IU/mg, respectively (Fig. 2r). Nevertheless, the anticoagulant activity of LH NP and TLH NP, expressed as the content of LMWH, was almost comparable to that of LMWH, indicating that nanotherapies did not compromise the LMWH activity. In vitro anticoagulant tests showed that LMWH, LH NP, and TLH NP significantly and comparably prolonged prothrombin time (PT), activated partial thromboplastin time (APTT), and thrombin time (TT), as compared to PBS (Fig. 2s–u). In addition, LMWH, LH NP, and TLH NP notably promoted the formation of thrombin-antithrombin complex (TAT), with more significant effects found for two nanotherapies (Fig. 2v). Platelet activation and aggregation are critical risk factors for the formation of DVT[43]. We found that pretreatment of rat whole blood with Tempol, LMWH, LH NP, or TLH NP considerably suppressed the $CaCl_2$-induced clot formation (Supplementary Fig. 4a), showing the smallest clot in the TLH NP group. Further quantitative analysis of blood-clotting index (BCI) also revealed the best anti-thrombotic capability of TLH NP (Fig. 2w). Consistently, SEM observation indicated that two nanotherapies, (in particular, TLH NP) more effectively inhibited thrombin-induced platelet aggregation (Fig. 2x). Moreover, TLH NP notably suppressed platelet activation, as implicated by dramatically reduced pseudopods. This was also confirmed by confocal microscopy observation of platelets stained with FITC-labeled anti-CD61 (Supplementary Fig. 4b).

We then examined in vitro thrombolytic capability of different formulations. Direct observation showed notable thrombolytic effects on rat blood clots after treatment with various agents (Supplementary Fig. 4c), and quantitative analysis of whole blood clot lysis demonstrated much better efficacy of TLH NP and urokinase (URK, a positive control with potent thrombolytic activity) (Supplementary Fig. 4d). Using rat platelet-rich plasma, we also found that either LH NP or TLH NP could effectively bind to thrombin-activated platelet aggregates (Supplementary Fig. 4e, f). Of note, TLH NP displayed significantly higher targeting capacity to activated and aggregated platelets. By contrast, LMWH, LH NP, and TLH NP showed similar binding capability to non-activated platelets (Supplementary Fig. 5). This thrombus-targeting capability of LMWH-derived nanotherapies can be attributed to the high binding capacity of LMWH with activated platelets and

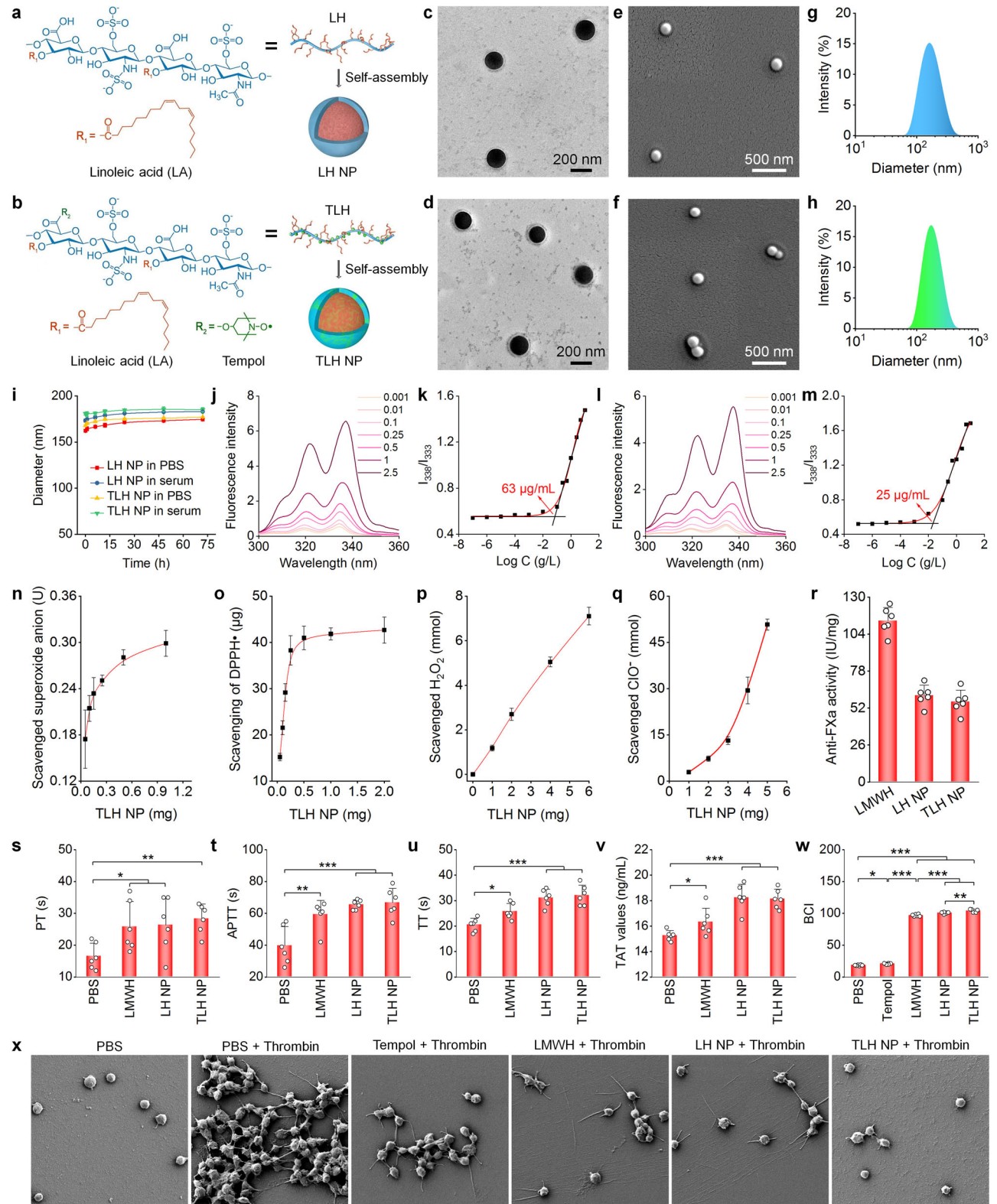

tissue factors of thrombi[44]. According to previous findings, platelet activation is mainly mediated by the interaction between thrombin and platelet glycoprotein Ib (GP-Ib)[45]. LMWH can replace the binding site of thrombin with GP-Ib to inhibit thrombin-mediated platelet activation[44]. Also, LMWH can bind to the C-terminus of tissue factor pathway inhibitor (TFPI), while TFPI further interacts with tissue factor to suppress its expression in thrombi[44]. On the other hand, oxidation of protein disulfide isomerase (PDI) on the platelet surface plays a key

role in platelet activation and aggregation[46]. As a ROS-scavenging agent, Tempol can covalently bind to PDI, thereby inhibiting PDI oxidation[47]. In addition, relatively small negative ζ-potential values of TLH NP in serum might be beneficial for enhanced thrombus or activated platelet targeting[48], although in-depth studies remain to be conducted to address mechanisms underlying the desirable targeting efficiency of TLH NP. In a separate study, we compared anticoagulant effects of two nanotherapies with the same doses of LA and a mixture

**Fig. 2 | Design, construction, and characterization of multifunctional antithrombotic nanotherapies based on LMWH-derived materials. a, b** Schematic illustration of nanotherapies self-assembled by linoleic acid (LA)-conjugated LMWH (LH) (**a**) or Tempol/LA-conjugated LMWH (TLH) (**b**). **c–h** Typical TEM images (**c, d**), SEM images (**e, f**), and size distribution profiles (**g, h**) of LH NP (**c, e, g**) and TLH NP (**d, f, h**). **i** Changes in the mean diameter of LH NP and TLH NP after incubation with PBS or serum for predetermined time periods. **j–m** Typical excitation fluorescence spectra (**j, l**) and the corresponding plots of the intensity ratio $I_{338}/I_{333}$ as a function of Log C (**k, m**) for pyrene in the presence of increasing concentrations (g/L) of LH (**j, k**) or TLH (**l, m**). **n–q** Dose-dependent elimination of superoxide anion (**n**), DPPH radical (**o**), $H_2O_2$ (**p**), and hypochlorite (**q**) by TLH NP. **r** The anti-FXa activity of

LMWH, LH NP, and TLH NP. **s–w** Effects of different formulations on PT (**s**), APTT (**t**), TT (**u**), TAT (**v**), and BCI (**w**). Of note, BCI increases with increased free hemoglobin in solutions. A higher BCI means stronger anti-thrombotic activity for the examined formulations. **x** SEM images showing thrombin-stimulated platelets after separate treatment with PBS, Tempol, LMWH, LH NP, or TLH NP. Platelets treated with PBS alone served as the normal control. Data in **c–h, x** are representative of six independent samples. Scale bar, 3 μm. Data in **i, n–w** are mean ± s.d. (*n* = 6 independent samples). Statistical significance was assessed by one-way ANOVA with post hoc LSD tests. **p* < 0.05, ***p* < 0.01, ****p* < 0.001. Source data are provided as a Source Data file.

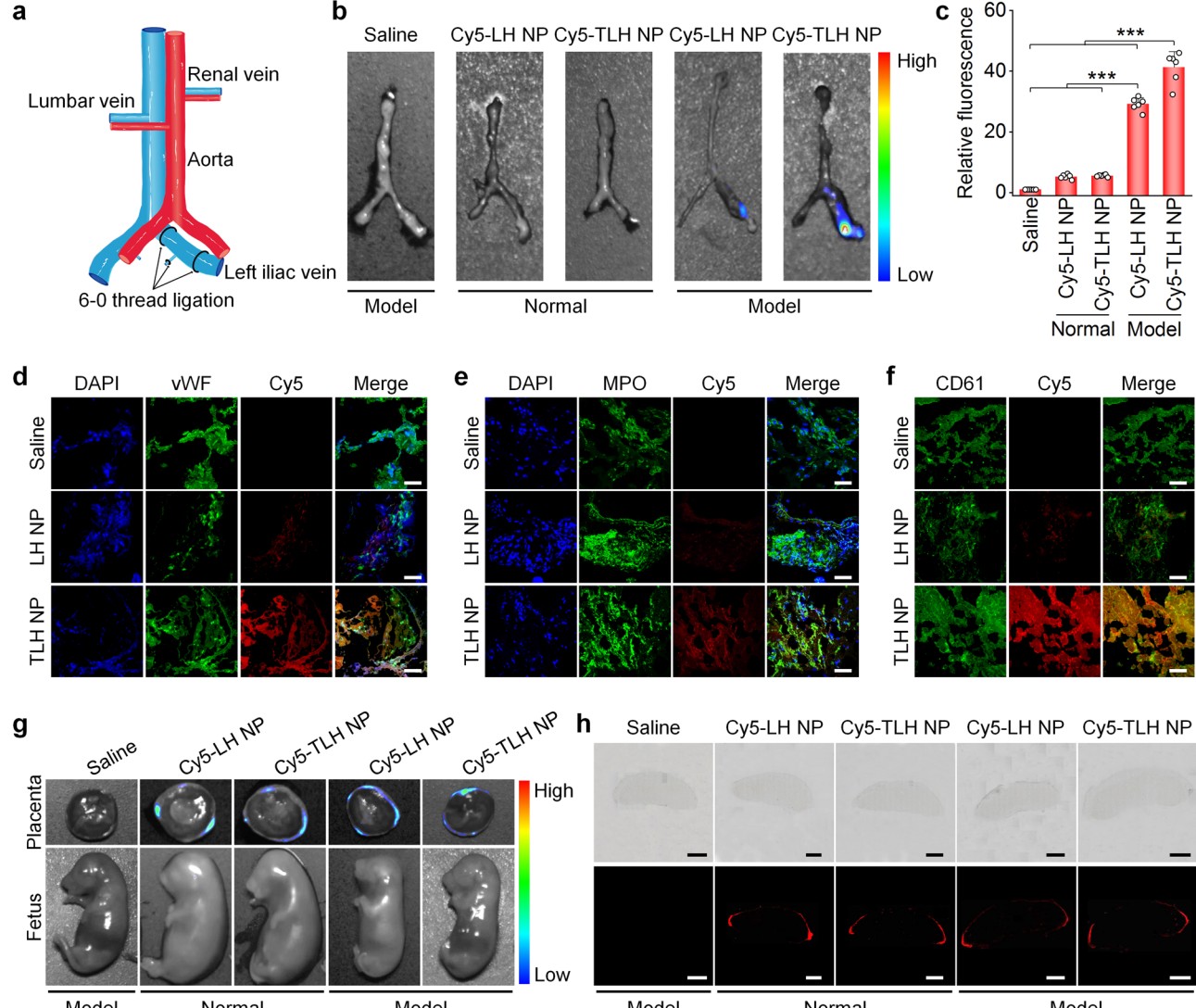

**Fig. 3 | Targeting effects of LH NP and TLH NP in pregnant rats with stenosis-induced DVT. a** A schematic diagram showing the establishment of a stenosis-induced DVT model in pregnant rats. **b, c** Ex vivo fluorescence images (**b**) and quantitative data (**c**) show the accumulation of Cy5-LH NP or Cy5-TLH NP in normal veins or veins with stenosis-induced thrombosis at 4 h after i.v. injection in pregnant rats at G10. **d–f** Immunofluorescence analysis of the co-localization of Cy5-LH NP or Cy5-TLH NP with vWF-positive endothelial cells (**d**), MPO-positive neutrophils (**e**), and CD61-positive platelets (**f**) in cryosections of left iliac veins of DVT rats. Pregnant rats (at G10) with stenosis-induced DVT were administered with Cy5-LH NP or Cy5-TLH NP via i.v. injection. At 4 h after administration, rats were euthanized

and left iliac veins were isolated for analysis. Scale bars, 40 μm. **g** Ex vivo imaging of placentas and fetuses at 4 h after i.v. injection of Cy5-LH NP or Cy5-TLH NP in normal or DVT pregnant rats (at G15). **h** Microscopic observation of placental sections. Upper panel, bright field images; lower panel, fluorescence images. Scale bars, 2 mm. In all cases, pregnant rats with thrombosis were treated with the same volume of saline in the saline group. Data in **b, d–h** are representative of six independent samples. Data in **c** are mean ± s.d. (*n* = 6 independent samples). Statistical significance was assessed by one-way ANOVA with post hoc LSD tests. ****p* < 0.001. Source data are provided as a Source Data file.

of Tempol, LA, and LMWH (i.e., TLH bolus). It was found that LA did not suppress the thrombin-induced platelet aggregation and $Ca^{2+}$-mediated clot formation or promote blood clot lysis (Supplementary Fig. 6). Whereas the TLH bolus showed notable anticoagulant activity, a significantly stronger efficacy was achieved by TLH NP, compared to the TLH bolus.

Collectively, these results demonstrated that LH NP and TLH NP can effectively prevent platelet aggregation and activation as well as promote thrombolysis. The anti-thrombotic activity of LMWH was notably potentiated by engineering into NPs, particularly for the ROS-scavenging LMWH nanotherapy, by enhancing thrombus-binding capability.

**Endothelial protective effects.** Endothelial cells play an important role in the thrombus formation by expressing different molecules associated with thrombosis[49]. Both confocal microscopic observation and flow cytometric quantification revealed time-dependent internalization of Cy5-labeled LH NP (Cy5-LH NP) and Cy5-labeled TLH NP (Cy5-TLH NP) in human umbilical vein endothelial cells (HUVECs) (Supplementary Fig. 7). Of note, endocytosed Cy5-LH NP and Cy5-TLH NP were mainly transported through the endolysosomal pathway, since considerable co-localization of Cy5 fluorescence with Lyso-Tracker fluorescence was observed after incubation for various time periods. In addition, clearly dose-response cellular uptake profiles were found for Cy5-LH NP and Cy5-TLH NP (Supplementary Fig. 8).

Overproduced ROS in endothelial cells can lead to oxidative damage of the endothelium, which further promotes thrombosis[29]. Therefore, we assessed the protective effects of LH NP and TLH NP on $H_2O_2$-induced oxidative stress in HUVECs. Pretreatment with Tempol, LMWH, LH NP, or TLH NP notably inhibited $H_2O_2$-induced intracellular ROS generation in HUVECs, as implicated by fluorescence images and flow cytometric analysis (Supplementary Fig. 9a, b). Nevertheless, the most effective anti-oxidative activity was observed for LH NP and TLH NP. In particular, the nanotherapy TLH NP even reduced ROS to a level comparable to the normal control. The TLH bolus exhibited a similar anti-oxidative effect in HUVECs (Supplementary Fig. 10a, b), but free LA could not inhibit $H_2O_2$-mediated oxidative stress. Consistent with the notable anti-oxidative activity of LH NP and TLH NP, both nanotherapies more effectively attenuated $H_2O_2$-induced apoptosis of HUVECs, as compared to Tempol and LMWH (Supplementary Fig. 9c, d). LH NP attenuated $H_2O_2$-induced apoptosis of HUVECs, largely by regulating intracellular $Ca^{2+}$ via the LMWH moiety[50,51]. TLH NP more effectively inhibited ROS-mediated HUVECs apoptosis than LH NP, resulting from the synergistic effects of LMWH and Tempol. Likewise, TLH NP showed a much better antiapoptotic activity than the TLH bolus (Supplementary Fig. 10c, d). Mechanistically, endothelial injury resulting from oxidative stress and apoptosis of endothelial cells directly contributes to the development of DVT[52]. Accordingly, LH NP and TLH NP can prevent and treat DVT by their anti-oxidative, anti-apoptotic, and anti-inflammatory effects.

After endothelial dysfunction, recruitment, migration, and proliferation of endothelial cells at the injury site are very important for regulation of endothelial homeostasis[53]. Transwell migration assay suggested that LMWH-derived nanotherapies more effectively promoted the migration of HUVECs (Supplementary Fig. 9e, f). By contrast, Tempol and LMWH showed notably low activity. This prominent migration-promoting effect of LH NP and TLH NP was also confirmed by the wound-healing assay (Supplementary Fig. 9g, h). In both cases, LA itself showed no significant activity (Supplementary Fig. 10e–h). In addition, TLH NP was more effective than the TLH bolus. Since LMWH itself also exhibits antioxidative and anti-inflammatory effects as well as promotes endothelial cell migration and wound healing in different cells and animal models[54,55], our findings demonstrated that the bioactivity of LMWH can be notably potentiated by introducing functional moieties and assembling into NPs.

## In vivo thrombus targeting in pregnant rats with stenosis-induced DVT

To examine in vivo targeting capability of nanotherapies, a stenosis-induced DVT model was established by ligation of the left iliac vein in pregnant rats at the 10th gestational day (i.e., G10; Fig. 3a and Supplementary Fig. 11). At 6 h after ligation, 89.7% of stenosis in the left iliac vein could form stable thrombi (Supplementary Fig. 12), which reduced blood flow by 74.5–84.3% according to Doppler ultrasound. Accordingly, different formulations were administered at this time point for in vivo targeting and therapeutic evaluations. At 4 h after intravenous (i.v.) injection of Cy5-labeled NPs based on LH or TLH, notable fluorescent signals were observed at the thrombus sites of the left iliac veins isolated from DVT pregnant rats (Fig. 3b). Quantitative analysis showed a significantly higher accumulation of Cy5-TLH NP at the thrombus site than that of Cy5-LH NP. By contrast, the iliac vein from normal rats treated with fluorescent NPs displayed slight fluorescence (Fig. 3c). Fluorescence observation on cryosections indicated the co-localization of Cy5-labeled NPs with vWF (a marker for endothelial cells), myeloperoxidase (MPO)-positive neutrophils, and CD61⁺ platelets, with more significant fluorescence in the Cy5-TLH NP group (Fig. 3d–f). These results demonstrated that i.v. administered LH NP and TLH NP can accumulate in thrombi of iliac veins via passive targeting. Of note, TLH NP exhibited a much better thrombus-targeting effect due to its high binding capacity with activated and aggregated platelets, agreeing with in vitro binding results.

The placenta needs to be carefully examined during pregnancy, in view of its critical role in fetal growth and development[56,57]. After i.v. injection of either Cy5-LH NP or Cy5-TLH NP in pregnant rats, ex vivo imaging indicated that Cy5 fluorescence signals mainly distributed at the decidua of isolated placentas for both normal and DVT rats (Fig. 3g), while almost no fluorescence was detected in fetuses. Consistently, microscopic observation of placental sections showed that fluorescence signals of Cy5-LH NP and Cy5-TLH NP mainly distributed at the placental edge, i.e., the decidua (Fig. 3h and Supplementary Fig. 13). These results suggested that i.v. delivered LH NP and TLH NP cannot cross the placental barrier. Therefore, both nanotherapies may not directly cause side effects on fetuses.

## Therapeutic effects of LH NP and TLH NP in pregnant rats with stenosis-induced DVT

Before therapeutic studies, the potential bleeding risk of different formulations was first examined in a tail vein bleeding model in pregnant rats. After i.v. injection, both nanotherapies (LH NP and TLH NP) showed negligible effects on bleeding (Supplementary Fig. 14). By contrast, i.v. injection of LMWH at 5 mg/kg caused significantly increased bleeding, and 2 among 8 rats even died due to excessive blood loss, although LMWH at 3 mg/kg exhibited no significant effects on bleeding. Of note, LMWH delivered via subcutaneous (s.c.) injection at 5 mg/kg showed significantly lower bleeding, compared to that of i.v. LMWH. Accordingly, LMWH injected subcutaneously (the clinically employed administration route) was initially used as a control, while other formulations were delivered via i.v. injection in following animal studies (Supplementary Fig. 11). At 6 h after ligation of the left iliac veins in pregnant rats, different formulations were administered (Fig. 4a). After 5 days of treatment, direct observation of isolated iliac veins showed the presence of thrombi in the model, Tempol, and LMWH groups (Fig. 4b). By contrast, the nanotherapy groups displayed notably attenuated thrombosis. Whereas treatment with all examined formulations significantly decreased thrombus weight and length (Fig. 4c, d), more prominent effects were achieved by two nanotherapies, particularly TLH NP. Examination on hematoxylin and eosin (H&E)-stained sections also revealed substantial thrombolytic

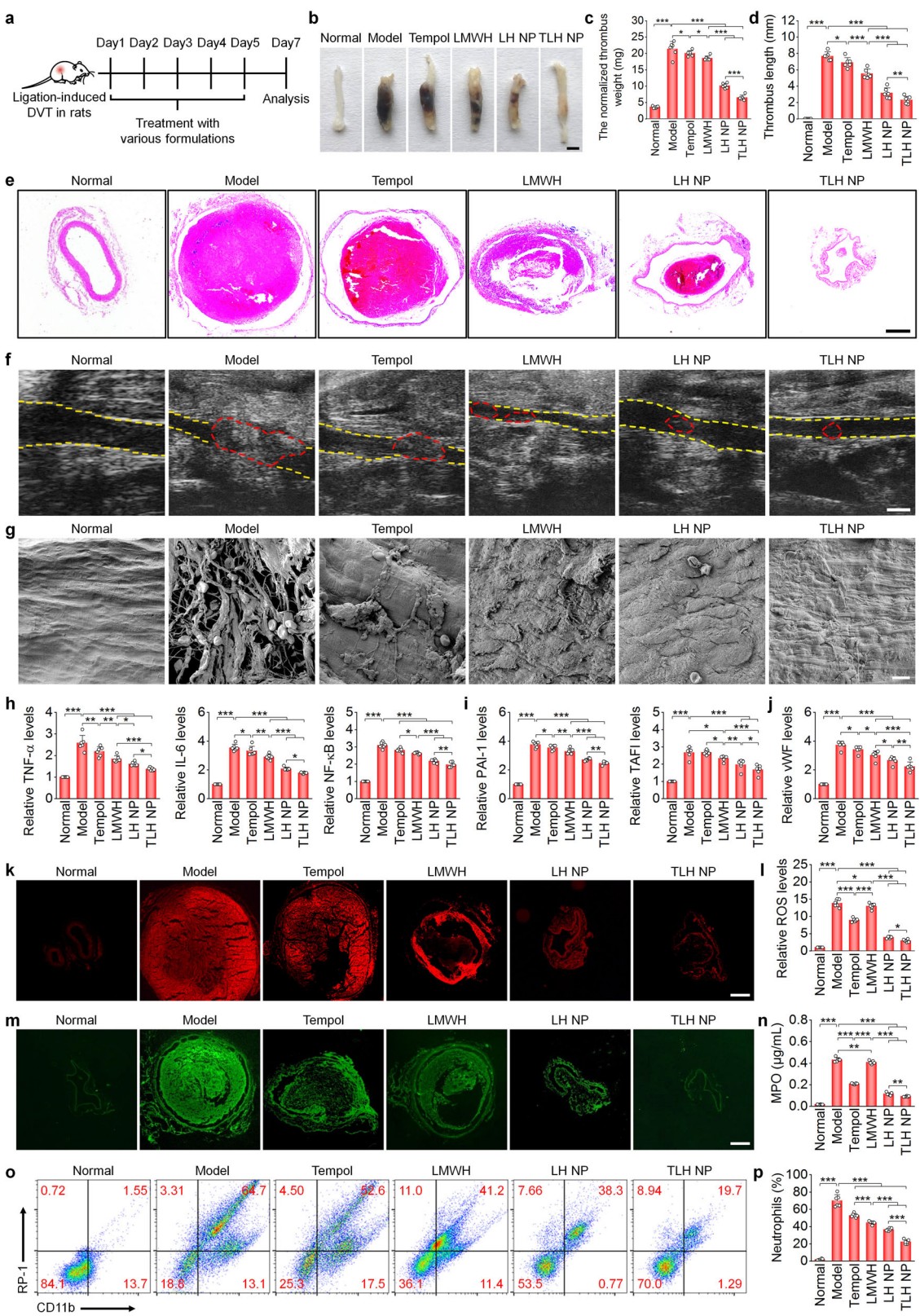

effects after treatment with LMWH formulations, with the best efficacy for the TLH NP group (Fig. 4e). Consistently, ultrasound imaging indicated that blood clots were almost completely eliminated and efficient recanalization was achieved after nanotherapy treatment (Fig. 4f), while Tempol and LMWH only partly reduced thromboembolism. These results demonstrated that the engineered nanotherapies, in particular TLH NP, can effectively dissolve the formed thrombus and inhibit the subsequent development of thrombosis in the iliac vein of pregnant rats.

Based on the above promising results, mechanisms underlying therapeutic advantages of LMWH-derived nanotherapies were examined. In normal pregnant rats (at G10), we found sustained high levels of anti-FXa in plasma for more than 24 h after a single i.v. administration of LH NP or TLH NP (Supplementary Fig. 15a). By contrast, anti-FXa

**Fig. 4 | Therapeutic effects of LH NP and TLH NP in pregnant rats with stenosis-induced DVT. a** Schematic illustration of the treatment regimens. **b** Representative digital photos of left iliac veins with thrombi isolated from pregnant rats after treatment with different formulations. Scale bar, 1 mm. **c, d** The normalized weight (**c**) and length (**d**) of thrombi in left iliac veins after different treatments. **e** H&E-stained histopathological sections of left iliac veins with thrombi. Scale bar, 400 µm. **f** Ultrasound images of left iliac veins. The yellow dashed lines indicate the vascular endothelium, while the red dashed lines indicate thrombi. Scale bar, 1 mm. **g** SEM observation of the endothelial surface of left iliac veins with thrombi and after different treatments. Scale bar, 10 µm. **h–j** Relative mRNA levels of TNF-α, IL-6, NF-κB, PAI-1, TAFI, and vWF in left iliac veins. **k, l** Fluorescence images of DHE-stained cryosections (**k**) and quantitative analysis of relative ROS levels (**l**) in left iliac veins. Scale bar, 400 µm. **m, n** Immunofluorescence images showing MPO-

positive neutrophils in cryosections (**m**) and quantified MPO levels (**n**) in left iliac veins. Scale bar, 400 µm. **o, p** Representative flow cytometric profiles (**o**) and quantified levels (**p**) of CD11b⁺/RP-1⁺ neutrophils in left iliac veins. In all these studies, left iliac veins of pregnant rats at G10 were ligated to induce DVT. At 6 h after the formation of stenosis-induced thrombi, diseased rats were daily administered with saline (the model group), free Tempol (8 mg/kg, *i.v.*), free LMWH (5 mg/kg, s.c.), LH NP (at 5 mg/kg of LMWH, i.v.), or TLH NP (at 5 mg/kg of LMWH, i.v.) for 5 days. In the normal group, healthy pregnant rats with sham operation were treated with saline. Data in **b, e–g, k, m, o** are representative of six independent samples. Data in **c, d, h–j, l, n, p** are mean ± s.d. (*n* = 6 independent samples). Statistical significance was assessed by one-way ANOVA with post hoc LSD tests. \**p* < 0.05, \*\**p* < 0.01, \*\*\**p* < 0.001. Source data are provided as a Source Data file.

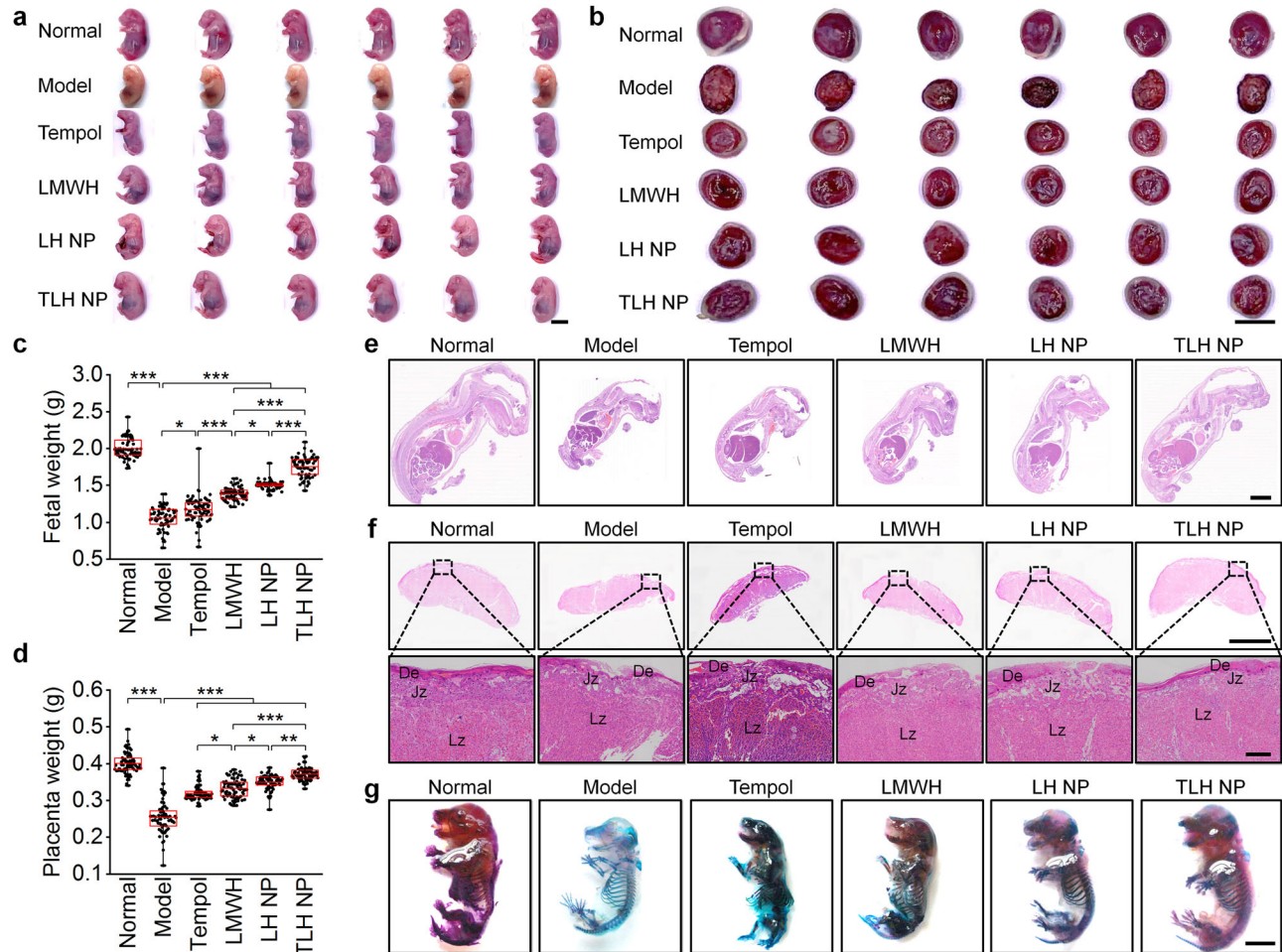

**Fig. 5 | Inhibition of DVT-induced embryonic developmental disorders and early fetal growth delay in pregnant rats by LMWH-derived nanotherapies. a, b** Digital photos show gross morphological appearance of representative rat fetuses (**a**) and placentas (**b**) from normal or DVT pregnant rats at G17. One fetus or placenta was randomly selected from each rat of different groups (*n* = 6 independent animals). Scale bar, 1 cm. **c, d** Fetal (**c**) and placental (**d**) weights of different groups. Both fetuses and placentas were from 6 pregnant rats in each group (independent animals). **e** H&E-stained histological sections of isolated fetuses at G17. Scale bar, 4 mm. **f** Whole slide and high-magnification images of H&E-stained placental sections. Scale bars, 4 mm (upper) and 500 µm (lower). De, decidua; Jz, junctional zone; Lz, labyrinth zone. **g** Whole-mount skeletal staining of fetuses via Alizarin red and Alcian blue in different groups at G17. Scale bar, 500 µm. In these

studies, stenosis-induced DVT in pregnant rats was established at G10. After thrombus formation, pregnant rats in the model group were treated with saline alone, while other groups were separately administered with free Tempol (8 mg/kg, i.v.), LMWH (5 mg/kg, s.c.), LH NP (at 5 mg/kg of LMWH, i.v.), or TLH NP (at 5 mg/kg of LMWH, i.v.) for 5 days. In the normal group, healthy pregnant rats with the sham operation were treated with saline. All fetuses and placentas were excised from uteruses at G17 for analyses. Data in box plots **c, d** show the mean value and extend from 25 to 75%, while the whiskers extend from the minimal to maximal values, which are based on all fetuses and placentas from 6 pregnant rats in each group. Statistical significance was assessed by one-way ANOVA with post hoc LSD tests. \**p* < 0.05, \*\**p* < 0.01, \*\*\**p* < 0.001. Data in **e–g** are representative of six independent samples. Source data are provided as a Source Data file.

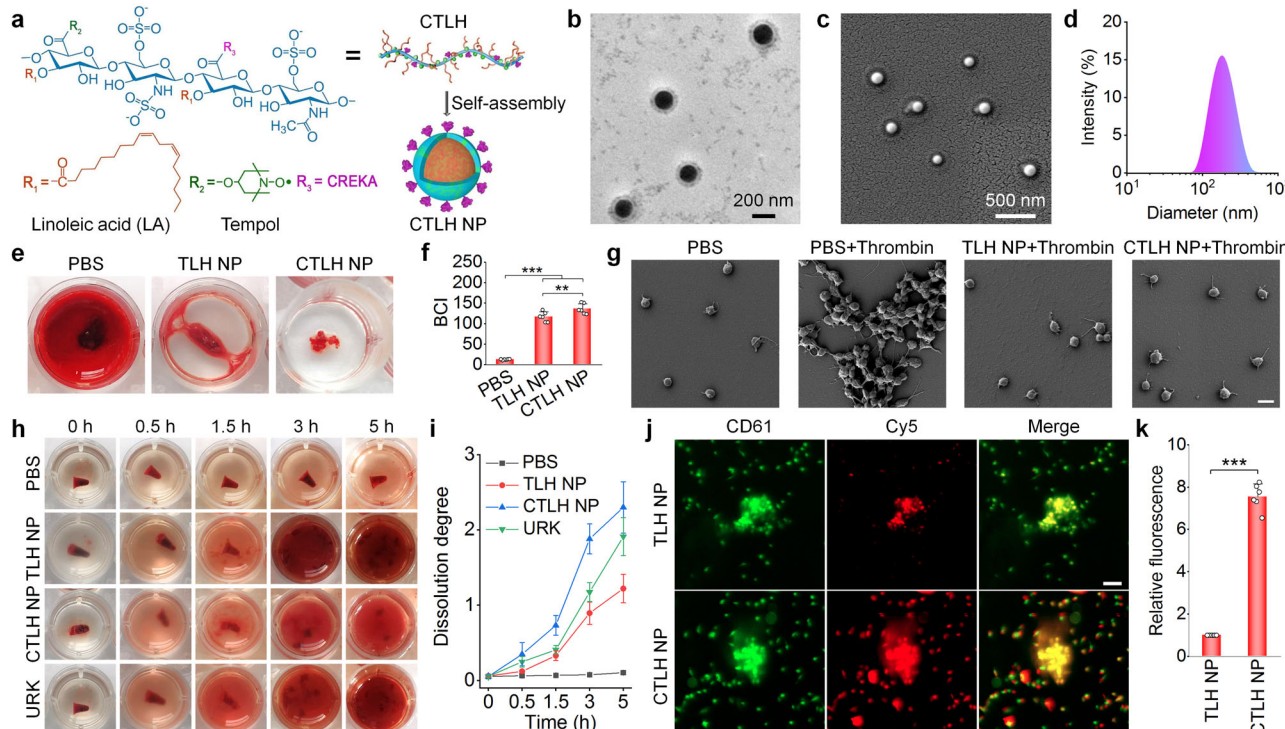

**Fig. 6 | Design, construction, characterization, and in vitro anti-coagulant/anti-thrombotic effects of an active targeting nanotherapy CTLH NP. a** Schematic illustration of engineering of CTLH NP. **b–d** TEM (**b**) and SEM (**c**) images as well as the size distribution (**d**) of CTLH NP. **e, f** Typical digital photographs (**e**) and quantified BCI values (**f**) showing anti-coagulant effects of TLH NP and CTLH NP. Blood samples were incubated with different formulations and then coagulation was induced by CaCl$_2$ for 5 min. After washing with deionized water, the absorbance of hemoglobin was determined to calculate BCI. **g** SEM observation of platelet aggregation. After pretreatment with PBS, TLH NP, or CTLH NP for 0.5 h, platelet aggregation was induced by thrombin. Platelets treated with PBS alone served as a negative control. Scale bar, 3 μm. **h, i** Representative digital photos (**h**) and quantified dissolution degrees (**i**) of thrombi at various time points after treatment with PBS, TLH NP, CTLH NP, or URK. **j, k** Confocal microscopy images (**j**) and quantitative analysis (**k**) indicate binding interactions between coagulated platelets and Cy5-TLH NP or Cy5-CTLH NP. Coagulation was induced by thrombin, and platelets were stained with FITC-labeled anti-CD61 antibody (green). Scale bars, 10 μm. Data in **b–e, g, h, j** are representative of six independent samples. Data in **f, i, k** are mean ± s.d. ($n = 6$ independent samples). Statistical significance was assessed by one-way ANOVA with post hoc LSD tests (**f**) and unpaired $t$-test (**k**). **$p < 0.01$, ***$p < 0.001$. Source data are provided as a Source Data file.

activity was not detected at 12 h after *s.c.* injection of the same dose of LMWH. The calculated area under the anti-FXa activity-time curve (AUC) of LH NP and TLH NP was 2.2 and 2.4 times higher than that of free LMWH, respectively (Supplementary Fig. 15b). Accordingly, nanotherapies can efficiently enhance bioavailability of LMWH.

As the best anticoagulant surface, the vascular endothelium plays a key role in maintaining vascular homeostasis[58]. After removal of thrombi, SEM observation revealed notable endothelial denudation in the model group, with considerable accumulation of erythrocytes and activated platelets as well as fibrin deposition (Fig. 4g). After different treatments, endothelial injury was substantially alleviated. Especially for the TLH NP group, almost normal endothelium was observed, without deposition of erythrocytes and platelets. This suggested that treatment with nanotherapies effectively promoted endothelial homeostasis, in line with the in vitro finding that both LH NP and TLH NP significantly promoted endothelial cell migration and accelerated wound healing (Supplementary Fig. 9e–h).

It has been well recognized that inflammation and oxidative stress are closely related to the pathogenesis of thrombosis, by inducing endothelial dysfunction, increasing pro-thrombotic factors, and causing platelet activation and aggregation[29,30]. We found that mRNA levels of tumor necrosis factor (TNF)-α, interleukin (IL)-6, and nuclear factor-κB (NF-κB) in iliac veins were significantly reduced by nanotherapies (Fig. 4h). Also, we detected notably decreased mRNA expression of plasminogen activator inhibitor (PAI)-1 and thrombin-activatable fibrinolysis inhibitor (TAFI; Fig. 4i), two pro-thrombotic proteins involved in DVT[59,60]. In addition, the abnormal expression of

von Willebrand factor (vWF, a biomarker of endothelial injury) in iliac veins was significantly inhibited by nanotherapies (Fig. 4j). The notably decreased pro-inflammatory cytokines, pro-thrombotic proteins, and vWF in iliac veins were also confirmed by immunohistochemistry and Western blot analyses (Supplementary Figs. 16 and 17). In addition, treatment with Tempol and nanotherapies significantly reduced ROS and MPO in iliac veins (Fig. 4k–n). Consistently, two nanotherapies most effectively reduced neutrophil counts in left iliac vein tissues and peripheral blood (Fig. 4o, p and Supplementary Fig. 18).

Further, we compared therapeutic effects of LH NP and TLH NP with free LA, i.v. LMWH at 3 mg/kg (a safe dose), and a TLH bolus. As expected, a low-dose i.v. LMWH and the TLH bolus significantly inhibited DVT in pregnant rats and provided considerable endothelial protective effects (Supplementary Fig. 19), concomitant with decreased mRNA levels of pro-thrombotic factors (PAI-1 and TAFI) and pro-inflammatory cytokines (TNF-α and IL-6) in iliac veins (Supplementary Fig. 20a–d). LA showed no anti-thrombotic activity, although it inhibited the inflammatory response to a certain degree. In all these cases, treatment with TLH NP afforded the best efficacy. Moreover, we found that treatment with all LMWH-containing formulations significantly increased the expression of tissue plasminogen activator (tPA) (Supplementary Fig. 20e), a well-recognized and powerful endogenous thrombolytic agent. Since PAI-1 specifically inhibits the tPA expression[61], these results suggested that LMWH nanotherapies can notably promote thrombolysis by increasing free tPA via reducing the PAI-1 expression. In addition, both nanotherapies and the TLH

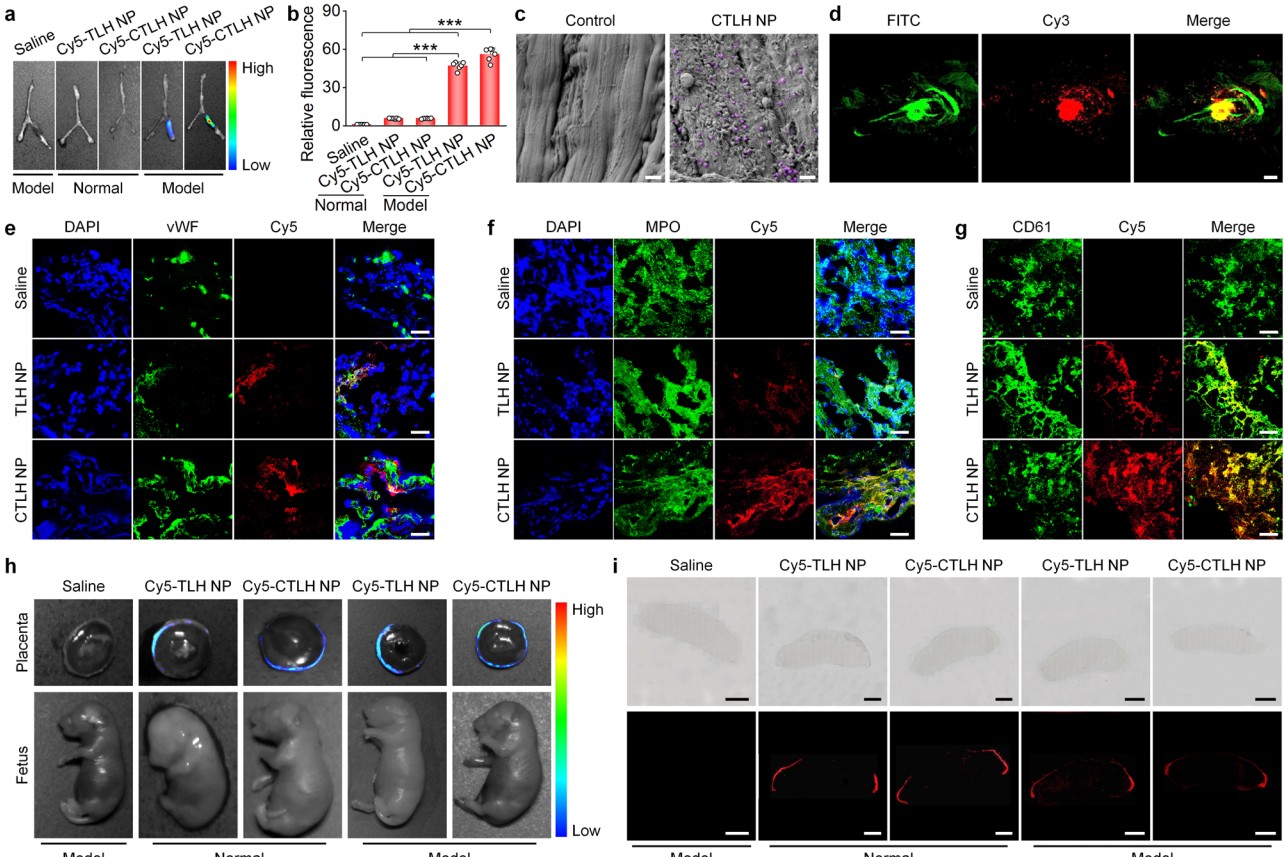

**Fig. 7 | In vivo targeting effects of CTLH NP in pregnant rats with stenosis-induced DVT. a, b** Ex vivo fluorescence images (**a**) and quantitative data (**b**) showing the accumulation of Cy5-TLH NP and Cy5-CTLH NP in normal veins and veins with thrombi at 4 h after injection in pregnant rats at G10. For the saline group, DVT rats were treated with saline. **c** Typical SEM images indicate the presence of CTLH NP on the endothelial surface of the left iliac vein with thrombosis. The control group was treated with saline. CTLH NP is illustrated in purple. Scale bars, 1 μm. **d** Two-photon microscopy observation of the co-localization of Cy3-CTLH NP with FITC-labeled fibrin in the left iliac vein with stenosis-induced DVT at 4 h after i.v. injection of Cy3-CTLH NP. Scale bar, 50 μm. **e–g** Immunofluorescence analysis of the co-localization of Cy5-TLH NP or Cy5-CTLH NP with vWF-positive endothelial cells (**e**), MPO-positive neutrophils (**f**), and CD61-positive platelets (**g**) in

cryosections of left iliac veins. Pregnant rats (at G10) with DVT were administered with Cy5-TLH NP or Cy5-CTLH NP by i.v. injection. At 4 h after administration, rats were euthanized and left iliac veins were isolated for analysis. Scale bars, 40 μm. **h** Ex vivo imaging of placentas and fetuses at 4 h after i.v. injection of Cy5-TLH NP or Cy5-CTLH NP in normal or DVT pregnant rats (at G15). **i** Microscopic observation of placental sections. Upper panel, bright field images; lower panel, fluorescence images. Scale bars, 2 mm. In the saline group, DVT rats were treated with saline. Data in **b** are mean ± s.d. (*n* = 6 independent samples). Statistical significance was assessed by one-way ANOVA with post hoc LSD tests. \*\*\**p* < 0.001. Data in **a, c–i** are representative of six independent samples. Source data are provided as a Source Data file.

bolus, particularly TLH NP, significantly decreased the levels of fibrin, platelets, and leukocytes in thrombi (Supplementary Fig. 21).

Taken together, LMWH-derived nanotherapies can effectively eliminate the formed thrombus in the deep vein and also efficaciously prevent the subsequent development of DVT in pregnant rats. In addition to prominent thrombolytic and anticoagulant effects, notable anti-oxidative and anti-inflammatory activities of the engineered nanotherapies are responsible for the beneficial outcomes.

**Alleviation of fetal developmental disorders as well as postnatal viability and growth of offspring in pregnant rats with stenosis-induced DVT by LMWH nanotherapies**

DVT will cause hypoxia in the placenta, thereby indirectly impairing placental functions by immunosuppression, delaying trophoblast development, and inducing apoptosis of trophoblast cells[62]. Furthermore, hypoxia of placental trophoblast cells can result in increased coagulation proteins and deposition of microthrombi in placental vessels[15]. These effects of DVT collectively cause intrauterine fetal growth restriction (IUGR)[63]. We further examined whether LMWH nanotherapies can reduce adverse pregnancy outcomes associated with DVT. For pregnant rats (at G17) with stenosis-induced DVT, we

found significantly reduced fetal and placental weights, but no significant difference in the average litter size, as compared to the healthy pregnant rats (Fig. 5a–d). Whereas Tempol and LMWH reversed weight loss of fetuses and placentas to a certain degree, the most desirable effects were achieved by both nanotherapies, especially TLH NP. Examination on H&E-stained histological sections of fetuses and placentas afforded similar results (Fig. 5e, f). Of note, structures of the decidua and junctional zones in the model group were partially damaged, indicating that the placenta was not fully developed in the third trimester, while treatment with nanotherapies (especially TLH NP) restored the integrity of the placental structure.

After removal of the internal organs of the rat fetus and dehydration, skeletal staining via Alizarin red and Alcian blue was performed to determine the development status of fetuses. Compared with those of the normal group, fetuses in the model group showed much more cartilage (Alcian blue staining), with a considerably lower content of bone (Alizarin red staining) (Fig. 5g). In particular, the occipital bone was considerably low in the model group. This revealed the notably delayed development of fetuses in DVT rats. Treatment with LH NP and TLH NP effectively improved the abnormal bone development of fetuses. Collectively, these results demonstrated that

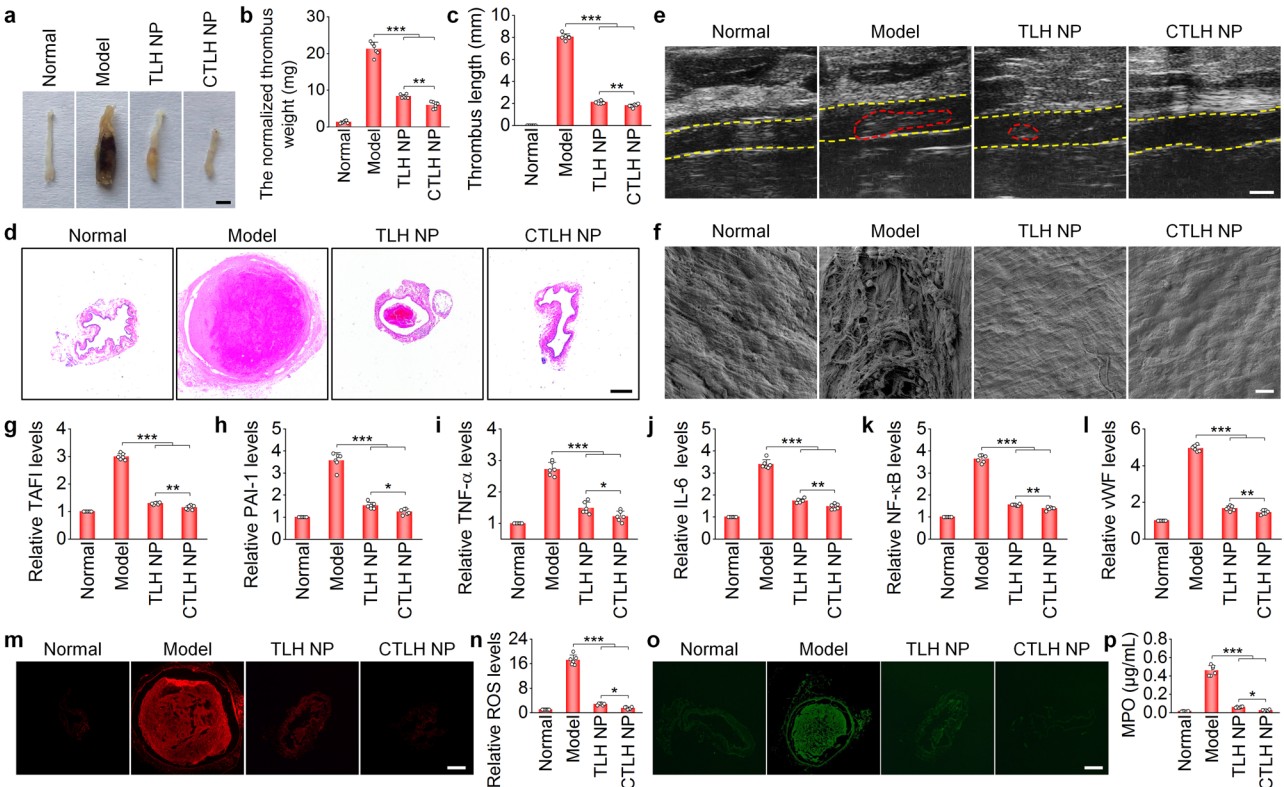

**Fig. 8 | Therapeutic effects of the active targeting nanotherapy CTLH NP in pregnant rats with stenosis-induced DVT. a** Representative digital photos showing thrombi in left iliac veins isolated from DVT pregnant rats after treatment with different formulations. Scale bar, 1 mm. **b, c** The normalized weight (**b**) and length (**c**) of thrombi in left iliac veins after different treatments. **d** H&E-stained sections of left iliac veins. Scale bar, 400 μm. **e** Ultrasound imaging of left iliac veins. The yellow dashed lines indicate the vascular endothelium, while the red dashed lines indicate thrombi. Scale bar, 1 mm. **f** SEM observation of the luminal surface of left iliac veins after different treatments. Scale bar, 10 μm. **g–l** Relative mRNA levels TAFI, PAI-1, TNF-α, IL-6, NF-κB, and vWF in left iliac veins of different groups. **m, n** Fluorescence images of DHE-stained cryosections (**m**) and quantitative

analysis of relative ROS levels (**n**) in left iliac veins. Scale bar, 400 μm. **o, p** Immunofluorescence analysis of MPO-positive neutrophils in cryosections (**o**) and quantification of MPO levels (**p**) in left iliac veins. Scale bar, 400 μm. In all these studies, left iliac veins of pregnant rats at G10 were ligated to induce DVT. At 6 h after the formation of thrombosis, DVT rats were daily administered with saline (the model group), TLH NP (5 mg/kg of LMWH, i.v.), or CTLH NP (5 mg/kg of LMWH, i.v.) for 5 days. In the normal group, healthy pregnant rats with sham operation were treated with saline. Data in **a, d–f, m, o** are representative of six independent samples. Data in **b, c, g–l, n, p** are mean ± s.d. (*n* = 6 independent samples). Statistical significance was assessed by one-way ANOVA with post hoc LSD tests. *$p < 0.05$, **$p < 0.01$, ***$p < 0.001$. Source data are provided as a Source Data file.

LMWH nanotherapies can effectively alleviate the DVT-mediated developmental disorders in pregnant rats.

In a separate study, we also determined the postnatal viability and offspring growth profiles. There were no significant differences in postnatal survival of rat offspring in different groups. However, the model group showed a notably high incidence of fetal malformation (Supplementary Fig. 22a). In addition, the growth of rat offspring in the model group was relatively slow (Supplementary Fig. 22b). Examination of H&E-stained sections of offspring lungs at day 21 after birth revealed partially damaged bronchial and alveolar structures in the model group (Supplementary Fig. 23). These abnormalities were significantly alleviated after treatment with different LMWH formulations, especially TLH NP.

### Engineering of an active targeting LMWH nanotherapy
To further improve therapeutic effects of TLH NP by enhancing thrombus-targeting capability, we developed an active targeting nanotherapy by functionalizing TLH NP with CREKA (Fig. 6a), a peptide ligand that can target thrombi by binding with fibrin[37–39]. To this end, CREKA was covalently conjugated onto TLH, giving rise to a multifunctional amphiphile termed as CTLH (Supplementary Fig. 24a). Successful synthesis of CTLH was confirmed by [1]H NMR and EPR spectroscopy (Supplementary Fig. 24b, c). The content of CREKA, Tempol, and LA in CTLH was estimated to be 16.8, 15.1, and 18.9%, respectively. Then the active targeting nanotherapy CTLH NP was also

prepared by self-assembly (Fig. 6a). CTLH NP displayed well-defined spherical shape, with a core-shell structure and uniform distribution (Fig. 6b–d). The mean diameter and ζ-potential of CTLH NP was 176 ± 2 nm and −44 ± 2 mV (Supplementary Fig. 25a), respectively. CTLH NP showed good colloidal stability in PBS and serum (Supplementary Fig. 25b, c), displaying the anti-FXa activity of 51.2 ± 2.1 IU/mg (Supplementary Fig. 25d). Also, CTLH NP could effectively eliminate different types of ROS (Supplementary Fig. 25e–h).

Compared with TLH NP, CTLH NP more effectively inhibited CaCl₂-induced thrombosis and thrombin-induced platelet aggregation (Fig. 6e–g and Supplementary Fig. 26). Likewise, CTLH NP had better thrombolytic capability than that of TLH NP and URK (Fig. 6h, i). Correspondingly, CTLH NP exhibited significantly higher binding capacity to aggregated platelets, compared to TLH NP (Fig. 6j, k). This can be attributed to the high deposition of fibrin around the activated and aggregated platelets[64,65]. Together, these results demonstrated that anticoagulant and thrombolytic effects of LMWH nanotherapies can be further potentiated by affording active targeting capability via CREKA decoration.

### In vivo targeting and efficacies of CTLH NP in pregnant rats with stenosis-induced DVT
We then examined in vivo targeting capability of CTLH NP. At 4 h after i.v. injection of Cy5-labeled CTLH NP in pregnant rats with stenosis-induced DVT, ex vivo imaging showed evident fluorescent signals at

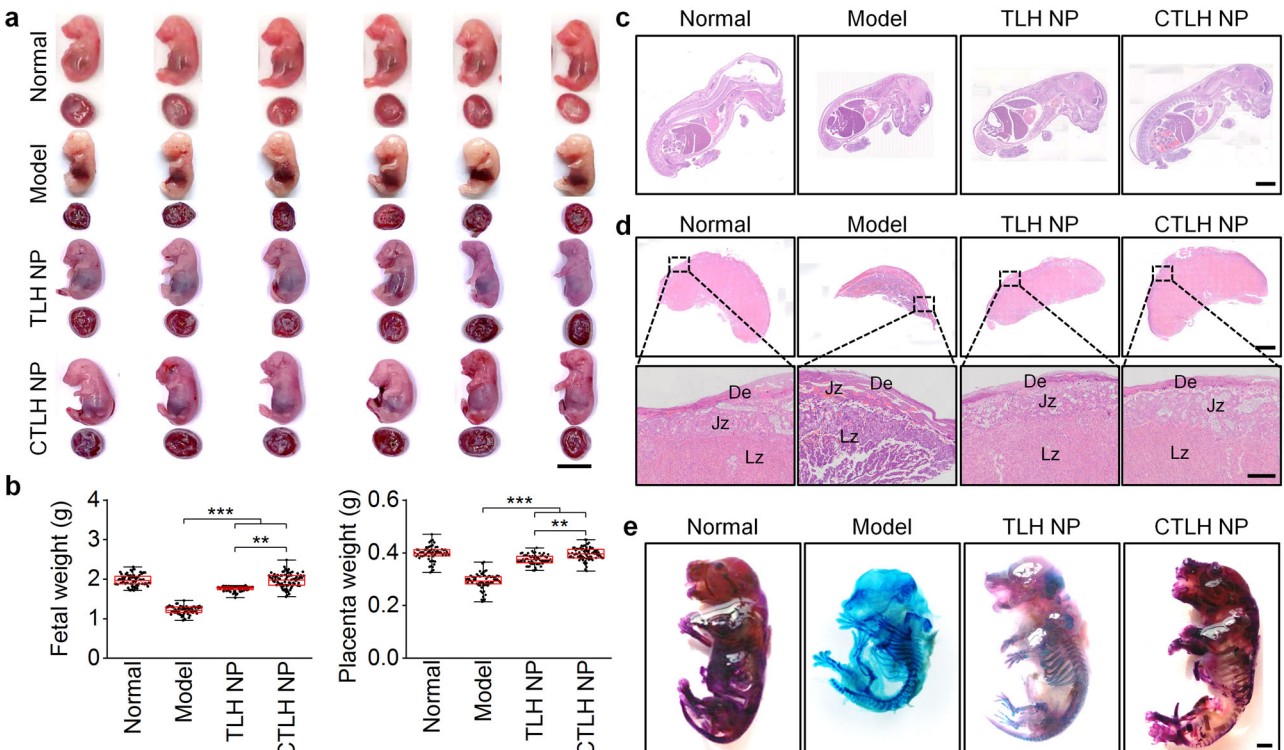

**Fig. 9 | Inhibition of DVT-induced embryonic developmental disorders and early fetal growth delay in pregnant rats by the active targeting nanotherapy CTLH NP. a** Digital photos of representative rat fetuses and placentas from normal or DVT pregnant rats at G17. One fetus or placenta was randomly selected from each rat of different groups (*n* = 6 independent animals). Scale bar, 1 cm. **b** Fetal and placental weights of different groups. Both fetuses and placentas were from 6 pregnant rats in each group (independent animals). **c** H&E-stained histological sections of isolated fetuses at G17. Scale bar, 4 mm. **d** Whole slide and high-magnification images of H&E-stained placental sections. Scale bars, 2 mm (upper) and 500 μm (lower). De, decidua; Jz, junctional zone; Lz, labyrinth zone. **e** Whole-mount skeletal staining of fetuses via Alizarin red and Alcian blue. Scale bar, 200 μm. In this study, stenosis-induced DVT in pregnant rats was established at

G10. After thrombus formation, pregnant rats in the model group were treated with saline, while other two groups were separately administered with TLH NP and CTLH NP at 5 mg/kg of LMWH by daily i.v. injection for 5 days. For the normal group, healthy pregnant rats with the sham operation were treated with saline. All fetuses and placentas were excised from uteruses at G17 for analyses. Data in box plots **b** show the mean value and extend from 25 to 75%, while the whiskers extend from the minimal to maximal values, which are based on all fetuses and placentas from six pregnant rats in each group. Statistical significance was assessed by one-way ANOVA with post hoc LSD tests. **p < 0.01, ***p < 0.001. Data in **c**–**e** are representative of six independent samples. Source data are provided as a Source Data file.

the thrombus site of the iliac vein (Fig. 7a, b). Quantitative analysis indicated that the fluorescent intensity of the Cy5-CTLH NP group was 1.2-fold higher than that of the Cy5-TLH NP group. SEM observation clearly showed the accumulation of CTLH NP on the endothelial surface of iliac vein thrombosis sites (Fig. 7c). Further in vivo two-photon microscopic observation revealed rapid and high efficient accumulation of Cy3-CTLH NP at the iliac vein thrombosis site of pregnant rats, which was significantly co-localized with FITC-labeled fibrin (Fig. 7d and Movie S1). Further immunofluorescence examination of cryosections also indicated the considerable distribution of Cy5-CTLH NP in the iliac vein, which was partly co-localized with endothelial cells, neutrophils, and platelets (Fig. 7e–g). In this case, CTLH NP exhibited a relatively high accumulation in iliac vein thrombosis regions, as compared to TLH NP. Notably, high co-localization of CTLH NP with endothelial cells and neutrophils was mainly due to fibrin deposition on these cells[12]. We also examined the distribution of CTLH NP in fetuses and placentas. At day 5 after DVT was induced in pregnant rats at G15, Cy5-CTLH NP was administered via i.v. injection. At 4 h after delivery, ex vivo imaging indicated fluorescence signals at the decidua of the placenta, while no fluorescence was detected in the fetus (Fig. 7h). Consistently, microscopic observation also revealed peripheral distribution of Cy5-CTLH NP (Fig. 7i and Supplementary Fig. 27). Of note, Cy5-TLH NP and Cy5-CTLH NP showed no significant differences in the distribution profiles in placentas and fetuses. Together, these results demonstrated that thrombus-targeting capability of TLH

NP can be significantly enhanced by decoration with CREKA, while CREKA coating did not increase penetration of CTLH NP across the placental barrier.

In vivo efficacies of CTLH NP were then assessed in pregnant rats with stenosis-induced DVT. Compared with TLH NP, CTLH NP more effectively eliminated iliac vein thrombi and more significantly reduced the thrombus weight and length (Fig. 8a–c). Histological analysis of H&E-stained iliac vein sections and ultrasound imaging further confirmed much better thrombolytic effects for the CTLH NP group (Fig. 8d, e). Mechanistically, CTLH NP notably promoted endothelial repair and more prominently decreased representative pro-thrombotic factors (TAFI and PAI-1), pro-inflammatory cytokines (TNF-α, IL-6, NF-κB), vWF, and oxidative mediators (ROS and MPO; Supplementary Fig. 28 and Fig. 8f–p). Correspondingly, treatment with CTLH NP significantly reduced neutrophils in both peripheral blood and iliac veins of pregnant rats (Supplementary Fig. 29). In all cases, CTLH NP showed more significant beneficial effects than those of TLH NP. In line with these results, treatment with CTLH NP afforded more prominent effects on reversing weight loss of fetuses and placentas, restoring the integrity of the placental structure, and promoting bone growth (Fig. 9). Consequently, the active targeting nanotherapy CTLH NP can more effectively eliminate the formed thrombus in the iliac vein and prevent the subsequent development of DVT in pregnant rats by the enhanced thrombus accumulation, thereby more efficiently preventing the DVT-mediated developmental disorders.

### In vivo targeting and efficacies of TLH NP and CTLH NP in pregnant rats with FeCl₃-induced DVT

Subsequently, in vivo targeting and therapeutic effects of TLH NP and CTLH NP were also examined in another FeCl₃-induced DVT model in pregnant rats (Supplementary Fig. 30a). After i.v. injection in pregnant rats at G10, ex vivo imaging revealed a considerable accumulation of both Cy5-TLH NP and Cy5-CTLH NP at the thrombus site of the left iliac vein (Supplementary Fig. 30b, c). Compared with Cy5-TLH NP, stronger fluorescent signals were detected for Cy5-CTLH NP at iliac vein thrombus sites. For CTLH NP, its effective accumulation in the FeCl₃-induced thrombus was directly observed by SEM and in situ two-photon microscopic imaging (Supplementary Fig. 30d, e and Movie S2). Of note, fluorescence signals of Cy3-CTLH NP and FITC-labeled fibrin were notably co-localized, in a time-dependent manner. Immunofluorescence analysis revealed co-localization of both nanotherapies with endothelial cells, neutrophils, and platelets in the iliac vein thrombus (Supplementary Fig. 30f–h). In addition, both TLH NP and CTLH NP displayed low penetration across the placental barrier (Supplementary Fig. 30i, j). These results collectively demonstrated that the engineered nanotherapies can accumulate in the iliac vein thrombus induced by FeCl₃ in pregnant rats. In this FeCl₃-induced DVT model, the active targeting nanotherapy also displayed significantly high targeting efficiency.

After i.v. treatment of pregnant rats bearing FeCl₃-induced DVT with TLH NP and CTLH NP (Supplementary Fig. 31a), both nanotherapies showed desirable anti-DVT effects, with respect to significantly reducing thrombosis, promoting thrombolysis, improving endothelial repair, attenuating inflammation, lowering oxidative stress, and decreasing pro-thrombotic factors in iliac vein tissues (Supplementary Figs. 31–33). Consistently, both TLH NP and CTLH NP significantly increased the weight of fetuses and placentas, restored normal structures of placentas (in the decidua, junctional, and labyrinthine zones), promoted bone development of fetuses, and therefore effectively rescued IUGR resulting from FeCl₃-induced DVT in pregnant rats (Supplementary Fig. 34). Notably, CTLH NP showed more effective therapeutic outcomes than TLH NP, which is consistent with the potentiated targeting efficiency after CREKA decoration.

### Safety studies

Finally, safety profiles of the engineered LMWH nanotherapies were tested. In vitro evaluation in HUVECs indicated that LH NP, TLH NP, and CTLH NP displayed low cytotoxicity, even at the examined highest dose of approximately 1 mg/mL (Supplementary Fig. 35a). Also, in vitro tests suggested that three nanotherapies displayed negligible hemolysis risk at different concentrations examined (Supplementary Fig. 35b–d).

Further in vivo tests were performed in pregnant rats. A high dose (150 mg/kg, 10-fold higher than that used in therapeutic studies) of LH NP, TLH NP, or CTLH NP was separately administrated to pregnant rats via i.v. injection once daily from days 10 to 20 of gestation. During 10 days of inspection, all pregnant rats displayed normal behaviors and comparable body weight gain (Supplementary Fig. 36a). After treatment, no significant differences in the organ index of typical major organs, fetuses, and placentas were found between the control and nanotherapy groups (Supplementary Fig. 36b–d). Also, no placental abnormality and fetal malformation were observed (Supplementary Fig. 36e–g). The numbers of implantations, loss rates, fetal deaths, and litter sizes were similar for different groups. The development degree of fetal skeleton was also comparable among control and nanotherapy groups (Supplementary Fig. 36h). Likewise, no injuries and infiltration of inflammatory cells were observed in H&E-stained sections of major organs isolated from maternal rats and fetuses (Supplementary Fig. 37). Furthermore, there were no significant changes in representative coagulation parameters including APTT, PT, TT, fibrinogen, and D-dimer for nanotherapy-treated pregnant rats (Supplementary

Fig. 38). Also, no abnormal variations in hematological parameters and typical biomarkers associated with hepatic and kidney functions were detected for different groups (Supplementary Table 1). These preliminary studies suggested that different LMWH nanotherapies are safe for i.v. administration, with no obvious side effects on both maternal rats and fetuses.

## Discussion

During pregnancy, there is a switch in the global hemostatic balance towards a hypercoagulable state, a mechanism that protects the mother against excessive bleeding during birth[7,8]. This phenomenon is associated with several physiological and anatomic changes that can increase the risk of DVT. LMWH is a clinically recommended anticoagulant to treat DVT during pregnancy[21]. However, its broad applications are severely restricted because the currently adopted s.c. administration may lead to skin allergy and bruises[9]. Meanwhile, the half-life of LMWH is short (~3–4 h)[66], and therefore multiple daily administrations for long periods of time are necessary, thereby resulting in poor patient compliance.

Herein we first designed and developed an anti-oxidative, anti-inflammatory, anticoagulant, and endothelial protective/repairing amphiphile, by simultaneously conjugating two bioactive and biocompatible moieties (i.e., Tempol and LA) onto LMWH. Thus obtained material TLH can afforded a multifunctional nanotherapy TLH NP by facile self-assembly. Therapeutically, TLH NP could effectively target and dissolve thrombi, recanalize occluded vessels, and eradicate the recurrence of thromboembolism in pregnant rats with induced DVT. Correspondingly, treatment with TLH NP notably reversed DVT-mediated intrauterine growth restriction and delayed development of fetuses in pregnant rats, which were largely caused by ischemia and hypoxia[14]. Mechanistically, the multiple therapeutic effects of TLH NP were achieved by inhibiting platelet aggregation, facilitating thrombolysis, reducing local inflammatory and oxidative responses relevant to thrombosis, and promoting endothelial repair. By decorating with a fibrin-binding peptide CREKA, targeting efficiency and therapeutic benefits of TLH NP were considerably improved in two DVT models in pregnant rats. Importantly, all the examined LMWH nanotherapies showed very limited accumulation in the fetus, mainly resulting from their low penetration through the placental barrier. According to previous findings, NPs with small sizes (diameter < 80 nm) and cationic surface displayed relatively high transport across the placental barrier, particularly when the placenta is not fully developed early in pregnancy[31,67]. Our LMWH nanotherapies (>100 nm) are negatively charged and were used in a late stage of pregnancy (G > 11), thereby resulting in low transplacental penetration. Consistently, preliminary experiments suggested that three nanotherapies are safe in pregnant rats at the dose 10 times higher than the therapeutic dosage.

Whereas different nanotherapies were developed for targeted treatment of DVT in previous studies[25–28], none of them have been examined in pregnant animal models. Therefore, their effectiveness and particularly safety remain to be validated for DVT therapy during pregnancy. Furthermore, most previously developed nanotherapies were formulated by loading anti-coagulants and/or thrombolytic agents into different NPs, which cannot alleviate other pathological changes closely related to thrombosis, such as endothelial injury and systemic/local inflammation and oxidative stress, thereby resulting in recurrent symptoms after temporary remission of thrombosis and the occurrence and development of other cardiovascular diseases[68]. Nevertheless, for clinical translation of our nanotherapies, the treatment regimens need to be further improved, such as i.v. injection within prolonged time intervals or delivery via s.c. injection. In view of the maintained anticoagulant activity of LMWH nanotherapies at 48 h after i.v. injection, we reasonably consider that i.v. injection of these nanotherapies once every two days or even longer should be effective. Also, therapeutic effects and safety profiles of our nanotherapies

should be confirmed in large animal models. Similar to other antithrombotic agents, clinical applications of our nanotherapies might be limited in hemostatically necessary cases to avoid uncontrolled hemorrhage. Comprehensive follow-up studies are necessary to elucidate all these aspects.

In summary, rational functional integration enabled the development of effective and safe nanotherapies for targeted treatment of DVT in pregnancy. To the best of our knowledge, this is the first study on thrombolytic nanotherapies in pregnant animal models. The promising results encourage further clinical translation studies of the examined nanotherapies for DVT or other cardiovascular diseases (such as atherosclerosis and abdominal aortic aneurysm) during pregnancy. Moreover, the underlying design principles can be generalized to engineer targeting therapies for other pregnancy complications, such as preeclampsia and premature delivery.

# Methods

## Materials
Linoleic acid (LA) was purchased from Aladdin (Shanghai, China). Low molecular weight heparin (LMWH), i.e., Enoxaparin, was obtained from Dalian Meilun Biological Technology Co., Ltd (Dalian, China). 4-Hydroxy-2,2,6,6-tetramethylpiperidine 1-oxyl (Tempol), $N$-(3-dimethylaminopropyl)-$N$-ethylcarbodiimide (EDC), $N$-hydroxysuccinimide (NHS), 4-diaminomethylpyridine (DMAP), anhydrous dimethylformamide (DMF), anhydrous dimethyl sulfoxide (DMSO), 2,2-diphenyl-2-picrylhydrazyl (DPPH), Alizarin red, and Alcian blue were purchased from Sigma-Aldrich (U.S.A.). The peptide CREKA (Cys-Arg-Glu-Lys-Ala) with purity of 99%, cyanine3 (Cy3) NHS ester, cyanine5 (Cy5) NHS ester, and FITC-conjugated human fibrinogen were provided by Xi'an Ruixi Biological Technology Co., Ltd (China). The cell line of HUVECs was obtained from ATCC (#CRL-1730). Penicillin, streptomycin, fetal bovine serum (FBS), and RPMI 1640 medium were purchased from Gibco (U.S.A.). 4′6-Diamidino-2-phenylindole (DAPI) and dihydroethidium (DHE) were purchased from Beyotime Biotechnology (Nantong, China). LysoTracker Green was purchased from Invitrogen (U.S.A.). Polyclonal antibodies to interleukin (IL)−6 (1:1000, #DF6087) and tumor necrosis factor (TNF)-α (1:1000, #AF7014) were purchased from Affinity Biosciences (U.S.A.). Polyclonal antibody to plasminogen activator inhibitor type 1 (PAI-1, 1:1000, #bs-6562R) was obtained from Bioss (Beijing, China). Nuclear factor κB (NF-κB, 1:1000, #ER0815) polyclonal antibody was provided by HuaBio Technology (Hangzhou, China). Monoclonal antibody to β-actin (1:1000, #66009, Clone-No.2D4H5), FITC-conjugated secondary antibody (1:100, #SA00003), HRP-conjugated AffiniPure goat anti-rabbit IgG secondary antibody (1:5000, #SA00001-2), and HRP-conjugated AffiniPure goat anti-mouse IgG secondary antibody (1:5000, #SA00001-1) were obtained from Proteintech Group, Inc (U.S.A.). Rat myeloperoxidase (MPO, 1:500, #GB11224) antibody and anti-von Willebrand factor (vWF, 1:200, #GB11020) antibody were purchased from Wuhan Servicebio Technology Co., Ltd (Wuhan, China). FITC mouse anti-rat CD61 (1:100, #561909), PE mouse anti-rat RP-1 (1:100, #550002), FITC mouse anti-rat CD11b (1:100, #561684), and PE mouse anti-rat CD45 (1:100, #554884) were purchased from BD Pharmingen (U.S.A.). PerCP/Cy5.5 mouse anti-rat CD68 antibody (1:100, #SC-20060) was purchased from Santa Cruz Biotechnology, Inc. (U.S.A.). Anti-thrombin III antibody, factor Xa, and S-2765 substrate were obtained from Aglyco International Technology Co., Ltd (UK). Rat thrombin-antithrombin complex (TAT) ELISA kit was purchased from Elabscience (U.S.A.).

## Synthesis of different LMWH derivatives
To synthesize linoleic acid (LA)-conjugated LMWH (defined as LH), LA (196 mg) was dissolved in DMF (10 mL), into which NHS (78 mg), EDC (150 mg), and DMAP (25 mg) were added. The obtained mixture was magnetically stirred under a nitrogen atmosphere at room temperature for 3 h. Then, LMWH (200 mg) dissolved in formamide (15 mL) at

50 °C for 0.5 h was added into the above solution. After the reaction mixture was stirred under a nitrogen atmosphere at room temperature for additional 24 h. The obtained solution was dialyzed against deionized water at 25 °C. The outer aqueous solution was exchanged every 3 h. After 72 h of dialysis, samples were collected by lyophilization.

LMWH simultaneously conjugated with Tempol and LA (defined as TLH) was synthesized by sequentially conjugating Tempol and LA onto LMWH. Briefly, Tempol-conjugated LMWH (Tpl-LMWH) was first synthesized by coupling reaction between 90 mg Tempol and 160 mg LMWH, following the similar procedures as mentioned above. Thus obtained Tpl-LMWH (200 mg) was dissolved in formamide, which was further reacted with LA (196 mg). The final product was purified by dialysis in deionized water and collected after lyophilization to give rise to TLH.

Also, Cy5 or Cy3-labeled LH and TLH were synthesized by reaction of 1 mg Cy5 or Cy3 NHS ester with 50 mg LH or TLH, respectively. The final product was collected after dialysis against deionized water and lyophilization. To synthesize CREKA-conjugated TLH (abbreviated as CTLH), 5 mg CREKA and 50 mg TLH were reacted in 1 mL of DMSO, followed by dialysis and lyophilization to collect the resulting product CTLH. To prepare Cy5 or Cy3-labeled CTLH, 50 mg Cy5 or Cy3-labeled TLH was reacted with 5 mg CREKA in 1 mL of DMSO, and the final product was collected after dialysis against deionized water and lyophilization.

## Materials characterization
[1]H NMR spectra were recorded on a spectrometer operating at 600 MHz (DD2, Agilent). Fourier transform infrared (FTIR) spectra were recorded on a PerkineElmer FT-IR spectrometer (100 S). Electron paramagnetic resonance (EPR) spectroscopy was carried out on a JES-FA200 EPR spectrometer (JEOL Ltd., Japan) operating at 9.4 GHz with a 100 kHz magnetic field modulation.

## Preparation and characterization of various nanoparticles
Dialysis was conducted to prepare NPs self-assembled from LH, TLH, or CTLH. Briefly, 100 mg LH, TLH, or CTLH was dissolved in 2 mL of DMSO, the obtained solution was dialyzed against deionized water at 25 °C. The outer aqueous solution was exchanged every 2 h. After 24 h of dialysis, NPs were collected by lyophilization and stored at 2–8 °C for further experiments. Different fluorescent NPs were prepared via the similar dialysis procedures.

Size, size distribution, and ζ-potential of various NPs in aqueous solutions were measured using a Malvern Zetasizer NanoZS instrument (Malvern). All measurements were performed at room temperature. The morphology of NPs was observed by transmission electron microscopy (TEM). Samples were prepared by dipping a Formvar-coated Cu grid into aqueous solutions of different NPs (~2 mg/mL). After water was evaporated at room temperature, samples were observed directly without any staining by a JEM-1400 microscope (JEOL Ltd., Japan). Scanning electron microscopy (SEM) was also employed to observe the morphology of NPs by a FIB-SEM microscope (Crossbeam 340, Zeiss). To evaluate in vitro stability of different nanoparticles, approximately 5 mg LH NP, TLH NP, or CTLH NP was dissolved in 4 mL of PBS (pH 7.4) or serum from rats. After incubation at 37 °C for predetermined time periods (0, 1, 6,12, 24, 48, and 72 h), mean size and ζ-potential values of different samples were measured.

## Determination of the critical aggregation concentration of different LMWH derivatives
A previously reported fluorescence spectrometry method was employed to determine the critical aggregation concentration (CAC) of different LMWH derivatives[41]. Aqueous solutions of various concentrations of NPs containing $6.0 \times 10^{-7}$ mol/L pyrene were prepared and incubated at 50 °C for 10 h. After cooling at room temperature for 10 h, the samples were measured using a F-7000 fluorescence

spectrophotometer (Hitachi, Japan) with a slit width of 2.5 nm at 25 °C. The emission wavelength was set at 390 nm to acquire excitation spectra.

### Thrombin-antithrombin complex measurement

The TAT value was determined using the ELISA kit. Briefly, thrombin and antithrombin were incubated with LMWH, LH NP, or TLH NP at 4 mg/mL of LMWH for 30 min. The test samples (50 μL) and the standard sample of TAT complex (50 μL) were separately added to a microplate. Then 50 μL of TAT antibody was added to each well and incubated at 37 °C for 90 min, followed by addition of different reaction solutions. The absorbance at 450 nm was measured to calculate the TAT value.

### Anti-factor Xa activity assay

The anti-factor Xa activity of LMWH and different NPs was determined by a chromogenic substrate assay. Briefly, 100 μL of aqueous solution of LMWH or NPs was mixed with 100 μL of antithrombin III (AT III, 1 U/mL) and incubated at 37 °C for 3 min. Then 200 μL of factor Xa (FXa) solution (0.4 IU/mL) was added and incubated for 2 min. Subsequently, 200 μL of S-2765 (1 mM) was added, followed by addition of 50% acetate (200 μL) to stop the reaction. The anti-FXa activity was determined by measuring absorbance at 405 nm.

### Determination of reactive oxygen species-scavenging capability

To determine the superoxide anion-eliminating capability of LH NP, TLH NP, and CTLH NP, different concentrations of NPs varying from 0.05, 0.1, 0.15, 0.25, 0.5, to 1 mg/mL were incubated with an excess amount of superoxide anion produced by the xanthine oxidase anion system. After incubation at 37 °C for 40 min, the remaining superoxide anion was measured by the Superoxide Anion Free Radical Detection Kit (Nanjing Jiancheng Bioengineering Institute, China). Then the superoxide anion-eliminating capacity was calculated.

The free radical-scavenging capability was measured using previously established protocols[42]. Briefly, 1.5 mL of fresh solution of DPPH• (100 μg/mL) was incubated with 3 mL of methanol containing different concentrations of LH NP, TLH NP, or CTLH NP (varying from 0.05, 0.1, 0.15, 0.25, 0.5, 1, to 2 mg/mL) in dark for 30 min. The absorbance at 517 nm was recorded by UV-visible spectroscopy to calculate the DPPH• elimination capacity.

The hypochlorite-eliminating capability of LH NP, TLH NP, and CTLH NP was quantified using a luminescence method developed in our previous study[69]. To evaluate the $H_2O_2$-scavenging capacity, various concentrations of LH NP, TLH NP, or CTLH NP (from 0, 1, 2, 4, to 6 mg/mL) were incubated with 2 mL of 0.01 M PBS containing 50 mM $H_2O_2$ for 24 h. Using a commercially available kit, residual $H_2O_2$ was determined by measuring the absorbance at 405 nm, and eliminated $H_2O_2$ was calculated.

### In vitro cytotoxicity of different nanoparticles.

Cytotoxicity of different NPs was evaluated in human umbilical vein endothelial cells (HUVECs) by CCK-8 assay. Briefly, HUVECs were seeded in 96-well plates ($1 \times 10^4$ cells/well) in 100 μL of 1640 medium containing 10% FBS, 100 U/mL penicillin, and 100 mg/mL streptomycin. After overnight incubation at 37 °C in a 5% $CO_2$ atmosphere, the culture medium was removed. Then, various doses of LH NP, TLH NP, or CTLH NP were added and incubated with cells for additional 24 h. Subsequently, the culture medium was removed and replaced with 200 μL of fresh 1640 medium, and 20 μL of CCK-8 solution was added to each well. After incubation for 1 h, the absorbance at 450 nm was measured by a Thermo Scientific Varioskan Flash multimode microplate reader. The cell viability (%) was calculated.

### Hemolysis assay.

Fresh blood samples from pregnant rats were collected in an EDTA-containing anticoagulant tube, which was centrifuged at 1000 rpm for 10 min and subsequently rinsed with PBS three times to collect erythrocytes. Then LH NP, TLH NP, or CTLH NP was separately incubated with erythrocytes at room temperature for 24 h, at the final concentrations of 0.08, 0.15, 0.3, 0.45, and 0.6 mg/mL. Erythrocytes in PBS and water served as the negative and positive control, respectively. Finally, the supernatant was obtained by centrifugation at 3000 rpm for 5 min, and absorbance at 541 nm was measured by a microplate reader. The hemolysis degree (HD) was calculated according to the following formula:

$$HD(\%) = [(OD_{test} - OD_{neg})/(OD_{pos} - OD_{neg})] \times 100\% \qquad (1)$$

Where $OD_{test}$ is the absorbance value of different nanoparticle groups, while $OD_{pos}$ and $OD_{neg}$ are the positive (water) and negative (PBS) control groups, respectively.

### Cellular uptake of nanoparticles.

HUVECs were cultured in 12-well plates ($1 \times 10^5$ cells/well) in 1 mL of culture medium overnight and then incubated with different Cy5-labeled NPs (Cy5-LH NP or Cy5-TLH NP). After incubation for different periods of time (0, 0.5, 1, 2, 4, and 8 h), HUVECs were washed by PBS and stained with LysoTracker Green. Cells were further washed with PBS, fixed with 4% paraformaldehyde, and stained with DAPI. Subsequently, fluorescence images were acquired by confocal laser scanning microscopy (CLSM) (Olympus FV1200, Japan).

In addition, flow cytometric analysis was conducted to quantify cellular uptake of different NPs. To this end, HUVECs were cultured in 6-well plates for 12 h. Then cells were treated with fresh medium containing Cy5-labeled NPs (100 μg/mL). At predefined time points, the cells were harvested for flow cytometric analysis (CytoFLEX, Beckman Coulter, U.S.A.). Following similar procedures, dose-dependent internalization of different Cy5-labeled NPs in HUVECs was examined after 2 h of incubation. The tested doses varied from 0, 5, 10, 20, 50, 100, to 200 μg/mL.

### Quantification of intracellular ROS generation in HUVECs

HUVECs were seeded in a 12-well plate ($2 \times 10^5$ per well) and incubated overnight. After the cells were treated with fresh medium containing NPs (LH NP or TLH NP) at 15 μg/mL of LMWH, Tempol (10 μg/mL), or free LMWH (15 μg/mL) for 24 h, the culture medium was replaced with fresh medium containing 150 μM $H_2O_2$ and incubated for 12 h. Then cells were washed with PBS and incubated with 2.5 μM DHE for 30 min. Fluorescence intensities were observed by CLSM and quantified via flow cytometry. Following similar procedures, the effects of LA (8 μg/mL), LH NP (at 15 μg/mL of LMWH), TLH bolus (i.e., a mixture of 10 μg/mL Tempol, 8 μg/mL LA, and 15 μg/mL LMWH), and TLH NP (at 15 μg/mL of LMWH) on intracellular ROS generation were compared in HUVECs.

### In vitro apoptosis assay of HUVECs

HUVECs were seeded in a six-well plate. After 24 h of incubation with various NPs (LH NP or TLH NP) at 15 μg/mL of LMWH, Tempol (10 μg/mL), or free LMWH (15 μg/mL), cells were treated with 150 μM $H_2O_2$ for 12 h. Then, cells were digested with 0.25% trypsin, and collected by centrifugation. Subsequently, apoptosis analysis was performed by flow cytometry after Annexin V and PI staining. Similarly, anti-apoptotic effects of LA (8 μg/mL), LH NP (at 15 μg/mL of LMWH), TLH bolus (i.e., a mixture of 10 μg/mL Tempol, 8 μg/mL LA, and 15 μg/mL LMWH), and TLH NP (at 15 μg/mL of LMWH) were compared.

### Effects of different nanoparticles on migration and wound healing in HUVECs

HUVECs were seeded in the upper compartment of a Transwell chamber at a density of $4 \times 10^4$ cells per well, and 100 μL of medium containing Tempol (10 μg/mL), LMWH (10 μg/mL), LH NP (at 10 μg/

mL of LMWH), or TLH NP (at 10 μg/mL of LMWH) was separately added. Fresh 1640 medium was added in the bottom compartment. After 12 h of incubation, cells invaded through the membrane were fixed, stained using crystal violet, and counted by optical microscopy.

To perform scratch wound healing assay, HUVECs were seeded in 6-well plates and cultured until confluence. Then an incision was scratched using a 200-μL micropipette tip. Fresh culture medium containing Tempol (10 μg/mL), LMWH (10 μg/mL), LH NP (at 10 μg/mL of LMWH), or TLH NP (at 10 μg/mL of LMWH) was then added and incubated for 24 h, with fresh 1640 medium serving as a control. Images were taken with an optical microscope. Migrated cells were analyzed by ImageJ (V1.8.0.112), and the inhibition percentage was calculated. In a separate study, the effects of LA (5 μg/mL), LH NP 10 μg/mL of LMWH, TLH bolus (i.e., a mixture of 10 μg/mL Tempol, 5 μg/mL LA, and 10 μg/mL LMWH), and TLH NP (at 10 μg/mL of LMWH) were evaluated.

## Animals
Female Sprague-Dawley rats (11-12 weeks, 240–250 g) and male Sprague-Dawley rats (12–13 weeks, 250–260 g) were obtained from the Laboratory Animal Center of Chongqing Medical University. Rats were housed under standard conditions (a 12-h light-dark cycle, 22 °C temperature, and 40% relative humidity) with ad libitum access to food and water. After one week of habitation, female rats were mated overnight during the pro-estrus period and timed pregnant rats (vaginal smear positive, gestational day 0 (G0), gestational term (G22)), which were used for the following experiments. All animals were monitored once a day during the whole experiments to confirm their health conditions. Animals were anesthetized by continuous inhalation of isoflurane-oxygen mixture (4%). After the related experiments were finished, animals were sacrificed by intraperitoneal injection of 3% sodium pentobarbital. All animal experiments were performed in accordance with ARRIVE guidelines and the Guide for the Care and Use of Laboratory Animals proposed by the Ministry of Health of the People's Republic of China. All procedures and protocols were approved by the Animal Ethics Committee at Chongqing Medical University (Chongqing, China; No. 2019-136).

## In vitro anticoagulation of various nanoparticles
First, 100 μL of PBS containing Tempol (8 mg/mL), LMWH (4 mg/mL), LH NP (at 4 mg/mL of LMWH), TLH NP (at 4 mg/mL of LMWH), or CTLH NP (at 4 mg/mL of LMWH) was added in 24-well dishes and pre-warmed at 37 °C for 5 min. PBS alone served as a control. Then, 300 μL of citrated whole blood was slowly added, followed by addition of 50 μL of 0.2 M CaCl$_2$ solution. The mixture was then incubated at 37 °C for 5 min. Next, 15 mL of deionized water was added to the wells and incubated at 37 °C with shaking at 30 rpm for 10 min. Erythrocytes that were not entrapped in the clots were hemolyzed with deionized water, and the absorbance of hemoglobin-containing solution was measured at 541 nm. The absorbance of citrated whole blood in deionized water was used as the blank. Finally, the blood-clotting index (BCI) was calculated according to the following equation:

$$BCI(\%) = (Absorbance\ of\ sample / Absorbance\ of\ blank) \times 100\% \quad (2)$$

In a separate study, freshly prepared rat plasma was incubated with LMWH, LH NP, or TLH NP at 4 mg/mL of LMWH at 37 °C for 30 min. Then neoplastin, CaCl$_2$, or thrombin was added to the mixture to determine activated partial thromboplastin time (APTT), prothrombin time (PT), and thrombin time (TT) by using an automatic blood analyzer (Mindray, BC-2800Vet). Following similar procedures, anticoagulation assay was conducted for LA (1.6 mg/mL), LH NP 4 mg/mL of LMWH, TLH bolus (i.e., a mixture of 8 mg/mL Tempol, 1.6 mg/

LA, and 4 mg/mL LMWH), and TLH NP (at 4 mg/mL of LMWH) were evaluated.

## In vitro inhibition of platelet aggregation by different nanoparticles
After normal pregnant rats were anesthetized, blood was collected and stored in EDTA tubes. In addition, platelet-rich plasma (PRP) was prepared by centrifugation of blood at 850×g for 10 min. The PRP supernatant was harvested, and centrifuged at 800×g for 10 min to discard erythrocytes. To determine the platelet aggregation, 100 μL of resuspended solution containing platelets (at $5 \times 10^8$ platelets/mL) was added to each well of 96-well plates, into which Tempol (400 μg/mL), LMWH (400 μg/mL), LH NP (at 400 μg/mL of LMWH), TLH NP (at 400 μg/mL of LMWH), and CTLH NP (at 400 μg/mL of LMWH) were separately added. After 30 min of incubation, thrombin at a final concentration of 1 U/mL was added. In the control group, the same volume of PBS was added. At 10 min after this procedure, samples were fixed for 1 h by adding an equal volume of 2% glutaraldehyde and dehydrated with an ethanol/acetone gradient to prepare specimens for SEM observation. Subsequently, specimens were mounted on freshly cleaved mica and coated with gold palladium. SEM images were taken on a FIB-SEM microscope (Crossbeam 340, Zeiss). Similarly, the effects of LA (200 μg/mL), LH NP 400 μg/mL of LMWH, TLH bolus (i.e., a mixture of 400 μg/mL Tempol, 200 μg/mL LA, and 400 μg/mL LMWH), and TLH NP (at 400 μg/mL of LMWH) on platelet aggregation were evaluated by SEM observation.

To observe in vitro aggregation of platelets by CLSM, freshly prepared platelets were suspended in PBS at $10^8$ platelets/mL, into which FITC-CD61 (0.5 mg/mL, 1:100) was added, followed by incubation at room temperature for 40 min. The samples were centrifuged at 1600g for 5 min. After the buffer was removed, platelets were reconstituted in PBS to a final concentration of $10^8$ platelets/mL. Subsequently, 150 μL of platelet-containing solution was added to each well, and 15 μL of PBS containing Tempol (400 μg/mL), free LMWH (400 μg/mL), LH NP (at 400 μg/mL of LMWH), TLH NP (at 400 μg/mL of LMWH), or CTLH NP (at 400 μg/mL of LMWH) was added and incubated with platelets for 30 min. Then thrombin was added to a final concentration of 1 U/mL. After platelet aggregation, samples were fixed using 2% glutaraldehyde at room temperature for 1 h, and imaged by CLSM.

## In vitro thrombolytic effects of various nanoparticles
Blood was collected from pregnant rats. Blood clots were prepared by adding 50 U thrombin into 100 μL of whole blood. After 30 min of incubation at 37 °C with continuous shaking at 50 rpm, clots were moved from Eppendorf tubes into 24-well dishes containing 1.5 mL of saline in each well. Clots were treated with PBS containing 8 mg/mL Tempol, 4 mg/mL LMWH, LH NP, TLH NP, or CTLH NP. For LMWH-containing NPs, the dose was 4 mg/mL of LMWH. PBS was used as a negative control, while urokinase (3 mg/mL) was used as a positive control. In a separate study, the examined formulations included LA (1.6 mg/mL), LH NP 4 mg/mL of LMWH, TLH bolus (i.e., a mixture of 8 mg/mL Tempol, 1.6 mg/mL LA, and 4 mg/mL LMWH), and TLH NP (at 4 mg/mL of LMWH) were evaluated. The mixture was incubated at 37 °C with continuous shaking at 50 rpm. At 0, 30, 90, 180, and 300 min, 200 μL of the supernatant was placed in a 96-well dish and the absorbance was measured at 415 nm by a microplate reader. The dissolution degree (DD) was calculated according to the following formula:

$$DD = (OD_i - OD_0)/(t_i - t_0) \quad (3)$$

Where $OD_i$ and $OD_0$ indicate the optical density at two consecutive time points $t_i$ and $t_0$, respectively.

### In vitro targeting capability of different nanoparticles to platelets

Freshly prepared platelets from pregnant rats in PBS were incubated with FITC-CD61 (0.5 mg/mL, 1:100) at room temperature for 40 min. Then the samples were centrifuged at 1600×$g$ for 5 min. After the buffer was removed, platelets were reconstituted in PBS to a final concentration of $10^8$ platelets/mL. Then 150 μL of platelet-containing PBS was added to each well of plates, into which thrombin was added to a final concentration of 1 U/mL. Platelets without thrombin pre-treatment served as a control. After platelet aggregation, 15 μL of PBS containing Cy5-LH NP, Cy5-TLH NP, or Cy5-CTLH NP at 200 μg/mL was added. Samples were fixed for 1 h using 2% glutaraldehyde at room temperature, and imaged by CLSM. Also, the fluorescence intensity was quantified.

### Establishment of two experimental models of deep vein thrombosis in pregnant rats

A stenosis-induced DVT model in pregnant rats was established at G10. To this end, Sprague-Dawley rats were anesthetized by isoflurane-oxygen mixture and placed in a supine position. After laparotomy, intestines were exteriorized and sterile saline was applied during the whole procedure to prevent drying. After gentle separation from the iliac artery, the two ends and branches of the left iliac vein were ligated by a 6-0 polypropylene suture immediately over a 30-gauge needle, and then the needle was removed to establish a partial flow restriction (stenosis). After surgery, the peritoneum and skin were closed by sutures 6-0 silk and 3-0 silk, respectively. The thrombus weight was measured at 0, 3, 6, 12, and 24 h after surgery. The iliac vein thrombosis was developed at 6 h post operation, and the blood flow velocity at the thrombosis site was confirmed by Doppler ultrasound.

To establish a rat model of FeCl$_3$-induced iliac vein thrombosis, the left iliac vein of pregnant rats at G10 was exposed, after anesthesia with isoflurane-oxygen mixture. A piece of filter paper saturated with 10% FeCl$_3$ solution was placed over the left iliac vein for 5 min. The treated area was flushed with warm saline. The thrombus formation was confirmed by Doppler ultrasound. Only rats with a confirmed thrombus were used in following experiments, while rats without thrombus were excluded.

### In vivo targeting capability of different nanoparticles

After DVT in pregnant rats at G10 was induced by stenosis or FeCl$_3$, Cy5-labeled NPs (Cy5-LH NP, Cy5-TLH NP, and Cy5-CTLH NP) at 15 mg/kg were separately administered in randomly assigned rats via intravenous (i.v.) injection. At 4 h after injection, rats were euthanized, and the inferior vena cava and iliac vein were harvested together. After rinsing with PBS, ex vivo imaging was carried out using an OI600 MF Touch Multifunction Imager (BIO-OI Biotechnology Co., Ltd, China) and relative mean fluorescence intensity (MFI) was analyzed. Similarly, at day 5 after DVT in pregnant rats was induced, Cy5-labeled NPs were administered via i.v. injection at G15. At 4 h after injection, rats were euthanized. Placentas and fetuses were isolated for ex vivo imaging. In a separate study, after DVT in pregnant rats (at G10) was induced by stenosis or FeCl$_3$, CTLH NP was administered via i.v. injection. The saline group served as a blank control. After 4 h, the iliac vein wall was fixed by 2% glutaraldehyde and dehydrated with an ethanol and acetone gradient to prepare specimens for SEM observation.

To visually analyze the target ability in living rats, pregnant rats (at G10) were anesthetized, and FITC-conjugated human fibrinogen (10 mg/kg rat) was administered via the tail vein. Simultaneously, DVT was induced via stenosis or FeCl$_3$ as described above. After thrombus formation, Cy3-CTLH NP was injected via the tail vein. Sequential images were acquired for 0 to 4 h using a Multiphoton Laser Scanning Microscope (FVMPE-RS, Olympus).

### Histological analysis of the distribution of nanoparticles

A DVT model was established in pregnant rats (at G10) and treated with different Cy5-labeled NPs according to the procedures described above. At 4 h after i.v. injection of different NPs, the left iliac vein was cut down and cryosections of 7-μm thickness were prepared. The slides were blocked and separately incubated with different primary antibodies (CD61 antibody, 1:100; vWF antibody, 1:200; MPO antibody, 1:500) and FITC-labeled secondary antibody (1:100). After nuclei were stained with DAPI, fluorescence images were acquired by CLSM. In addition, placentas (at G15) were isolated and fixed with 4% neutral buffered formalin, and coronal and transverse sections of 4−6 μm thickness in paraffin were prepared. Subsequently, fluorescence images were acquired.

### Tail bleeding in pregnant rats treated with different formulations

Pregnant rats at G10 were anesthetized. Rats were placed on a shelf with the tail immersed in a test tube containing saline for 10 min at 37 °C. Then rats in different groups were separately treated with 5 mg/kg LMWH by subcutaneous (s.c.) injection and i.v. injection at 3 or 5 mg/kg LMWH. Other rats were separately administered with LH NP or TLH NP at 5 mg/kg of LMWH via i.v. injection. At 10 min after injection of different agents, tail bleeding was induced by cutting off 2 cm from the tip of the tail and the tip of the tail was removed with a razor. The tail was placed in a plastic tube containing saline. Bleeding was considered stopped when no blood flowed from the vessel for at least 2 min. The bleeding time and bleeding volume were recorded.

### Therapeutic effects of different nanotherapies in pregnant rats with DVT

To evaluate therapeutic effects of different NPs, DVT in pregnant rats was induced by stenosis or FeCl$_3$. At day 1 after induction, Tempol (8 mg/kg) or various nanotherapies (at 5 mg/kg of LMWH) were i.v. injected in randomly assigned rats via the tail vein once a day for 5 days. Free LMWH at 5 mg/kg was subcutaneously injected. In the normal control group, healthy rats were treated with saline. After 5 days of treatment, rats were euthanized on day 17. Left iliac veins were excised from rats, and the weight and length of vessels with thrombi were measured. To reduce deviations due to differences in the body weight of pregnant rats with varied numbers of litters, the thrombus weight data were normalized using the formula:

$$\text{The normalized thrombus weight} = (\text{Weight of thrombus} / \text{Weight of pregnant rat}) \times \text{Mean weight of pregnant rats}$$

$$(4)$$

Also, the middle segments of left iliac veins were collected, and coronal/transverse paraffin sections (5 μm) were prepared. The sections were separately stained by hematoxylin and eosin (H&E), anti-IL-6, anti-TNF-α, vWF antibody, or PAI-1 antibody. Meanwhile, left iliac veins excised from rats were embedded in OCT compound, and 7-μm vessel cryosections were prepared and incubated with 5 μM DHE for 30 min. After rinsing with PBS, cryosections were imaged by fluorescence microscopy. Similarly, sequential staining with MPO antibody and FITC-conjugated secondary antibody was conducted, while coronal cryosections were stained with DAPI and FITC-CD61, followed by fluorescence microscopy. In addition, the iliac veins with thrombi were fixed with 2% glutaraldehyde and dehydrated with an ethanol/acetone gradient to prepare specimens for SEM observation. After different treatments, thrombi in the left iliac veins in pregnant rats (at G17) were also confirmed by an ultrasound scanner (Vevo 3100 LT, FujiFilm VisualSonics Inc.). Also, a part of the left iliac vein was homogenized and lysed. The contents of ROS and MPO were quantified by commercially available kits. The mRNA levels of TNF-α (forward, TTCATCCGTTCTCTACCCA; reverse, GAGCCACAATTCCCTTTCT), IL-6

(forward, AGGAACGAAAGTCAACTCCA; reverse, TTGTGAAGTAGGG AAGGCA), NF-κB (forward, AAAAACGCATCCCAAGGTGC; reverse, AAGCTCAAGCCACCATACCC), PAI-1 (forward, CCTCCTCATCCTGCCT AA; reverse, CTGCTCTTGGTCGGAAAG), TAFI (forward, TGCCAGTGA TGAATGTGG; reverse, CCAGTGTTTGGAAGCGA), vWF (forward, TTG CTCTGCCCTTGGTATGG; reverse, TCCGAAAGGTCCCAGAGCTA), and tPA (forward, AGAGCCTGCAGGAACTCAAG; reverse, CTCCCATGT ATTCCCTGGTC) were quantified by RT-qPCR (Bio-Rad CFX). Western blot analyses were conducted to determine the levels of NF-κB and IL-6. Following similar procedures, therapeutic effects of LA (2 mg/kg), LMWH (3 mg/kg), LH NP (at 3 mg/kg of LMWH), TLH bolus (a mixture of 8 mg/kg Tempol, 2 mg/kg LA, 3 mg/kg LMWH), and TLH NP (at 3 mg/ kg of LMWH) were compared after i.v. injection each day for 5 days.

### Flow cytometry analysis of neutrophils in peripheral blood and left iliac veins

DVT in pregnant rats at G10 was induced via stenosis or $FeCl_3$ as aforementioned. After treatment with different formulations, the collected blood samples were lysed by adding a lysing solution, mixing, and incubating on ice for 5 min. Then samples were centrifuged at 500×$g$ for 5 min. The thoroughly lysed samples were mixed with 100 µL of PBS, and incubated with 1 µL of primary antibody solution (PE mouse anti-rat RP-1 and FITC mouse anti-rat CD11b at 100 µg/mL) in the dark at 4 °C for 1 h. After centrifugation at 500 $g$ for 5 min, the supernatant was discarded, and pellets were resuspended in 0.5 mL of serum-free PBS for analysis by flow cytometry (CytoFLEX, Beckman Coulter, USA). A minimum of 5000 cells within the gated region were analyzed. In addition, the fragments of left iliac veins with thrombi in various groups were collected and the arteries were removed. Tissue samples were treated with 1 mL of the mixed enzyme digestion solution (collagenase I, 450 U/mL; collagenase XI, 125 U/mL; DNase I, 60 U/mL; hyaluronidase, 60 U/mL) and cut into small segments. The suspensions were then transferred to centrifuge tubes, into which an equal volume of PBS containing 2% FBS was added to stop the digestion. The suspensions were filtered through a nylon mesh with 70-µm pores and centrifuged at 500×$g$ for 5 min. The obtained pellets were treated with PE-RP-1 and FITC-CD11b antibodies, followed by analysis via flow cytometry.

### Flow cytometry analysis of leukocytes in left iliac veins

Pregnant rats with stenosis-induce DVT received daily i.v. injection of LA (2 mg/kg), LMWH (3 mg/kg), TLH bolus (a mixture of 8 mg/kg Tempol, 2 mg/kg LA, and 3 mg/kg LMWH), LH NP (at 3 mg/kg LMWH), or TLH NP (at 3 mg/kg LMWH) for 5 days. Cell suspensions from left iliac veins and thrombi were prepared, stained with PE mouse anti-rat CD45 (100 µg/mL) for 1 h, and fixed, followed by resuspension with 50 µL of PBS mixed with 100 µL permeabilization medium containing PerCP/Cy5.5 mouse anti rat CD68 (100 µg/mL) for 30 min. Pellets were obtained after centrifuged at 500×$g$ for 5 min and resuspended for flow cytometric analyses.

### In vivo anti-FXa activity of different NPs

Normal pregnant rats were divided into three groups, which were separately treated with free LMWH at 5 mg/kg (s.c. injection) or two types of NPs (LH NP and TLH NP) at 5 mg/kg of LMWH via i.v. injection. At different time points after treatment, the anti-FXa activity of plasma samples was determined. Also, the area under the anti-FXa activity-time curve (AUC) was calculated.

### Inhibition of fetal developmental disorders in pregnant rats with DVT by different nanotherapies

DVT in pregnant rats was induced via stenosis or $FeCl_3$. After rats were treated with saline, Tempol (8 mg/kg), LMWH (5 mg/kg, s.c. injection), LH NP (at 5 mg/kg of LMWH), TLH NP (at 5 mg/kg of LMWH), or CTLH NP (at 5 mg/kg of LMWH), they were sacrificed at G17. The litter size in

each group was calculated. All live fetuses and placentas were collected, photographed, and weighed. Placentas and half of the live fetuses from each litter were fixed in 4% formalin solution for histological analysis via H&E staining. The other half fetuses were preserved in 95% ethanol solution, eviscerated, and then processed for skeletal staining with Alcian blue and Alizarin red S for subsequent skeletal examination.

### Evaluation of reproductive toxicity of different nanoparticles

Pregnant rats at G10 received i.v. injection of different NPs (LH NP, TLH NP, and CTLH NP) at 150 mg/kg per day for 10 days. The control group was treated with the same volume of saline. All rats were monitored daily throughout the gestation period for mortality, morbidity, general appearance, and behaviors. Also, maternal body weights were measured daily. Cesarean sections were performed on G20. The uterus of each rat was removed and examined for live and dead fetuses, resorptions, and total implantations. All live fetuses and placentas were photographed and weighed individually. The crown-rump length of all live fetuses was measured. In addition, half of the live fetuses and placentas from each litter were fixed in 4% formalin solution for histological analysis after H&E staining. The other half fetuses were used for skeletal staining with Alcian blue and Alizarin red S as aforementioned. Upon cesarean section, all pregnant females were euthanized for collection of blood samples. Total protein (TP), albumin (ALB), total bilirubin (T-BIL), creatinine (CRE), blood urea nitrogen (BUN), alkaline phosphatase (ALP), aspartate aminotransferase (AST), alanine aminotransferase (ALT), total cholesterol (TC), triglyceride (TG), white blood cells (WBC), red blood cells (RBC), PT, TT, APTT, fibrinogen (FIB), D-dimer, and platelets were measured. Also, a complete gross postmortem examination was performed. The absolute and relative (organ-to-body weight ratios) weight of heart, liver, spleen, lung, and kidneys were measured. The major organs of females and fetuses were fixed in 4% formalin solution for histological analysis.

### Evaluation of postnatal viability and growth of offspring

Pregnant rats with stenosis-induced DVT received daily i.v. injection of LA (2 mg/kg), LMWH (3 mg/kg), TLH bolus (a mixture of 8 mg/kg Tempol, 2 mg/kg LA, and 3 mg/kg LMWH), LH NP (at 3 mg/kg LMWH), or TLH NP (at 3 mg/kg LMWH) for 5 days. The weight of offspring rats in each group at days 0, 7, and 21 after birth was measured. At day 21 after birth, morphological abnormalities of the offspring were evaluated, and their lung sections were prepared for histological analysis after H&E staining.

### Statistics and reproducibility

The SEM and TEM images of NPs in Figs. 2c–f and 6b, c as well as the TEM and SEM images of platelets (Figs. 2x and 6g) were repeated six times independently with similar results, and one representative image from each group was shown. Confocal images (such as Supplementary Figs. 4b, e, 7, and 9a), digital photos (such as Supplementary Fig. 4a, c), microscopic images (such as Supplementary Fig. 9e, g), and flow cytometric profiles (such as Supplementary Fig. 9c) were repeated six times with similar results, and a series of representative images from each group were shown. In animal studies, for targeting/biodistribution profiles of NPs in veins with thrombi (such as Fig. 3b, d–f), fetuses, and placentas (such as Fig. 3g, h) as well as immunofluorescence images (such as Fig. 4m), digital photos (such as Fig. 4b), H&E-stained sections (such as Fig. 4e), ultrasound images (such as Fig. 4f), fluorescence images (such as Fig. 4k), immunohistochemistry analyses of sections of veins with thrombi (such as Supplementary Fig. 17), SEM images of endothelial surface (such as Fig. 4g), and flow cytometric profiles (such as Fig. 4o), all the related experiments were repeated six times with similar results, a series of representative images were shown.

## Statistical analysis

Statistical analysis was performed by SPSS19.0. Flow cytometry data were analyzed with CytExpert 2.4 software (2.4.0.28) and FlowJo 8.0. All quantitative data are expressed as means ± standard deviation (s.d.). The Mauchly's test of sphericity was used to evaluate heterogeneity of variances. Data in accordance with normal distribution was compared by unpaired *t*-test with two groups or one-way ANOVA for more than 2 groups.

## Reporting summary

Further information on research design is available in the Nature Portfolio Reporting Summary linked to this article.

## Data availability

All relevant data supporting the findings of this study are available within the article and its Supplementary Information files or from the corresponding authors upon request. The source data underlying Figs. 2, 3, 4, 5, 6, 7, 8, and 9 and Supplementary Figs. 2, 3, 4, 5, 6, 7, 8, 9, 10, 12, 14, 15, 16, 18, 19, 20, 21, 22, 24, 25, 28, 29, 30, 31, 32, 33, 34, 35, 36, and 38 are provided as a Source Data file. Source data are provided with this paper.

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

## Acknowledgements

This study was supported by the National Natural Science Foundation of China (No. 82171662, to H.Q.), Joint Funds of the National Natural Science Foundation of China (No. U21A20346, to H.Q.), the Natural Science Foundation of Chongqing (No. CSTB2022NSCQ-MSX0856, to J.C.), the Program for Scientific and Technological Innovation Leader of Chongqing (No. CQYC20210302362, to J.Z.), and the Program for Distinguished Young Scholars of TMMU (to J.Z.).

## Author contributions

J.Z. and J.C. conceived the project, and J.Z., J.C., H.Q., and S.Z. designed the experiments. J.C., S.Z., C.L., K.L., X.J., and Q.W. performed all the experiments. J.Z., J.C., S.Z., and H.Q. analyzed the data and composed the manuscript. All authors discussed the results and reviewed the manuscript.

## Competing interests

H.B.Q., J.C., S.Q.Z., and X.Y.J. are inventors in a pending patent filed by the China National Intellectual Property Administration (No. 202210798192.X, July 06 2022) related to TLH NP and CTLH NP, but the rights belong to the Women and Children's Hospital of Chongqing Medical University. All other authors declare that they have no competing interests.
