## [Peer Review File · Nature Communications]

Functionally integrating nanoparticles alleviate deep vein thrombosis in pregnancy and rescue intrauterine growth restrictionREVIEWER COMMENTS

Reviewer #1 (Remarks to the Author):

This is a very interesting manuscript dealing with an innovative topic: In brief, the authors suggest that current treatment of pregnancy-associated deep vein thrombosis has shortcomings, both in terms of patient friendliness (daily LMW heparin-injections over several weeks or months) but also in terms of treatment effectiveness.

In order to overcome this deficit they set out to develop a nanoparticle-based treatment targeting at several different pathophysiologically relevant processes such as coagulation activation, platelet aggregation, thrombolysis, local inflammation and oxidative stress as well as vascular endothelial damage by the ongoing thrombotic process. They also considered the important aspect of protecting the growing fetus from potentially damaging unwanted effects of antithrombotic therapy in the pregnant mother.

Thus they constructed LMWH-based nanoparticles, based on a basic 4-saccharide moiety to which two linoleic acid residues were attached via ester bonds, and furthermore, Tempol was added to one saccharide at the 6' carboxyl group. This compound self associated to nanoparticles and was extensively tested in vitro and in vivo, i.e. in pregnant mouse models with induced left iliac vein thrombosis. Then, the nanoparticles were further modified, attaching a peptide (CREKA) to another 6'-carboxyl group of a saccharide. These nanoparticles, named TLH NP (Tempol/Linoleic acid/LMW-Heparin nanoparticles) or CTLH NP (CREKA/Tempol/Linoleic acid/LMW heparin nanoparticles) assembled the antithrombotic activity of (LMW)-heparin, the antioxidative function of Tempol and the anti-inflammatory role of linoleic acid. Finally, fibrin- (and thereby thrombus-) targeting was achieved by adding CREKA to the nanoparticles.

In vitro and in vivo systematic testing for anticoagulant, thrombolytic, endothelial protective effects and effects on the stenosis-induced iliac vein thrombi in pregnant mice were performed. Safety for placenta and fetus was investigated as well. All tests were systematically performed with the sequentially developed nanoparticles (starting with LH, then TLH, and finally CTLH NPs). Ultimately, an alternative DVT model, i.e. a FeCl₃-induced DVT was similarly studied.

All data are completely and systematically presented and, in summary, show that these nanoparticles, optimally the CTLH NP, are highly effective in resolving the induced thrombi, protecting the endothelium and dampening the inflammatory reaction. Also, the development of the fetus seemed normal and there was good evidence for (almost) no diaplacental transfer of the NP.

The authors conclude that this preclinical model merits further development and study in order to improve the current therapeutic modalities in pregnancy-associated DVT.

Comments, suggestions, criticisms:

- 1) Generally, this seems to be a highly innovative approach for improving therapy of pregnancy-associated venous thrombosis and I could well imagine that the applicability does not have to be restricted to pregnancy-associated DVT. Whether there is such a high clinical need to improve the treatment of pregnancy-associated DVT from the fetal perspective is not as evident to this reviewer. Even though a certain risk for the mother regarding development of osteopenia or frank osteoporosis, and the cumbersome daily sc injections of LMW heparin are indeed concerning, relevant fetal toxicity of current LMW heparin therapy are not a major problem as far as I know.
- 2) I wonder whether the authors did preliminary tests of the anticoagulant effects of their nanoparticles. LMW heparin' anticoagulant effect depend on antithrombin and I wonder whether binding of AT to the NP has been tested. Would it be an idea to include AT in the NP to make the therapy independent from endogenous AT?
- 3) At several places in the manuscript the authors refer to a thrombotic effect of their NPs. In my view, LMW is mainly antithrombotic (in conjunction with endogenous AT) and dissolution of the thrombus rather depends on endogenous fibrinolytic mechanisms. Could the authors comment on this point?
- 4) Specific comment, line 58: Antithrombotic treatment in pregnant women by LMW heparin is almost never associated with heparin-induced thrombocytopenia, and congenital malformations are exclusively associated with coumarins given during embryogenesis as to my knowledge.
- 5) Line 67: This should read "...mainly caused by twice-a-day..." instead of "...causing..."

- 6) Line 88: I believe that the nature of this "fibrin-targeting peptide" should be somewhat outlined (here or later in the MS). What peptide is it, where in fibrin does it specifically bind.
- 7) Line 103: as mentioned above, the anticoagulant effect of this nanotherapy will likely depend on endogenous antithrombin. Could this be relevant, e.g. in an antithrombin deficiency state?
- 8) Lines 128ff: I appreciate how the authors try to carefully attribute the antioxidant effects to the individual components of their NPs.
- 9) Lines 150ff: Again, I am not sure whether true thrombolytic therapy is provided by the NPs or rather by the endogenous fibrinolytic system (whose activity might be improved by the NPs)? Further, I wonder why LMWH should be primarily responsible to mediate binding to activated platelets and tissue factor in thrombi? Could the authors explain their reasoning?
- 10) Lines 240ff: The comparison of the NP-mediated anti-FXa activity with that mediated by sc injection of LMWH is excellent and shows one major advantage of the new therapeutic modality.
- 11) Lines 276ff: I am puzzled about the major role of DVT itself on the impairment of the fetal development in this mouse model. Is this similarly relevant in human pregnancy-associated DVT? (which, of course, hardly remains untreated). Is Reference 15 a relevant and correct reference to this point?
- 12) Lines 311-317: Convincingly presented further improvement of the NPs by introducing CREKA making the NPs more specifically directed to the thrombus. Nevertheless, in case of a preceding surgical intervention or traumatic lesion, the NPs would obligatorily lose their thrombus specificity and also potentially target "hemostatically needed fibrin clots".
- 13) Line 345: "...representative pro-thrombolytic factors..." is a misnomer: TAFI and PAI-1 are anti-thrombolytic factors, not pro-thrombolytic.
- 14) Line 402: "...D2-dimer." seems a misnomer?
- 15) Lines 440-445: Excellent summary, however, "anti-thrombolytic nanotherapies" is a very bad misnomer, please, correct.
- 16) Figure 1: Here, CREKA is already presented, but all initial experiments are done with NPs not yet containing this peptide moiety. Could the authors somewhat detail the Figure legend? And again, is "Thrombolysis" directly provided by the NPs, or by the endogenous fibrinolytic system?
- 17) Figure 3, panel A: is the delay between ligation, DVT appearance, and start of treatment unambiguously clear? and Panel L: Explain RP-1 and CD11b in the legend. Percent of Neutrophils corresponds to the upper right rectangle. Would the absolute number of neutrophils be more relevant?
- 18) Figure 4: Again, I am impressed by the dramatic effect of DVT (model) on the fetal development. Does this have any bearing on human situation?
- 19) Figure 5: What exactly is the "Blood clotting index"? What is "cleaned deionized water"?
- 20) Figure 6: In Panel G, the fetus in the "model" does not seem to be "small for age". In the legend to Figure 6 "FITC" is misspelled.
- 21) I congratulate the authors for this excellent concept, the carefully developed multifunctional nanoparticles, the carefully done experiments and the interesting paper.

Bernhard Lämmle
21-November-2021

Reviewer #2 (Remarks to the Author):

This manuscript, by Chen et al, demonstrates the potential of Tempol and linoleic acid, conjugated to LMWH, as a potential nanoparticle therapy for DVT in pregnancy. The authors use multiple approaches to test their hypothesis and conclude that these nanoparticles are safe, do not cross the placenta to reach the fetus, and efficiently dissolve thrombi and improve pregnancy outcomes in a rat model as a result. The manuscript appears novel and addresses an important topic. I do have some queries regarding methodology, analysis and interpretation which I feel need to be addressed.

Major comments/queries:

- For many of the studies, there are controls to the LH, TLH and CTLH groups in the form of LMWH or Tempol alone. However, there is not a comparable linoleic acid control nor groups that were exposed to tempol, LMWH and linoleic acid together as a bolus to directly compare the NP conjugate groups to. This makes comparisons, including full mechanistic insight and the added value of the nanoparticles, problematic.
- To further this comment, in figure 3G, some markers of inflammation are measured but as linoleic acid has anti-inflammatory properties, it is perhaps not surprising that the NP groups with linoleic acid showed reduced levels of IL-6 and TNF-alpha. A linoleic acid control group would have shown the added value of the LH and TLH NP groups but this is not discussed.
- For the rat studies, I was unable to find the detail on n's nor whether the data presented (including fetal weight and placental weight) is expressed as litter averages or individual fetuses/placentas. This needs to be explained and justified with litter averages being the gold standard. There is also no information on whether litter size was affected in any of these groups- please provide this information.
- Many of the figures describe an n=4 but I find it difficult to see how normality can be appropriately determined with such a small number. This makes the use of any parametric statistical tests an issue. Additionally, with n=4, the use of bar-graphs are not very intuitive and dot plots as in Figures 4 and 8 are preferable.
- For the data in figures 2G/H and 6G/H it is stated that Cy5 fluorescence was limited to the 'maternal side' of the placenta. Please be specific on this, are you saying there is only partial staining in the decidua? In figures 2G and 6G, there is a fluorescent halo around the placenta which usually corresponds to the junctional zone of the rat placenta and implies staining should be present within this zone. However, in Fig 2H and 6H, there is only very limited staining on the periphery of the placenta which seems at odds with this? This is also even more apparent in supplementary Figure 20H for the Cy5-TLH NP model example. Please can you comment.
- In figures 4E and 8E it is difficult to make out any detail on placental morphology from these images. Adding a closer magnification of the junctional and labyrinthine zones would be more intuitive for the reader. Additionally, in the skeletal staining, examples of the TLH NP group in figures 4F and 8E both suggest a much smaller skeleton but this does not appear to fit with the fetal weight data. This should be discussed, as should the fact that whilst there is more mineralisation of the fetal skeletons in NP v model groups, the ribs in particular still appear underdeveloped versus the normal fetus. Do you have any idea about postnatal viability of treated fetuses and whether they have respiratory issues? Were there gross fetal morphological abnormalities in any of the groups? It is difficult to tell from the figures showing multiple fetuses.
- In the methods section, the information on animals is very brief and does not include information on housing conditions or number of animals. Ideally, the ARRIVE guidelines should be followed (Kilkenny et al) for reporting of in vivo experiments.
- The number of animal studies are very complex and difficult to follow. A flow chart highlighting the different studies and n's used would be helpful for the reader.
- The discussion is very brief and represents mainly a re-cap of the major findings. There is very limited discussion of how this data compares with published work and a lack of consideration of the limitations of this study. This needs to be addressed.

Minor points:

- In the introduction (lines 61-62) it is stated that aspirin is prohibited in the last trimester of pregnancy due to risk of bleeding with a reference from 1998. This statement is incorrect. In higher risk pregnancies, aspirin may be prescribed, often up until approx 36 weeks of pregnancy. Please amend or delete this statement.
- On line 67, the necessity to inject LMWH twice-daily is listed as an issue with compliance. If this nanoparticle therapy is translated to patients, I presume patients would have to visit hospital for regular i.v injections? This could also prove a problem with compliance and should be discussed.
- Information of the statistical test used should be added to each figure legend.

Reviewer #3 (Remarks to the Author):

In this research article, the authors develop a nanoparticulate platform for the treatment of deep vein thrombosis. Low molecular weight heparin (LMWH) is conjugated to Tempol as an antioxidative agent and lauric acid as an anti-inflammatory agent. The construct is further functionalized with CREKA, which targets fibrin. After fabricating the nanoparticles by self-assembly and performing in vitro characterizations, the authors evaluate their platform in vivo. It is shown that the nanoparticles effectively target sites of thrombosis and have considerable therapeutic efficacy in multiple rodent models. Overall, the authors have performed a substantial amount of work to support their claims, although there are still some important points that need to be addressed.

1. Structurally, the manuscript was made unnecessarily long because the authors didn't add the CREKA moiety until later. Why did they not just add this experimental group to the first set of in vivo studies?
2. The main figures are focused mainly on the in vivo studies, but the formulation development data should also be included in the main manuscript.
3. It is unclear how the authors normalized the dosages for the formulations. The most reasonable approach is to evaluate the formulations at the same concentration/dosage of LMWH. If the authors used a different approach, then they need to justify why they believe that was most appropriate. Regardless, this point needs to be explicitly clarified.
4. The mechanism for self-assembly of the nanostructure should be discussed more in detail.
5. There are multiple examples of interesting and unexpected phenomena that go unexplained by the authors.
 - a. The anticoagulant activity of the LMWH was halved as a result of its incorporation into the nanoformulation. The authors should discuss the reason for this decrease. It is also odd that the nanoformulations outperform free LMWH in the in vitro anticoagulation assays despite having significantly reduced activity.
 - b. It is claimed that Tempol has high affinity to activated platelets. Do the authors have any references to support this claim? If not, what explanation do they have for this phenomenon, since Tempol was included as the ROS scavenger and not as a targeting moiety. Overall, it is hard to imagine that the Tempol can have such a marked effect on targeting both in vitro and in vivo.
 - c. How did the LH NP formulation attenuate ROS-induced endothelial cell apoptosis when it does not have an ROS-scavenging component?
6. The dose-dependent uptake data in Fig. S8 is unexpected. For example, why would the uptake at 200 ug/mL be more than double the uptake at 100 ug/mL? This points to a faulty experimental setup or insufficient washing of the samples prior to analysis.
7. Subcutaneous LMWH does not provide a fair comparison for the nanoformulations, which are administered intravenously. The authors should consider comparing the nanoformulations with a "safe" LMWH intravenous dose for the in vivo studies. This is reasonable considering that the authors demonstrated that LMWH has higher biological activity in free form compared with in nanoparticulate form.
8. The authors claimed that "no beneficial effects were observed after treatment with Tempol" in Figure 4, yet there was a significant recovery of fetal weight.
9. It is incorrectly stated that "the fluorescent intensity of the Cy5-CTLH NP group was 1.6-fold higher than that of the Cy5-TLH NP group" in Figure 6A. The difference between the two groups is much less than 1.6-fold.

Reviewer #4 (Remarks to the Author):

This is an experimental rat study of gestation and induced DVT, to assess a novel nano-compound, with fibrin binding for LMWH and tempol and linoleic acid.

1. The stenosis model of iliac vein is not well described -- how was the degree of stenosis determined for reproducibility and reliability?
2. Were the agents given in a blinded fashion separate from the surgical model?
3. It is hard to define the range of thrombi in terms of resolution without a dot plot -- please change that up. And, it is hard to reconcile, knowing the stenosis model is quite variable in terms of thrombogenesis, that some % of rats will have no thrombus whatsoever. This needs to be clarified.
4. Can you characterize the thrombus from the control vs agents, in terms of fibrin content, platelet content, etc - -given you claim that your agent impacts all these components?
5. Did you assess the thrombus leukocyte influx and characterize this? Would be interesting to do with your compound's potential effects.
6. What were the statistical methods used for these multi-comparisons? Again, I find it hard to believe a N = 4 at each treatment would give you adequate power for such dramatic P values.

Response to Reviewers

REVIEWER COMMENTS

Reviewer #1 (Remarks to the Author):

This is a very interesting manuscript dealing with an innovative topic: In brief, the authors suggest that current treatment of pregnancy-associated deep vein thrombosis has shortcomings, both in terms of patient friendliness (daily LMW heparin-injections over several weeks or months) but also in terms of treatment effectiveness.

In order to overcome this deficit they set out to develop a nanoparticle-based treatment targeting at several different pathophysiologically relevant processes such as coagulation activation, platelet aggregation, thrombolysis, local inflammation and oxidative stress as well as vascular endothelial damage by the ongoing thrombotic process. They also considered the important aspect of protecting the growing fetus from potentially damaging unwanted effects of antithrombotic therapy in the pregnant mother.

Thus they constructed LMWH-based nanoparticles, based on a basic 4-saccharide moiety to which two linoleic acid residues were attached via ester bonds, and furthermore, Tempol was added to one saccharide at the 6' carboxyl group. This compound self associated to nanoparticles and was extensively tested in vitro and in vivo, i.e. in pregnant mouse models with induced left iliac vein thrombosis. Then, the nanoparticles were further modified, attaching a peptide (CREKA) to another 6'-carboxyl group of a saccharide. These nanoparticles, named TLH NP (Tempol/Linoleic acid/LMW-Heparin nanoparticles) or CTLH NP (CREKA/Tempol/Linoleic acid/LMW heparin nanoparticles) assembled the antithrombotic activity of (LMW)-heparin, the antioxidative function of Tempol and the anti-inflammatory role of linoleic acid. Finally, fibrin- (and thereby thrombus-) targeting was achieved by adding CREKA to the nanoparticles.

In vitro and in vivo systematic testing for anticoagulant, thrombolytic, endothelial protective effects and effects on the stenosis-induced iliac vein thrombi in pregnant mice were performed. Safety for placenta and fetus was investigated as well. All tests were systematically performed with the sequentially developed nanoparticles (starting with LH, then TLH, and finally CTLH NPs). Ultimately, an alternative DVT model, i.e. a FeCl₃-induced DVT was similarly studied.

All data are completely and systematically presented and, in summary, show that these nanoparticles, optimally the CTLH NP, are highly effective in resolving the induced thrombi, protecting the endothelium and dampening the inflammatory reaction. Also, the development of the fetus seemed normal and there was good evidence for (almost) no diaplacental transfer of the NP.

The authors conclude that this preclinical model merits further development and study in order to improve the current therapeutic modalities in pregnancy-associated DVT.

Response: We deeply appreciate the reviewer's careful reading of our manuscript and the valuable comments provided. According to the related comments, additional experiments were performed and the manuscript was substantially revised.

Comments, suggestions, criticisms:

1) Generally, this seems to be a highly innovative approach for improving therapy of pregnancy-associated venous thrombosis and I could well imagine that the applicability does not have to be restricted to pregnancy-associated DVT. Whether there is such a high clinical need to improve the treatment of pregnancy-associated DVT from the fetal perspective is not as evident to this reviewer.

Even though a certain risk for the mother regarding fetal development of osteopenia or frank osteoporosis, and the cumbersome daily sc injections of LMW heparin are indeed concerning, relevant fetal toxicity of current LMW heparin therapy are not a major problem as far as I know.

Response: Thanks for this good suggestion.

Indeed, as indicated by the reviewer, our nanotherapies may not be restricted to pregnancy-associated DVT. We will further verify their applications for other thrombosis-related diseases, such as atherosclerosis and abdominal aortic aneurysm.

Of note, some clinical studies showed that thrombosis has negative impacts on pregnancy such as fetal growth restriction (FGR). According to evidence-based investigation, there is a link between DVT and FGR delivered after 37 weeks^{1, 2}. In addition, clinical guidelines on DVT during pregnancy suggested that modification of blood flow in pregnant women with thrombosis and hypercoagulability in the second and third trimesters of pregnancy can improve fetal growth and development³.

It is true that LMWH cannot pass through the placental barrier and has less toxicity on the fetus. However, LMWH has a short half-life, and therefore it needs injection multiple times per day, especially subcutaneous injection in the abdomen, which will lead to poor compliance for pregnant women⁴. Meanwhile, although LMWH reduces the incidence of hemorrhage, the related risk still exists. Once this condition happens, most hemostatic drugs cannot be given to pregnant women, which is likely to cause adverse pregnancy outcomes. Accordingly, more effective and safe therapies are still required for pregnant women with thrombosis.

According to the above comment, some relevant descriptions were revised (page 2).

2) I wonder whether the authors did preliminary tests of the anticoagulant effects of their nanoparticles. LMW heparin' anticoagulant effect depend on antithrombin and I wonder whether binding of AT to the NP has been tested. Would it be an idea to include AT in the NP to make the therapy independent from endogenous AT?

Response: Thanks for this good question.

We co-incubated either LMWH or nanotherapies (LH NP and TLH NP) with rat plasma to detect the coagulation index (PT, APTT, and TT). Compared with PBS, both LMWH and nanotherapies significantly increased PT, APTT, and TT, thereby showing an obvious anticoagulant effect. Notably, the anticoagulant effect of nanotherapies is similar to that of LMWH. The related results have been presented in Fig. 2m-o.

Fig. 2m-o Effects of different formulations on PT (m), APTT (n), and TT (o) values after rat plasma was incubated with LMWH and LMWH-derived nanotherapies. Data are mean \pm s.d. (n = 6). Statistical significance was assessed by one-way ANOVA with post hoc LSD tests. * $p < 0.05$, ** $p < 0.01$, *** $p < 0.001$.

On the other hand, LMWH exerts its anticoagulant effect by activating antithrombin activity rather than directly binding to antithrombin⁵. LMWH will combine with thrombin and antithrombin rapidly

to form the thrombin-antithrombin complex (TAT). Accordingly, we examined the TAT formation by co-incubating thrombin, antithrombin, and nanotherapies to verify the antithrombin binding activity of different nanotherapies. Compared with the PBS control, LMWH and LMWH-derived nanotherapies significantly promoted the TAT formation. Notably, nanotherapies even showed more significantly enhanced activity than LMWH. The related results have been presented in Fig. 2p of the revised version.

Fig. 2p The effect of different formulations on the TAT formation after incubation with thrombin and antithrombin. Data are mean \pm s.d. (n = 6). Statistical significance was assessed by one-way ANOVA with post hoc LSD tests. * $p < 0.05$, *** $p < 0.001$.

Moreover, the formation of TAT by specific interactions between antithrombin and thrombin, which is irreversible, will reduce free thrombin in vivo ⁶. Since thrombin can promote the transformation of fibrinogen into fibrin, a decrease in thrombin further inhibits the expression of fibrin in vivo. Therefore, antithrombin increases the bleeding risk in pregnant women, and antithrombin is not recommended for diseases in pregnancy in clinical practice. Accordingly, we consider that antithrombin-conjugated nanoparticles might also lead to other uncontrolled bleeding complications, particularly in the case of pregnant women.

3) At several places in the manuscript the authors refer to a thrombotic effect of their NPs. In my view, LMW is mainly antithrombotic (in conjunction with endogenous AT) and dissolution of the thrombus rather depends on endogenous fibrinolytic mechanisms. Could the authors comment on this point?

Response: Thanks for this good question.

As well documented, tissue-type plasminogen activator (tPA) exhibits notable thrombolytic effects ⁷. The expression of free tPA can be specifically inhibited by plasminogen activator inhibitor-1 (PAI-1), thereby leading to reduced thrombolysis ⁸. As indicated by our newly performed experiments, LMWH and LMWH-derived nanotherapies can increase the level of free tPA by reducing the expression of PAI-1, thereby exerting their thrombolytic effect in vivo. The related results have been presented in Supplementary Fig. 20a,e of the revised supplementary information.

Supplementary Fig. 20 | a,e Relative mRNA levels of PAI-1 (a) and tPA (e) in left iliac veins. Left iliac veins of pregnant rats at G10 were ligatured to induce DVT. At 6 h after the formation of stenosis-induced thrombi, diseased rats were daily administered with saline (the model group), LA (2 mg/kg), LMWH (3 mg/kg), TLH bolus (*i.e.*, a mixture containing 8 mg/kg Tempol, 2 mg/kg LA, and

3 mg/kg LMWH), LH NP (at 3 mg/kg LMWH), or TLH NP (at 3 mg/kg LMWH) by *i.v.* injection for 5 days. In the normal group, healthy pregnant rats with sham operation were treated with saline. Data are mean \pm s.d. (n = 6). Statistical significance was assessed by one-way ANOVA with post hoc LSD tests. * $p < 0.05$, ** $p < 0.01$, *** $p < 0.001$.

4) Specific comment, line 58: Antithrombotic treatment in pregnant women by LMW heparin is almost never associated with heparin-induced thrombocytopenia, and congenital malformations are exclusively associated with coumarins given during embryogenesis as to my knowledge.

Response: We appreciate this valuable comment.

Actually, we mentioned adverse effects of other antithrombotic drugs. For example, warfarin can cross the placenta barrier and cause fetal malformation and excessive bleeding⁹. Indomethacin may inhibit platelet aggregation, but it causes premature occlusion of fetal ductus arteriosus¹⁰. LMWH is a relatively safe antithrombotic agent, and therefore it was chosen as a bioactive scaffold for constructing nanotherapies. To avoid misunderstanding, the related description was re-organized in the revised version (page 2).

5) Line 67: This should read "...mainly caused by twice-a-day..." instead of "...causing..."

Response: We apologize for this mistake, which has been corrected in the revised manuscript.

6) Line 88: I believe that the nature of this "fibrin-targeting peptide" should be somewhat outlined (here or later in the MS). What peptide is it, where in fibrin does it specifically bind.

Response: Thanks for this good suggestion.

CREKA, a tumor-homing peptide, can effectively bind to fibrin and clotted plasma proteins in the blood vessels and stroma of tumors. Therefore, CREKA was chosen as a targeting unit to develop an active targeting nanotherapy toward thrombi. We added the related contents in the revised manuscript (page 3).

7) Line 103: as mentioned above, the anticoagulant effect of this nanotherapy will likely depend on endogenous antithrombin. Could this be relevant, e.g. in an antithrombin deficiency state?

Response: Thanks for this good question.

The anticoagulant effect of nanotherapies mainly depends on peripheral LMWH segments on the corresponding nanoparticles. Indeed, as indicated by the reviewer, the anticoagulant effect of LMWH is associated with endogenous antithrombin. Whereas LMWH-derived nanotherapies can enhance the anticoagulant activity of LMWH, their effect is also related to endogenous antithrombin. Antioxidant and anti-inflammatory effects alone might not afford desirable thrombolytic outcomes. Therefore, we consider that nanotherapies cannot exert a potent anticoagulant effect in the absence of antithrombin.

8) Lines 128ff: I appreciate how the authors try to carefully attribute the antioxidant effects to the individual components of their NPs.

Response: Thanks for this good question.

The antioxidant effect of TLH NP is mainly attributed to Tempol, a superoxide dismutase (SOD) analog that can scavenge different reactive oxygen species (ROS) *in vitro* and *in vivo*, such as superoxide anion, free radical, hydrogen peroxide, and hypochlorite¹¹. This issue is supported by the results illustrated in Fig. 2h-k and our previous studies¹².

Fig. 2h-k Dose-dependent elimination of superoxide anion (h), DPPH radical (i), H₂O₂ (j), and hypochlorite (k) by TLH NP. Data are mean ± s.d. (n = 6).

9) Lines 150ff: Again, I am not sure whether true thrombolytic therapy is provided by the NPs or rather by the endogenous fibrinolytic system (whose activity might be improved by the NPs)? Further, I wonder why LMWH should be primarily responsible to mediate binding to activated platelets and tissue factor in thrombi? Could the authors explain their reasoning?

Response: Thanks for these good questions.

PAI-1 can reduce the expression of free tPA, thereby inhibiting thrombolysis. LMWH increases the activity of free tPA to activate the endogenous fibrinolytic system by decreasing PAI-1. Meanwhile, our nanotherapies can prolong the half-life of LMWH and enhance its targeting ability, thereby notably promoting thrombolysis.

According to previous findings, platelet activation is mainly mediated by the interaction between thrombin and platelet glycoprotein Ib (GP-Ib)¹³. LMWH can replace the binding site of thrombin with GP-Ib to inhibit the thrombin-mediated platelet activation¹⁴. Also, LMWH may bind to the C-terminal part of tissue factor pathway inhibitor (TFPI), while TFPI can further combine with tissue factor to inhibit its expression in thrombi¹⁵.

According to the above comment, we added some relevant contents in the revised version (pages 5 and 8).

10) Lines 240ff: The comparison of the NP-mediated anti-FXa activity with that mediated by sc injection of LMWH is excellent and shows one major advantage of the new therapeutic modality.

Response: Thanks for this comment.

11) Lines 276ff: I am puzzled about the major role of DVT itself on the impairment of the fetal development in this mouse model. Is this similarly relevant in human pregnancy-associated DVT? (which, of course, hardly remains untreated). Is Reference 15 a relevant and correct reference to this point?

Response: Thanks for this good question.

Increasing clinical evidence has indicated that thrombosis has a negative impact on pregnancy, such as fetal growth restriction (FGR)¹. In particular, thrombosis is considerably associated with FGR delivered after 37 weeks². In this study, the related pathogenesis was further explored to explain this phenomenon. There is evidence that thrombosis can indirectly affect placental function, because thrombosis will cause hypoxia in the placenta, thereby indirectly impairing placental functions by immunosuppression, delaying trophoblast development, and inducing apoptosis of trophoblast cells¹⁶. In addition, hypoxia of placental trophoblast cells will accelerate the release of PAI-1, resulting in more coagulation proteins and microthrombi, which may deposit in placental vessels and further affect fetal

development². The relevant contents were added in the revision version (page 8).

We apologize for the mistake that Reference 15 is not a correct reference, which has been corrected in the revised version.

12) Lines 311-317: Convincingly presented further improvement of the NPs by introducing CREKA making the NPs more specifically directed to the thrombus. Nevertheless, in case of a preceding surgical intervention or traumatic lesion, the NPs would obligatorily lose their thrombus specificity and also potentially target "hemostatically needed fibrin clots".

Response: Thanks for this good question.

For thrombi caused by surgical intervention and traumatic injury, they also show an increased fibrin content, and therefore CREKA-decorated nanotherapies can also target these thrombi by binding with fibrin¹⁷. Moreover, previous bleeding experiments in a rat model of the tail vein bleeding showed that LMWH-derived nanotherapies did not prolong the bleeding time (Supplementary Fig. 14). Indeed, as indicated by the reviewer, in hemostatically necessary cases, the applications of our nanotherapies and other antithrombotic agents should be limited to avoid uncontrolled hemorrhage. Some relevant contents were added in the revised manuscript (page 12).

Supplementary Fig. 14 | The effects of different formulations on tail vein bleeding in pregnant

rats. (a-b) Quantified bleeding time (a) and volume (b) after injection of different formulations.

LMWH was administered by either *i.v.* injection at 3 and 5 mg/kg or subcutaneous (*s.c.*) injection at 5 mg/kg, while all nanotherapies were delivered via *i.v.* injection at 5 mg/kg of LMWH. In the normal group, rats were treated with saline. At 10 min after administration of different formulations, pregnant rats were subjected to cutting off the entire tail tip. Both bleeding time and bleeding volume were quantified. Data are mean \pm s.d. ($n = 8$). Statistical significance was assessed by one-way ANOVA with post hoc LSD tests. * $p < 0.05$, *** $p < 0.001$; ns, no significance.

13) Line 345: "...representative pro-thrombolytic factors..." is a misnomer: TAFI and PAI-1 are anti-thrombolytic factors, not pro-thrombolytic.

Response: We are sorry for this mistake.

In the revised version, "pro-thrombolytic factors" has been changed into "anti-thrombolytic factors".

14) Line 402: "...D2-dimer." seems a misnomer?

Response: We are sorry for this typo error.

In the revised version, "...D2-dimer" has been corrected as "...D-dimer".

15) Lines 440-445: Excellent summary, however, "anti-thrombolytic nanotherapies" is a very bad

misnomer, please, correct.

Response: Sorry for this misnomer.

In the revised version, "anti-thrombolytic nanotherapies" has been corrected as "thrombolytic nanotherapies".

16) Figure 1: Here, CREKA is already presented, but all initial experiments are done with NPs not yet containing this peptide moiety. Could the authors somewhat detail the Figure legend? And again, is "Thrombolysis" directly provided by the NPs, or by the endogenous fibrinolytic system?

Response: Thanks for these good questions.

Initially, we designed and developed LH NP and TLH NP. After confirmation of their desirable anti-thrombotic effects in the examined DVT model and in order to develop a nanotherapy with more beneficial efficacy, CREKA was used to functionalize TLH, thereby affording an active targeting nanotherapy CTLH NP. Therefore, these studies were performed step by step rather than at the same time. In Fig. 1, we only provide a schematic illustration of the nanotherapy CTLH NP with the best therapeutic effect on thrombosis to give impressive notification to readers.

Also, we appreciate the reviewer's consideration regarding thrombolytic effects of nanotherapies. As we answered in above responses, tPA exhibits a thrombolytic effect, while PAI-1 can inhibit thrombolysis by interacting with tPA⁸. LMWH can increase the level of free tPA by reducing the expression of PAI-1, thereby exerting the thrombolytic effect. Similarly, the thrombolytic activity of our nanotherapies depends on the endogenous fibrinolytic system.

17) Figure 3, panel A: is the delay between ligation, DVT appearance, and start of treatment unambiguously clear? and Panel L: Explain RP-1 and CD11b in the legend. Percent of Neutrophils corresponds to the upper right rectangle. Would the absolute number of neutrophils be more relevant?

Response: Thanks for these questions.

In order to control the stability of the thrombosis model, we confirmed the formed thrombus in pregnant rats in preliminary experiments, by measuring the thrombus weight at different time points (0, 3, 6, 12, and 24 h) after ligation. It was found that the thrombosis rate was low after ligation within the first 3 h, while the formed thrombus remained stable for 6-24 h. The related result is shown in Supplementary Fig. 12 of the revised supplementary information. Once thrombosis is developed, different treatments need to be performed. Therefore, we conducted treatment via different formulations at 6 h after ligation.

Supplementary Fig. 12 | The weight of thrombi in left iliac veins at different time points after ligation. Data are mean \pm s.d (n = 6). Statistical significance was assessed by one-way ANOVA with post hoc LSD tests. *** $p < 0.001$; ns, no significance.

RP-1 is a monoclonal antibody against rat neutrophils, while CD11b is generally expressed on the

surface of granulocytes. RP-1⁺ and CD11b⁺ mark neutrophils. We added the related contents in the corresponding legend of the revised manuscript.

Due to the difference in cell numbers for different samples, there are certain errors if absolute cell counts are used. By contrast, the percentage of neutrophils has relatively small errors. According to the reviewer's comment, we also provided the absolute cell counts of neutrophils in the revised supplementary material (Supplementary Fig. 18a,18d, 29c, 29f).

Supplementary Fig. 18 | a,d Quantified cell counts of neutrophil (CD11b⁺/RP-1⁺) in the left iliac veins (a) and peripheral blood (d). Left iliac veins of pregnant rats at G10 were ligatured to induce DVT. At 6 h after the operation, DVT rats were administered with saline (the model group), free Tempol (8 mg/kg), free LMWH (5 mg/kg), LH NP (at 5 mg/kg of LMWH), or TLH NP (at 5 mg/kg of LMWH) by daily injection for 5 days. In the normal group, healthy pregnant rats with sham operation were treated with saline. Blood samples and veins with thrombi were collected at G17 for flow cytometric analysis. Data are mean ± s.d. (n = 6). Statistical significance was assessed by one-way ANOVA with post hoc LSD tests. **p* < 0.05, ***p* < 0.01, ****p* < 0.001.

Supplementary Fig. 29 | c,f Quantified cell numbers of CD11b⁺/RP-1⁺ neutrophils in the left iliac veins (c) or peripheral blood (f). In these studies, left iliac veins of pregnant rats at G10 were ligatured to induce DVT. At 6 h after the stenosis operation, DVT rats were daily *i.v.* administered with saline (the model group), TLH NP at 5 mg/kg of LMWH, or CTLH NP at 5 mg/kg of LMWH for 5 days. In the normal group, healthy pregnant rats with sham operation were treated with saline. Peripheral blood samples and veins with thrombi were collected for the corresponding analysis at G17. Data are mean ± s.d. (n = 6). Statistical significance was assessed by one-way ANOVA with post hoc LSD tests. ****p* < 0.001.

18) Figure 4: Again, I am impressed by the dramatic effect of DVT (model) on the fetal development. Does this have any bearing on human situation?

Response: Thanks again for this important question.

Increasing clinical evidence has indicated that thrombosis has a negative impact on pregnancy, such as fetal growth restriction (FGR) ¹. In particular, thrombosis is considerably associated with FGR delivered after 37 weeks ². In this study, the related pathogenesis was further explored to explain this phenomenon. There is evidence that thrombosis can indirectly affect placental function, because thrombosis will cause hypoxia in the placenta, thereby indirectly impairing placental functions by

immunosuppression, delaying trophoblast development, and inducing apoptosis of trophoblast cells¹⁶. In addition, hypoxia of placental trophoblast cells will accelerate the release of PAI-1, resulting in more coagulation proteins and microthrombi, which may deposit in placental vessels and further affect fetal development².

19) Figure 5: What exactly is the "Blood clotting index"? What is "cleaned deionized water"?

Response: Thanks for these questions.

Herein the blood clotting index (BCI) is used to reflect the agglutination ability of red blood cells¹⁸. The BCI will increase with increased free hemoglobin in solutions. A higher BCI means weaker blood clotting, while a lower BCI value represents an enhanced hemostatic activity for the examined formulations.

$$\text{BCI (\%)} = (\text{Absorbance of sample} / \text{Absorbance of blank}) \times 100\%$$

As for "cleaned deionized water", it means 15 mL of deionized water which was used for cleaning the untrapped clot.

In the revised figure, some related contents were added for a clearer presentation.

20) Figure 6: In Panel G, the fetus in the "model" does not seem to be "small for age". In the legend to Figure 6 "FITC" is misspelled.

Response: We appreciate the above comment.

Firstly, each female had a large number of offspring (6-17). Fetuses were also different in the model group. Pregnant rats were euthanized on G15, and the difference in fetal sizes was relatively small. In addition, different batches of rats may affect the experimental results. Therefore, we repeated the related experiments. The new results further indicated that fetuses in the model group are generally smaller than those in the normal group (Fig. 7g).

Fig. 7g *Ex vivo* imaging of placentas and fetuses at 4 h after *i.v.* injection of Cy5-TLH NP or Cy5-CTLH NP in normal or stenosis-induced DVT pregnant rats (at G15).

In addition, "FTIC" has been corrected as "FITC" in the legend of Figure 7.

21) I congratulate the authors for this excellent concept, the carefully developed multifunctional nanoparticles, the carefully done experiments and the interesting paper.

Response: Thanks for the positive comments on our manuscript. Also, we really appreciate the constructive questions and suggestion proposed by the reviewer, which are helpful for further improving the quality of our paper.

Reviewer #2 (Remarks to the Author):

This manuscript, by Chen et al, demonstrates the potential of Tempol and linoleic acid, conjugated to

LMWH, as a potential nanoparticle therapy for DVT in pregnancy. The authors use multiple approaches to test their hypothesis and conclude that these nanoparticles are safe, do not cross the placenta to reach the fetus, and efficiently dissolve thrombi and improve pregnancy outcomes in a rat model as a result. The manuscript appears novel and addresses an important topic. I do have some queries regarding methodology, analysis and interpretation which I feel need to be addressed.

Response: We really appreciate the reviewer's careful reading of our manuscript and the proposed helpful comments. According to these comments, additional experiments were performed and the manuscript was substantially revised.

Major comments/queries:

- For many of the studies, there are controls to the LH, TLH and CTLH groups in the form of LMWH or Tempol alone. However, there is not a comparable linoleic acid control nor groups that were exposed to tempol, LMWH and linoleic acid together as a bolus to directly compare the NP conjugate groups to. This makes comparisons, including full mechanistic insight and the added value of the nanoparticles, problematic.

Response: Thanks for this good suggestion.

According to the above comment, we performed additional experiments to cover the controls mentioned by the reviewer, including LA and the mixture of Tempol/ LA/LMWH (i.e., TLH bolus). The related results have been presented in Supplementary Fig. 6,10, 19-23 of the revised version.

Supplementary Fig. 6 | Comparison of *in vitro* anticoagulant effects of two nanotherapies with LA and a mixture of Tempol, LA, and LMWH. (a) SEM images showing thrombin-stimulated platelets after separate treatment with PBS, LA, LH NP, TLH bolus (*i.e.*, a mixture of Tempol, LA, and LMWH), or TLH NP. Platelets treated with PBS alone served as the normal control. Scale bar, 3 μm . (b-c) Typical digital photos (b) and quantified BCI values (c) show anti-coagulant activities of different formulations. (d-e) Representative images (d) and quantified dissolution degrees (e) of blood thrombi after treatment with different formulations. Data in (c,e) are mean \pm s.d. ($n = 6$). Statistical significance was assessed by one-way ANOVA with post hoc LSD tests. $*p < 0.05$, $**p < 0.01$, $***p < 0.001$.

Supplementary Fig. 10 | Comparison of *in vitro* antioxidative and endothelial protective effects of two nanotherapies with LA and a mixture of Tempol, LA, and LMWH. (a-b) Fluorescence images (a) and flow cytometric quantification (b) of DHE-stained HUVECs after treatment with different formulations and stimulation with H_2O_2 . Scale bar, 20 μm . (c-d) Representative flow cytometric profiles (c) and quantitative analysis (d) of apoptotic HUVECs after different pre-treatments and stimulation with H_2O_2 . In both cases, the normal control group was treated with PBS alone, while the model group was treated with H_2O_2 alone. For the TLH bolus group, a mixture of Tempol, LA, and LMWH was employed. (e-f) Microscopic images (e) and quantitative analysis (f) of migrated HUVECs after separate treatment with PBS (control), free LA, LH NP, TLH bolus (i.e., a mixture of Tempol, LA, and LMWH), and TLH NP for 12 h. Migrated cells were stained with crystal violet. Scale bar, 200 μm . (g-h) Microscopic images (g) and quantified healing rates (h) based on the wound healing assay after separate incubation with different formulations for 24 h. Scale bar, 800 μm . Data in (b,d,f,h) are mean \pm s.d. ($n = 6$). Statistical significance was assessed by one-way ANOVA with post hoc LSD tests. * $p < 0.05$, ** $p < 0.01$, *** $p < 0.001$; ns, no significance.

Supplementary Fig. 19 | Comparison of therapeutic effects of two nanotherapies with LA, the TLH bolus, and *i.v.* LMWH in pregnant rats with stenosis-induced DVT. (a) Representative digital photos of left iliac veins with thrombi isolated from pregnant rats after treatment with different formulations. Scale bar, 1 mm. (b-c) The weight (b) and length (c) of thrombi in left iliac veins. (d) H&E-stained histopathological sections of left iliac veins with thrombi. Scale bar, 400 μ m. (e) Ultrasound images of left iliac veins. The yellow dashed lines indicate the vascular endothelium, while the red dashed lines indicate thrombi. Scale bar, 1 mm. (f) SEM observation of the endothelial surface of left iliac veins with thrombi. Scale bar, 10 μ m. In all these studies, left iliac veins of pregnant rats at G10 were ligatured to induce DVT. At 6 h after the formation of stenosis-induced thrombi, diseased rats were daily administered with saline (the model group), LA (2 mg/kg), LMWH (3 mg/kg), TLH bolus (containing 8 mg/kg Tempol, 2 mg/kg LA, and 3 mg/kg LMWH), LH NP (at 3 mg/kg LMWH), or TLH NP (at 3 mg/kg LMWH) by *i.v.* injection for 5 days. In the normal group, healthy pregnant rats with sham operation were treated with saline. Data in (b-c) are mean \pm s.d. ($n = 6$). Statistical significance was assessed by one-way ANOVA with post hoc LSD tests. * $p < 0.05$, ** $p < 0.01$, *** $p < 0.001$.

Supplementary Fig. 20 | Effects of different formulations on expression levels of typical inflammatory cytokines and thrombosis-relevant proteins. (a-e) Relative mRNA levels of PAI-1 (a), TAFI (b), TNF- α (c), IL-6 (d), and tPA (e) in left iliac veins. Left iliac veins of pregnant rats at G10 were ligatured to induce DVT. At 6 h after the formation of stenosis-induced thrombi, diseased rats were daily administered with saline (the model group), LA (2 mg/kg), LMWH (3 mg/kg), TLH bolus (*i.e.*, a mixture containing 8 mg/kg Tempol, 2 mg/kg LA, and 3 mg/kg LMWH), LH NP (at 3 mg/kg LMWH), or TLH NP (at 3 mg/kg LMWH) by *i.v.* injection for 5 days. In the normal group, healthy pregnant rats with sham operation were treated with saline. Data are mean \pm s.d. ($n = 6$). Statistical significance was assessed by one-way ANOVA with post hoc LSD tests. * $p < 0.05$, ** $p < 0.01$, *** $p < 0.001$.

Supplementary Fig. 21 | Evaluations of thrombus components after different treatments. (a) H&E-stained histopathological sections of left iliac veins with thrombi. The pink zone indicates the distribution of fibrin. (b-c) Fluorescence images showing CD61-positive platelets (b) and DAPI-stained leukocytes (c) in cryosections of left iliac veins. Scale bars, 1 mm. (d-f) Representative flow cytometric profiles (left) and quantitative analysis (right) showing levels of total leukocytes (d), CD68⁺ macrophages (e), and CD11b⁺/RP-1⁺ neutrophils (f) in the left iliac veins. Left iliac veins of pregnant rats at G10 were ligatured to induce DVT. At 6 h after the formation of stenosis-induced thrombi, diseased rats were daily administered with saline (the model group), LA (2 mg/kg), LMWH (3 mg/kg), TLH bolus (*i.e.*, a mixture containing 8 mg/kg Tempol, 2 mg/kg LA, and 3 mg/kg LMWH), LH NP (at 3 mg/kg LMWH), or TLH NP (at 3 mg/kg LMWH) by *i.v.* injection for 5 days. In the normal group, healthy pregnant rats with sham operation were treated with saline. Data in (d-f) are mean \pm s.d. ($n = 6$). Statistical significance was assessed by one-way ANOVA with post hoc LSD tests. * $p < 0.05$, ** $p < 0.01$, *** $p < 0.001$.

Supplementary Fig. 22 | Evaluations of postnatal fetal malformation and offspring growth profiles after different treatments. (a) The fetal malformation rate of pregnant rats after different treatments. (b) The offspring weight at different time points after birth. For different groups, pregnant rats at G10 received daily *i.v.* injection of LA (2 mg/kg), LMWH (3 mg/kg), TLH bolus (containing 8 mg/kg Tempol, 2 mg/kg LA, and 3 mg/kg LMWH), LH NP (at 3 mg/kg LMWH), or TLH NP (at 3 mg/kg LMWH) for 5 days. The weight of offspring rats in each group at days 0, 7, and 21 after birth was measured. At day 21 after birth, morphological abnormalities of the offspring were evaluated. Data are mean \pm s.d., which are based on offspring rats from 6 pregnant rats in each group. Statistical significance was assessed by one-way ANOVA with post hoc LSD tests. * $p < 0.05$, ** $p < 0.01$, *** $p < 0.001$.

Supplementary Fig. 23 | H&E-stained histological sections of lung bronchi and alveoli of offspring rats after different treatments. For different groups, pregnant rats at G10 received daily *i.v.* injection of LA (2 mg/kg), LMWH (3 mg/kg), TLH bolus (containing 8 mg/kg Tempol, 2 mg/kg LA, and 3 mg/kg LMWH), LH NP (at 3 mg/kg LMWH), or TLH NP (at 3 mg/kg LMWH) for 5 days. The offspring lungs at postnatal day 21 were collected for histological analysis after H&E staining. Scale bars, 100 μ m.

- To further this comment, in figure 3G, some markers of inflammation are measured but as linoleic acid has anti-inflammatory properties, it is perhaps not surprising that the NP groups with linoleic acid showed reduced levels of IL-6 and TNF-alpha. A linoleic acid control group would have shown the added value of the LH and TLH NP groups but this is not discussed.

Response: Thanks for this good suggestion.

According to the proposed comment, we detected mRNA levels of representative pro-inflammatory cytokines (TNF- α and IL-6) after treatment with different formulations. In these cases, free LA was used as a control. While LH NP, TLH NP, and LA significantly decreased the mRNA levels of TNF- α

and IL-6, LH NP and TLH NP showed much better effects than free LA. The related results have been presented in Supplementary Fig. 20 c,d of the revised supplementary material.

Supplementary Fig. 20 | c,d Relative mRNA levels of TNF- α (c) and IL-6 (d) in left iliac veins. Left iliac veins of pregnant rats at G10 were ligatured to induce DVT. At 6 h after the formation of stenosis-induced thrombi, diseased rats were daily administered with saline (the model group), LA (2 mg/kg), LMWH (3 mg/kg), TLH bolus (*i.e.*, a mixture containing 8 mg/kg Tempol, 2 mg/kg LA, and 3 mg/kg LMWH), LH NP (at 3 mg/kg LMWH), or TLH NP (at 3 mg/kg LMWH) by *i.v.* injection for 5 days. In the normal group, healthy pregnant rats with sham operation were treated with saline. Data are mean \pm s.d. ($n = 6$). Statistical significance was assessed by one-way ANOVA with post hoc LSD tests. * $p < 0.05$, ** $p < 0.01$, *** $p < 0.001$.

- For the rat studies, I was unable to find the detail on n's nor whether the data presented (including fetal weight and placental weight) is expressed as litter averages or individual fetuses/placentas. This needs to be explained and justified with litter averages being the gold standard. There is also no information on whether litter size was affected in any of these groups- please provide this information.

Response: Thanks for this good suggestion.

In the original manuscript, we presented digital photos and quantified weight values of all fetuses and placentas from one of the same mother rats. For the original studies, there are 4 pregnant rats in each group. According to the reviewer's comment that the sample sizes are small, we repeated the related experiments with $n = 6$. In the revised manuscript, we selected one representative fetus and placenta from those isolated from individual pregnant rats ($n = 6$), according to the reviewer's suggestion. In addition, the average weight data are based on all fetuses and placentas from 6 pregnant rats. Please see the revised Fig. 5a-d, Fig. 9a-b, Supplementary Fig. 34a-b, and Supplementary Fig. 36c-d.

Fig. 5 a-b Digital photos show gross morphological appearance of representative rat fetuses (a) and placentas (b) from normal or DVT pregnant rats at G17. One fetus or placenta was randomly selected from each rat of different groups ($n = 6$). Scale bar, 1 cm. **c-d** Fetal (c) and placental (d) weights of different groups. Both fetuses and placentas were from 6 pregnant rats in each group. Data in (c-d) are mean \pm s.d., which are based on all fetuses and placentas from 6 pregnant rats in each group. Statistical significance was assessed by one-way ANOVA with post hoc LSD tests. * $p < 0.05$, ** $p < 0.01$, *** $p < 0.001$.

Fig. 9 a Digital photos of representative rat fetuses (upper) and placentas (lower) from normal or DVT pregnant rats at G17. One fetus or placenta was randomly selected from each rat of different groups ($n = 6$). Scale bar, 1 cm. **b** Fetal and placental weights of different groups. Both fetuses and placentas were from 6 pregnant rats in each group. Data in (b) are mean \pm s.d., which are based on all fetuses and placentas from 6 pregnant rats in each group. Statistical significance was assessed by one-way ANOVA with post hoc LSD tests. ** $p < 0.01$, *** $p < 0.001$.

Supplementary Fig. 34 | (a) Digital photos of representative fetuses (upper) and placentas (lower) from normal or DVT pregnant rats subjected to different treatments at G17. One fetus or placenta was randomly selected from each rat of different groups ($n = 6$). Scale bar, 1 cm. (b) Fetal and placental weights of different groups. Both fetuses and placentas were from 6 pregnant rats in each group. Data in (b) are mean \pm s.d., which are based on all fetuses and placentas from 6 pregnant rats in each group. Statistical significance was assessed by one-way ANOVA with post hoc LSD tests. ** $p < 0.01$, *** $p < 0.001$; ns, no significance.

Supplementary Fig. 36 | (c) Digital photos of representative fetuses (upper) and placentas (lower) from pregnant rats in different groups. One fetus or placenta was randomly selected from each rat of different groups ($n = 6$). Scale bar, 1 cm. (d) Fetal and placental weight indices of different groups. Data in (d) are mean \pm s.d., which are based on all fetuses and placentas from 6 pregnant rats in each group.

In experimental studies, fetal growth restriction (FGR) is defined as a fetal body weight below two standard deviations of the mean of normal fetuses¹⁹. According to our results, nanotherapies effectively alleviated FGR caused by DVT. Nevertheless, we counted the numbers of litter size in different groups. It was found that there was no significant difference in the average litter size. We added the related content in the revised manuscript (page 8).

- Many of the figures describe an $n=4$ but I find it difficult to see how normality can be appropriately

determined with such a small number. This makes the use of any parametric statistical tests an issue. Additionally, with $n=4$, the use of bar-graphs are not very intuitive and dot plots as in Figures 4 and 8 are preferable.

Response: Thanks for this good suggestion.

In order to further improve the accuracy of the mentioned studies. We repeated the related experiments with the sample size of 6 (i.e., $n = 6$) in vitro and in vivo. Meanwhile, bar-graphs have been changed to bar graphs with overlapping dots.

- For the data in figures 2G/H and 6G/H it is stated that Cy5 fluorescence was limited to the ‘maternal side’ of the placenta. Please be specific on this, are you saying there is only partial staining in the decidua? In figures 2G and 6G, there is a fluorescent halo around the placenta which usually corresponds to the junctional zone of the rat placenta and implies staining should be present within this zone. However, in Fig 2H and 6H, there is only very limited staining on the periphery of the placenta which seems at odds with this? This is also even more apparent in supplementary Figure 20H for the Cy5-TLH NP model example. Please can you comment.

Response: Thanks for these good questions.

First, we are sorry for the inaccurate expression of ‘maternal side’, which has been changed to ‘decidua’.

For fluorescence images in original Figure 2G/6G and Supplementary Figure 20H, ex vivo imaging was performed. Images in original Figure 2H/6H and Supplementary Figure 20I are based on fluorescence microscopy, which show fluorescence signals from an in-focus plane of the placental section. Since it is impossible for any nanoparticles to fully cover the placenta, fluorescence signals at the in-focus plane of placental sections will be relatively weak or even absent.

According to previous studies, the location of the fluorescent halo around the placenta is the placental decidua²⁰. To further confirm this issue, we repeated related experiments. In this case, Cy5-labeled nanoparticles (including Cy5-LH NP, Cy5-TLH NP, and Cy5-CTLH NP) were i.v. injected to pregnant rats with DVT. Fluorescence microscopy observation of the placental transverse sections indicated that Cy5 fluorescence could be partially detected in the decidua of the placenta, while no fluorescent signals were found in the junctional zone and labyrinthine zone of the placenta (Supplementary Fig. 13 and 27).

Supplementary Fig. 13 | Fluorescence observation of transverse placental sections after i.v. injection of Cy5-LH NP and Cy5-TLH NP in pregnant rats with or without DVT. Scale bar, 200 μm .

Supplementary Fig. 27 | Fluorescence observation of transverse placental sections after *i.v.* injection of Cy5-TLH NP and Cy5-CTLH NP in pregnant rats with DVT. Scale bar, 200 μm .

- In figures 4E and 8E it is difficult to make out any detail on placental morphology from these images. Adding a closer magnification of the junctional and labyrinthine zones would be more intuitive for the reader. Additionally, in the skeletal staining, examples of the TLH NP group in figures 4F and 8E both suggest a much smaller skeleton but this does not appear to fit with the fetal weight data. This should be discussed, as should the fact that whilst there is more mineralisation of the fetal skeletons in NP v model groups, the ribs in particular still appear underdeveloped versus the normal fetus. Do you have any idea about postnatal viability of treated fetuses and whether they have respiratory issues? Were there gross fetal morphological abnormalities in any of the groups? It is difficult to tell from the figures showing multiple fetuses.

Response: Thanks for these good questions and suggestion.

First, we provided closer magnification images for H&E-stained sections originally illustrated in Figure 4E and Figure 8E, showing the decidua, junctional, and labyrinthine zones of the placenta. According to these results, the structures of the decidua and junctional zones in the model group were partially damaged. Treatment with nanotherapies (especially CTLH NP) restored the integrity of the placental structure, as illustrated in the revised Fig. 5f and Fig. 9d.

Fig. 5f Whole slide (upper) and high-magnification (lower) images of H&E-stained placental sections. Scale bars, 4 mm (upper) and 500 μm (lower). De, decidua; Jz, junctional zone; Lz, labyrinth zone.

Fig. 9d Whole slide (upper) and high-magnification (lower) images of H&E-stained placental sections. Scale bars, 2 mm (upper) and 500 μm (lower). De, decidua; Jz, junctional zone; Lz, labyrinth zone.

On the other hand, each pregnant rat had a large number of offspring rats that showed certain

differences in the size of fetuses. In addition, all the internal organs of the fetus need to be removed for bone staining. After one month of staining, dehydration, and washing, it is very hard to maintain the original shape and size of the fetus. Also, the fetal ribs might be partially damaged during the isolation of fetal organs, due to their small size, resulting in some discrepancies. In order to reduce the errors, we repeated the related experiments by increasing the sample size to $n = 6$. Samples from the same batch of animals were randomly selected for fetal staining and examination of the fetal bone development. The newly obtained results have been presented in the revised manuscript (Fig. 5g and Fig. 9e). According to the reviewer's suggestion, the related discussion was added in the revised manuscript (page 9).

Fig. 5g Whole-mount skeletal staining of fetuses via Alizarin red and Alcian blue in different groups at G17. Scale bar, 500 μm .

Fig. 9e Whole-mount skeletal staining of fetuses via Alizarin red and Alcian blue. Scale bar, 200 μm .

After pregnant rats with DVT were treated with different formulations, the postnatal viability and weight of offspring were determined at days 0, 7, and 21 after birth. The results showed that there was no difference in the postnatal viability for offspring rats in different groups. However, the development of offspring in the model group was relatively slow, which could be improved after treatment with nanotherapies (Supplementary Fig. 22b).

Supplementary Fig. 22b | The offspring weight at different time points after birth. For different groups, pregnant rats at G10 received daily i.v. injection of LA (2 mg/kg), LMWH (3 mg/kg), TLH bolus (containing 8 mg/kg Tempol, 2 mg/kg LA, and 3 mg/kg LMWH), LH NP (at 3 mg/kg LMWH), or TLH NP (at 3 mg/kg LMWH) for 5 days. The weight of offspring rats in each group at days 0, 7, and 21 after birth was measured. Data are mean \pm s.d., which are based on offspring rats from 6 pregnant rats in each group.

In addition, histological analysis was conducted for H&E-stained sections of bronchi and alveoli of fetal lungs at day 21 post birth. It could be found that the structure of fetal lung bronchi was abnormal

and the development of alveoli was imperfect in the model group, which could be improved by nanotherapies (Supplementary Fig. 23). Besides, the incidence of fetal malformation in the model group was 17.7%, which was notably increased after treatment with nanotherapies (Supplementary Fig. 22a).

Supplementary Fig. 22a | The fetal malformation rate of pregnant rats after different treatments. Data are mean \pm s.d., which are based on offspring rats from 6 pregnant rats in each group. Statistical significance was assessed by one-way ANOVA with post hoc LSD tests. * $p < 0.05$, ** $p < 0.01$, *** $p < 0.001$.

Supplementary Fig. 23 | H&E-stained histological sections of lung bronchi and alveoli of offspring rats after different treatments. For different groups, pregnant rats at G10 received daily *i.v.* injection of LA (2 mg/kg), LMWH (3 mg/kg), TLH bolus (containing 8 mg/kg Tempol, 2 mg/kg LA, and 3 mg/kg LMWH), LH NP (at 3 mg/kg LMWH), or TLH NP (at 3 mg/kg LMWH) for 5 days. The offspring lungs at postnatal day 21 were collected for histological analysis after H&E staining. Scale bars, 100 μ m.

- In the methods section, the information on animals is very brief and does not include information on housing conditions or number of animals. Ideally, the ARRIVE guidelines should be followed (Kilkenny et al) for reporting of in vivo experiments.

Response: Thanks for this suggestion.

According to this comment, we added the related contents in the revised methods section.

- The number of animal studies are very complex and difficult to follow. A flow chart highlighting the different studies and n's used would be helpful for the reader.

Response: Thanks for this suggestion.

According to this comment, we provided a flowchart highlighting different studies and n values used in Supplementary Fig. 11.

Supplementary Fig. 11 | A flowchart showing procedures and animal numbers for different *in vivo* studies.

- The discussion is very brief and represents mainly a re-cap of the major findings. There is very limited discussion of how this data compares with published work and a lack of consideration of the limitations of this study. This needs to be addressed.

Response: Thanks for this suggestion.

According to this comment, we revised the discussion section by comparing with published work and providing limitations of this study (page 12).

Minor points:

- In the introduction (lines 61-62) it is stated that aspirin is prohibited in the last trimester of pregnancy due to risk of bleeding with a reference from 1998. This statement is incorrect. In higher risk pregnancies, aspirin may be prescribed, often up until approx 36 weeks of pregnancy. Please amend or delete this statement.

Response: Thanks for this suggestion. We apologized for this mistake, and the related description was deleted.

- On line 67, the necessity to inject LMWH twice-daily is listed as an issue with compliance. If this nanoparticle therapy is translated to patients, I presume patients would have to visit hospital for regular i.v injections? This could also prove a problem with compliance and should be discussed.

Response: Thanks for this good suggestion.

Indeed, as indicated by the reviewer, i.v. injection of nanotherapies involves the similar patient compliance issue. For clinical translation of our nanotherapies, the injection regimens need to be further improved, such as i.v. injection within prolonged time intervals or delivery via subcutaneous injection in future studies. Since the anticoagulant activity of nanotherapies remains after 48 h of i.v. injection, we reasonably consider that i.v. injection of our nanotherapies once every two days or even longer should be effective. We added the related content as the limitation of our study in the revised manuscript (page 12).

- Information of the statistical test used should be added to each figure legend.

Response: We have provided the information of the statistical tests in each figure legend.

Reviewer #3 (Remarks to the Author):

In this research article, the authors develop a nanoparticulate platform for the treatment of deep vein thrombosis. Low molecular weight heparin (LMWH) is conjugated to Tempol as an antioxidative agent

and lauric acid as an anti-inflammatory agent. The construct is further functionalized with CREKA, which targets fibrin. After fabricating the nanoparticles by self-assembly and performing in vitro characterizations, the authors evaluate their platform in vivo. It is shown that the nanoparticles effectively target sites of thrombosis and have considerable therapeutic efficacy in multiple rodent models. Overall, the authors have performed a substantial amount of work to support their claims, although there are still some important points that need to be addressed.

Response: We appreciate the above positive comments on our manuscript.

According to the constructive comments proposed by the reviewer, additional experiments were performed and the manuscript was substantially revised.

1. Structurally, the manuscript was made unnecessarily long because the authors didn't add the CREKA moiety until later. Why did they not just add this experimental group to the first set of in vivo studies?

Response: Thanks for this question.

Initially, we designed and developed LH NP and TLH NP. After confirmation of their desirable anti-thrombotic effects in the examined DVT model and in order to develop a nanotherapy with more beneficial efficacy, CREKA was used to functionalize TLH, thereby affording an active targeting nanotherapy CTLH NP. Therefore, these studies were performed step by step rather than at the same time.

2. The main figures are focused mainly on the in vivo studies, but the formulation development data should also be included in the main manuscript.

Response: Thanks for this suggestion.

Some formulation development data have been presented in Fig. 2 of the revised main manuscript.

3. It is unclear how the authors normalized the dosages for the formulations. The most reasonable approach is to evaluate the formulations at the same concentration/dosage of LMWH. If the authors used a different approach, then they need to justify why they believe that was most appropriate. Regardless, this point needs to be explicitly clarified.

Response: Thanks for this question.

We normalized the dosages for different formulations at the same dose of LMWH in both in vitro and in vivo experiments. We explicitly clarified this point in the revised manuscript.

4. The mechanism for self-assembly of the nanostructure should be discussed more in detail.

Response: Thanks for this good question.

Covalent incorporation of hydrophobic linoleic acid and Tempol onto LMWH resulted in amphiphilic LMWH derivatives, which can form core-shell structured nanoparticles by hydrophobic interaction-mediated self-assembly in aqueous solution. We further examined self-assembly behaviors of amphiphilic LMWH derivatives by fluorescence spectrometry using pyrene as a probe. Based on pyrene fluorescence spectra, the critical aggregation concentration (CAC) of different LMWH derivatives was quantified. The related results have been presented in Fig. 2f,g and discussed in the main text (page 4).

Fig. 2 f,g Typical excitation fluorescence spectra (left) and the corresponding plots of the intensity ratio I_{338}/I_{333} as a function of $\text{Log } C$ (right) for pyrene in the presence of increasing concentrations (g/L) of LH (f) or TLH (g).

5. There are multiple examples of interesting and unexpected phenomena that go unexplained by the authors.

a. The anticoagulant activity of the LMWH was halved as a result of its incorporation into the nanoformulation. The authors should discuss the reason for this decrease. It is also odd that the nanoformulations outperform free LMWH in the in vitro anticoagulation assays despite have significantly reduced activity.

Response: Thanks for this good question.

In the anticoagulant tests, we calculated and compared the anti-FXa activity per mg sample of LMWH, LH NP, and TLH NP. Of note, there are LA moieties in LH NP, while TLH NP contains both LA and Tempol. Nevertheless, the anticoagulant activity of LH NP and TLH NP, expressed as the content of LMWH, was almost comparable to that of LMWH, indicating that nanotherapies did not compromise the LMWH activity. In addition, in vitro experiments indicated that nanotherapies showed more excellent anticoagulant activity than free LMWH, mainly by increasing the binding interactions with platelets and tissue factors. We added the related description in the revised manuscript (page 4).

b. It is claimed that Tempol has high affinity to activated platelets. Do the authors have any references to support this claim? If not, what explanation to they have for this phenomenon, since Tempol was included as the ROS scavenger and not as a targeting moiety. Overall, it is hard to imagine that the Tempol can have such a marked effect on targeting both in vitro and in vivo.

Response: Thanks for this good question.

According to previous findings, oxidation of protein disulfide isomerase (PDI) on the platelet surface plays a key role in platelet activation and aggregation²¹. As a ROS-scavenging agent, Tempol can covalently bind to PDI, thereby inhibiting PDI oxidation²². In addition, relatively small negative ζ -potential values of TLH NP in serum might be beneficial for enhanced thrombus or activated platelet targeting²³, although in-depth studies remain to be conducted to address mechanisms underlying the desirable targeting efficiency of TLH NP. The related contents were added in the revised manuscript (page 5).

c. How did the LH NP formulation attenuate ROS-induced endothelial cell apoptosis when it does not have an ROS-scavenging component?

Response: Thanks for this good question.

It is well known that Ca^{2+} is related to cell apoptosis. Increased intracellular Ca^{2+} can trigger endoplasmic reticulum stress and activate caspase-mediated cell apoptosis. Previous studies showed that LMWH can inhibit cell apoptosis by regulating intracellular Ca^{2+} ²⁴. In addition, it was found that

LMWH may prevent oxidative stress-induced cell apoptosis by increasing the expression of SOD2²⁵. We added the relevant contents in the revised manuscript (page 6).

6. The dose-dependent uptake data in Fig. S8 is unexpected. For example, why would the uptake at 200 $\mu\text{g}/\text{mL}$ be more than double the uptake at 100 $\mu\text{g}/\text{mL}$? This points to a faulty experimental setup or insufficient washing of the samples prior to analysis.

Response: Thanks for this good question.

First, we are sorry for mistakes during quantification of fluorescence intensities based on flow cytometric analyses. The related flow cytometric curves were re-analyzed. In addition, since the sample sizes were 4 in original experiments, we performed additional experiments to afford results with $n = 6$. Quantitative data based on original and new experiments have been presented in the revised Supplementary Fig. 8.

Supplementary Fig. 8 | Flow cytometric quantification of dose-dependent cellular uptake of LH NP and TLH NP in HUVECs. (a-b) Typical flow cytometric curves (left) and quantitative data (right) indicating cellular uptake of various doses of Cy5-LH NP (a) or Cy5-TLH NP (b) after 2 h of incubation in HUVECs. Data are mean \pm s.d. ($n = 6$).

7. Subcutaneous LMWH does not provide a fair comparison for the nanoformulations, which are administered intravenously. The authors should consider comparing the nanoformulations with a “safe” LMWH intravenous dose for the in vivo studies. This is reasonable considering that the authors demonstrated that LMWH has higher biological activity in free form compared with in nanoparticulate form.

Response: Thanks for this good suggestion.

Clinically, subcutaneous injection of LMWH is recommended for gestational diseases to reduce the risk of bleeding and improve bioavailability. We agree with the reviewer that a safe LMWH intravenous injection should be used as the control. Therefore, we selected 3 mg/kg LMWH for intravenous injection based on in vivo safety studies in pregnant rats with tail vein bleeding (Supplementary Fig. 14). We compared therapeutic effects of LMWH (intravenous injection at 3 mg/kg) with nanotherapies and other formulations by both in vitro and in vivo experiments. The related results have been presented in the revised supplementary material (Supplementary Fig. 19-23).

Supplementary Fig. 14 | The effects of different formulations on tail vein bleeding in pregnant rats. (a-b) Quantified bleeding time (a) and volume (b) after injection of different formulations. LMWH was administered by either *i.v.* injection at 3 and 5 mg/kg or subcutaneous (*s.c.*) injection at 5 mg/kg, while all nanotherapies were delivered via *i.v.* injection at 5 mg/kg of LMWH. In the normal group, rats were treated with saline. At 10 min after administration of different formulations, pregnant rats were subjected to cutting off the entire tail tip. Both bleeding time and bleeding volume were quantified. Data are mean \pm s.d. ($n = 8$). Statistical significance was assessed by one-way ANOVA with post hoc LSD tests. * $p < 0.05$, *** $p < 0.001$; ns, no significance.

Supplementary Fig. 19 | Comparison of therapeutic effects of two nanotherapies with LA, the TLH bolus, and *i.v.* LMWH in pregnant rats with stenosis-induced DVT. (a) Representative digital photos of left iliac veins with thrombi isolated from pregnant rats after treatment with different formulations. Scale bar, 1 mm. (b-c) The weight (b) and length (c) of thrombi in left iliac veins. (d) H&E-stained histopathological sections of left iliac veins with thrombi. Scale bar, 400 μ m. (e) Ultrasound images of left iliac veins. The yellow dashed lines indicate the vascular endothelium,

while the red dashed lines indicate thrombi. Scale bar, 1 mm. (f) SEM observation of the endothelial surface of left iliac veins with thrombi. Scale bar, 10 μ m. In all these studies, left iliac veins of pregnant rats at G10 were ligatured to induce DVT. At 6 h after the formation of stenosis-induced thrombi, diseased rats were daily administered with saline (the model group), LA (2 mg/kg), LMWH (3 mg/kg), TLH bolus (containing 8 mg/kg Tempol, 2 mg/kg LA, and 3 mg/kg LMWH), LH NP (at 3 mg/kg LMWH), or TLH NP (at 3 mg/kg LMWH) by *i.v.* injection for 5 days. In the normal group, healthy pregnant rats with sham operation were treated with saline. Data in (b-c) are mean \pm s.d. ($n = 6$). Statistical significance was assessed by one-way ANOVA with post hoc LSD tests. * $p < 0.05$, ** $p < 0.01$, *** $p < 0.001$.

Supplementary Fig. 20 | Effects of different formulations on expression levels of typical inflammatory cytokines and thrombosis-relevant proteins. (a-e) Relative mRNA levels of PAI-1 (a), TAFI (b), TNF- α (c), IL-6 (d), and tPA (e) in left iliac veins. Left iliac veins of pregnant rats at G10 were ligatured to induce DVT. At 6 h after the formation of stenosis-induced thrombi, diseased rats were daily administered with saline (the model group), LA (2 mg/kg), LMWH (3 mg/kg), TLH bolus (*i.e.*, a mixture containing 8 mg/kg Tempol, 2 mg/kg LA, and 3 mg/kg LMWH), LH NP (at 3 mg/kg LMWH), or TLH NP (at 3 mg/kg LMWH) by *i.v.* injection for 5 days. In the normal group, healthy pregnant rats with sham operation were treated with saline. Data are mean \pm s.d. ($n = 6$). Statistical significance was assessed by one-way ANOVA with post hoc LSD tests. * $p < 0.05$, ** $p < 0.01$, *** $p < 0.001$.

Supplementary Fig. 21 | Evaluations of thrombus components after different treatments. (a) H&E-stained histopathological sections of left iliac veins with thrombi. The pink zone indicates the distribution of fibrin. (b-c) Fluorescence images showing CD61-positive platelets (b) and DAPI-stained leukocytes (c) in cryosections of left iliac veins. Scale bars, 1 mm. (d-f) Representative flow cytometric profiles (left) and quantitative analysis (right) showing levels of total leukocytes (d), CD68⁺ macrophages (e), and CD11b⁺/RP-1⁺ neutrophils (f) in the left iliac veins. Left iliac veins of pregnant rats at G10 were ligatured to induce DVT. At 6 h after the formation of stenosis-induced thrombi, diseased rats were daily administered with saline (the model group), LA (2 mg/kg), LMWH (3 mg/kg), TLH bolus (*i.e.*, a mixture containing 8 mg/kg Tempol, 2 mg/kg LA, and 3 mg/kg LMWH), LH NP (at 3 mg/kg LMWH), or TLH NP (at 3 mg/kg LMWH) by *i.v.* injection for 5 days. In the normal group, healthy pregnant rats with sham operation were treated with saline. Data in (d-f) are mean \pm s.d. ($n = 6$). Statistical significance was assessed by one-way ANOVA with post hoc LSD tests. * $p < 0.05$, ** $p < 0.01$, *** $p < 0.001$.

Supplementary Fig. 22 | Evaluations of postnatal fetal malformation and offspring growth profiles after different treatments. (a) The fetal malformation rate of pregnant rats after different treatments. (b) The offspring weight at different time points after birth. For different groups, pregnant rats at G10 received daily *i.v.* injection of LA (2 mg/kg), LMWH (3 mg/kg), TLH bolus (containing 8 mg/kg Tempol, 2 mg/kg LA, and 3 mg/kg LMWH), LH NP (at 3 mg/kg LMWH), or TLH NP (at 3 mg/kg LMWH) for 5 days. The weight of offspring rats in each group at days 0, 7, and 21 after birth was measured. At day 21 after birth, morphological abnormalities of the offspring were evaluated. Data are mean \pm s.d., which are based on offspring rats from 6 pregnant rats in each group. Statistical significance was assessed by one-way ANOVA with post hoc LSD tests. * $p < 0.05$, ** $p < 0.01$, *** $p < 0.001$.

Supplementary Fig. 23 | H&E-stained histological sections of lung bronchi and alveoli of offspring rats after different treatments. For different groups, pregnant rats at G10 received daily *i.v.* injection of LA (2 mg/kg), LMWH (3 mg/kg), TLH bolus (containing 8 mg/kg Tempol, 2 mg/kg LA, and 3 mg/kg LMWH), LH NP (at 3 mg/kg LMWH), or TLH NP (at 3 mg/kg LMWH) for 5 days. The offspring lungs at postnatal day 21 were collected for histological analysis after H&E staining. Scale bars, 100 μ m.

8. The authors claimed that “no beneficial effects were observed after treatment with Tempol” in Figure 4, yet there was a significant recovery of fetal weight.

Response: We are sorry for this inaccurate description, which has been changed in the revised manuscript (page 8).

9. It is incorrectly stated that “the fluorescent intensity of the Cy5-CTLH NP group was 1.6-fold higher than that of the Cy5-TLH NP group” in Figure 6A. The difference between the two groups is much less than 1.6-fold.

Response: We are sorry for this mistake.

We checked the related data and “1.6-fold” was changed into “1.2-fold” in the revised manuscript

(page 9).

Reviewer #4 (Remarks to the Author):

This is an experimental rat study of gestation and induced DVT, to assess a novel nano-compound, with fibrin binding for LMWH and tempol and linoleic acid.

1. The stenosis model of iliac vein is not well described -- how was the degree of stenosis determined for reproducibility and reliability?

Response: Thanks for this good question.

To establish the stenosis models, needles, threads, or gauzes are generally used to cause vascular stenosis, which are then removed after ligation. According to the previously reported method, 30G needles were used, in combination with ligation, to create stenosis in our studies, which can ensure 80% reduction of blood flow²⁶. Moreover, we used Doppler ultrasound to observe the stenosis of iliac vein thrombosis after modeling. It was found our model could reduce blood flow by 74.5-84.3%. The related contents were added in the revised manuscript (page 6).

2. Were the agents given in a blinded fashion separate from the surgical model?

Response: Thanks for this good question.

Yes, we administered all the examined formulations in a blinded fashion. Firstly, we labeled all pregnant rats, which were randomly assigned into two groups, i.e., the sham and surgery groups. Except pregnant rats in the sham operation group, other rats were then randomly assigned to different groups according to their numbers. Subsequently, treatment with different formulations was conducted by another person who is blind to different groups and agents. We added the related description in the revised methods.

3. It is hard to define the range of thrombi in terms of resolution without a dot plot -- please change that up. And, it is hard to reconcile, knowing the stenosis model is quite variable in terms of thrombogenesis, that some % of rats will have no thrombus whatsoever. This needs to be clarified.

Response: Thanks for this good question.

According to the above comment, we changed all bar-graphs into bar graphs with overlapping dots.

In addition, we used ultrasound imaging to detect the formation of thrombosis after 6 h of modeling. It was found that the formation rate of thrombosis at 6 h after operation was 89.7%. Different treatment experiments were performed only after confirming the formation of thrombosis.

4. Can you characterize the thrombus from the control vs agents, in terms of fibrin content, platelet content, etc - -given you claim that your agent impacts all these components?

Response: Thanks for this good suggestion.

In order to verify the effects of different formulations on thrombus components, histological analyses were performed for iliac veins with thrombi from different groups. Specifically, the fibrin content was examined based on H&E-stained histological sections²⁷. The pink zone indicates the distribution of fibrin, while the distribution of platelets was examined by immunofluorescence analysis of sections stained with CD61 antibody. Infiltration of leukocytes was examined by DAPI staining²⁸. The results showed that nanotherapies significantly reduced the fibrin content as well as counts of platelets and leukocytes in thrombi. The related results have been presented in Supplementary Fig.

21a-c.

Supplementary Fig. 21 | (a) H&E-stained histopathological sections of left iliac veins with thrombi. (b-c) Fluorescence images showing CD61-positive platelets (b) and DAPI-stained leukocytes (c) in cryosections of left iliac veins. Scale bars, 1 mm.

5. Did you assess the thrombus leukocyte influx and characterize this? Would be interesting to do with your compound's potential effects.

Response: Thanks to this good suggestion.

According to this comment, histological analyses were performed for iliac veins with thrombi from different groups. Infiltration of leukocytes was examined by DAPI staining²⁸. The results showed that nanotherapies significantly reduced the counts of leukocytes in thrombi. The related results have been presented in Supplementary Fig. 21c.

Supplementary Fig. 21 | (c) Fluorescence images showing DAPI-stained leukocytes in cryosections of left iliac veins. Scale bar, 1 mm.

In addition, we also analyzed the leukocyte influx in thrombi by flow cytometry. The results indicated that treatment with nanotherapies significantly reduced the cell populations of leukocytes, neutrophils, and macrophages in thrombi (Supplementary Fig. 21d-f).

Supplementary Fig. 21 | (d-f) Representative flow cytometric profiles (left) and quantitative analysis (right) showing levels of total leukocytes (d), CD68⁺ macrophages (e), and CD11b⁺/RP-1⁺ neutrophils (f) in the left iliac veins. Data are mean \pm s.d. ($n = 6$). Statistical significance was assessed by one-way ANOVA with post hoc LSD tests. * $p < 0.05$, ** $p < 0.01$, *** $p < 0.00119$.

6. What were the statistical methods used for these multi-comparisons? Again, I find it hard to believe a $N = 4$ at each treatment would give you adequate power for such dramatic P values.

Response: Thanks for this good question.

For experiments consisting of more than 2 groups, we used one-way ANOVA, while unpaired t-test was performed for experiments with two groups. The related information was given in the corresponding figure captions.

To maintain good reproducibility for all in vivo studies, experimental conditions such as the operators, instruments, and standardized operation procedures, were well controlled, which ensure relatively small variances. According to the reviewers' comments, we repeated the related studies using the sample size of 6 (i.e., $n = 6$). Also, the related source data have been provided.

References

- Groom KM, David AL. The role of aspirin, heparin, and other interventions in the prevention and treatment of fetal growth restriction. *Am. J. Obstet. Gynecol.* **218**, S829-S840 (2018).
- Peeters LL. Thrombophilia and fetal growth restriction. *Eur. J. Obstet. Gynecol. Reprod. Biol.* **95**, 202-205 (2001).
- Papadakis E, et al. Low molecular weight heparins use in pregnancy: a practice survey from greece and a review of the literature. *Thromb. J.* **17**, 23 (2019).
- Mulloy B, Hogwood J, Gray E, Lever R, Page CP. Pharmacology of heparin and related drugs. *Pharmacol. Rev.* **68**, 76-141 (2016).
- Chen Y, et al. Antithrombin iii-binding site analysis of low-molecular-weight heparin fractions. *J. Pharm. Sci.* **107**, 1290-1295 (2018).
- Kjellberg M, Ikonomou T, Stenflo J. The cleaved and latent forms of antithrombin are normal constituents of blood plasma: a quantitative method to measure cleaved antithrombin. *J. Thromb. Haemost.* **4**, 168-176 (2006).

7. Kanji R, Kubica J, Navarese EP, Gorog DA. Endogenous fibrinolysis-relevance to clinical thrombosis risk assessment. *Eur. J. Clin. Invest.* **51**, e13471 (2021).
8. Obi AT, *et al.* Low-molecular-weight heparin modulates vein wall fibrotic response in a plasminogen activator inhibitor 1-dependent manner. *J. Vasc. Surg. Venous Lymphat. Disord.* **2**, 441-450 (2014).
9. Monagle P, *et al.* American society of hematology 2018 guidelines for management of venous thromboembolism: Treatment of pediatric venous thromboembolism. *Blood Adv.* **2**, 3292-3316 (2018).
10. Zierler S, Rothman KJ. Congenital heart disease in relation to maternal use of bendectin and other drugs in early pregnancy. *N. Engl. J. Med.* **313**, 347-352 (1985).
11. Wilcox CS. Effects of Tempol and redox-cycling nitroxides in models of oxidative stress. *Pharmacol. Ther.* **126**, 119-145 (2010).
12. Li L, *et al.* A broad-spectrum ROS-eliminating material for prevention of inflammation and drug-induced organ toxicity. *Adv. Sci.* **5**, 1800781 (2018).
13. Yin H, Liu J, Li Z, Berndt MC, Lowell CA, Du X. Src family tyrosine kinase lyn mediates VWF/GPIb-IX-induced platelet activation via the cGMP signaling pathway. *Blood* **112**, 1139-1146 (2008).
14. De Candia E, De Cristofaro R, Landolfi R. Thrombin-induced platelet activation is inhibited by high- and low-molecular-weight heparin. *Circulation* **99**, 3308-3314 (1999).
15. Darwish NHE, Godugu K, Mousa SA. Sulfated non-anticoagulant low molecular weight heparin in the prevention of cancer and non-cancer associated thrombosis without compromising hemostasis. *Thromb. Res.* **200**, 109-114 (2021).
16. Sagrillo-Fagundes L, Assuncao Salustiano EM, Ruano R, Markus RP, Vaillancourt C. Melatonin modulates autophagy and inflammation protecting human placental trophoblast from hypoxia/reoxygenation. *J. Pineal Res.* **65**, e12520 (2018).
17. Cheng J, Fu Z, Zhu J, Zhou L, Song W. The predictive value of plasminogen activator inhibitor-1, fibrinogen, and D-dimer for deep venous thrombosis following surgery for traumatic lower limb fracture. *Ann. Palliat. Med.* **9**, 3385-3392 (2020).
18. Zheng Y, Pan N, Liu Y, Ren X. Novel porous chitosan/n-halamine structure with efficient antibacterial and hemostatic properties. *Carbohydr. Polym.* **253**, 117205 (2021).
19. American College of Obstetricians and Gynecologists' Committee on Practice Bulletins—Obstetrics and the Society for Maternal-Fetal Medicine. ACOG practice bulletin No. 204: fetal growth restriction. *Obstet. Gynecol.* **133**, e97-e109 (2019).
20. Turanov AA, *et al.* RNAi modulation of placental sFLT1 for the treatment of preeclampsia. *Nat. Biotechnol.* **36**, 1164-1173 (2018).
21. Wang L, *et al.* The extracellular Ero1 α /PDI electron transport system regulates platelet function by increasing glutathione reduction potential. *Redox Biol.* **50**, 102244 (2022).
22. Santos GB, *et al.* Nitroxide 4-hydroxy-2,2',6,6'-tetramethylpiperidine 1-oxyl (Tempol) inhibits the reductase activity of protein disulfide isomerase via covalent binding to the Cys400 residue on CXXC redox motif at the a'active site. *Chem. Biol. Interact.* **272**, 117-124 (2017).
23. Cheng J, *et al.* Facile assembly of cost-effective and locally applicable or injectable nanohemostats for hemorrhage control. *ACS Nano* **10**, 9957-9973 (2016).
24. Hao LN, Zhang QZ, Yu TG, Cheng YN, Ji SL. Antagonistic effects of ultra-low-molecular-weight heparin on A β 25-35-induced apoptosis in cultured rat cortical neurons. *Brain Res.* **1368**, 1-10 (2011).
25. Tamaru S, *et al.* Heparin prevents oxidative stress-induced apoptosis in human decidualized

- endometrial stromal cells. *Med. Mol. Morphol.* **52**, 209-216 (2019).
26. Schonfelder T, Jackel S, Wenzel P. Mouse models of deep vein thrombosis. *Gefasschirurgie* **22**, 28-33 (2017).
 27. Darcourt J, *et al.* Absence of susceptibility vessel sign is associated with aspiration-resistant fibrin/platelet-rich thrombi. *Int. J. Stroke* **16**, 972-980 (2021).
 28. Von Bruhl ML, *et al.* Monocytes, neutrophils, and platelets cooperate to initiate and propagate venous thrombosis in mice in vivo. *J. Exp. Med.* **209**, 819-835 (2012).

REVIEWER COMMENTS

Reviewer #1 (Remarks to the Author):

This is a thoroughly revised version of a MS I had reviewed in November 2021. I carefully read all criticisms and comments of my 3 fellow reviewers of the initial MS and studied the responses of the authors who did a large series of additional experiments in response to the reviewers' comments.

I remain with my initial assessment that this is a highly innovative study on a nanoparticle-based antithrombotic treatment (with thrombolytic potential) applied to pregnant rats with an induced iliac vein thrombosis (stenosis model).

In sum the nanoparticles containing LMW-heparin moieties covalently linked to linoleic acid, tempol and CREKA (the latter peptide targeting fibrin) seem to be effective and safe (also for the fetus). The authors answered adequately to each individual comment/criticism of the 4 referees. All my suggestions and comments have been addressed, I have 2 mini points and would ask the authors to address them (I am not asking for rereviewing the MS):

- In the responses of the authors, page 3, Fig. 2p. Do not label the TAT values (usually measured in micrograms/L) as "TAT activity", label them as "TAT concentration" or just values.

- In the Responses the authors repeatedly use the wording "...we ligatured....", I believe this should better be "...we ligated"

I propose to accept the MS for publication and congratulate the authors for this study which has, in this referee's opinion a high translational potential.

Bernhard Lämmle (11-June-2022)

Reviewer #2 (Remarks to the Author):

The authors should be commended for their attempts to amend the manuscript according to the initial reviewer comments. This includes an impressive amount of new data which I feel allows firmer conclusions to be made regarding this paper. I have no further comments to make.

Reviewer #3 (Remarks to the Author):

The authors have satisfactorily addressed the previous round of comments.

Reviewer #4 (Remarks to the Author):

Thank you for adding in the N = 2 for the DVT assays.

1. Given your stenosis model, how did not find some rats without any thrombus, especially with your compounds that seem to inhibit all major thrombogenic mechanisms (inflammatory, PAI-1, vWF)?

2. I reviewed your source data -- how can you get a thrombus measurement with any accuracy to the microgram range??

Response to Reviewers

REVIEWER COMMENTS

Reviewer #1 (Remarks to the Author):

This is a thoroughly revised version of a MS I had reviewed in November 2021. I carefully read all criticisms and comments of my 3 fellow reviewers of the initial MS and studied the responses of the authors who did a large series of additional experiments in response to the reviewers' comments.

Response: Thanks for careful reading of our revised manuscript and responses to previous comments by 4 reviewers. Also, we sincerely appreciate the positive comment and additional suggestion provided by reviewer #1.

I remain with my initial assessment that this is a highly innovative study on a nanoparticle-based antithrombotic treatment (with thrombolytic potential) applied to pregnant rats with an induced iliac vein thrombosis (stenosis model).

In sum the nanoparticles containing LMW-heparin moieties covalently linked to linoleic acid, tempol and CREKA (the latter peptide targeting fibrin) seem to be effective and safe (also for the fetus).

The authors answered adequately to each individual comment/criticism of the 4 referees.

Response: Thanks again for these positive comments on our manuscript.

All my suggestions and comments have been addressed, I have 2 mini points and would ask the authors to address them (I am not asking for rereviewing the MS):

- In the responses of the authors, page 3, Fig. 2p. Do not label the TAT values (usually measured in micrograms/L) as "TAT activity", label them as "TAT concentration" or just values.

Response: Thanks for this suggestion.

According to this comment, we changed "TAT activity" into "TAT values (ng/mL)". The unnormalized "TAT values (ng/mL)" are shown in revised Fig. 2p and presented in the source data as well.

- In the Responses the authors repeatedly use the wording "...we ligatured...", I believe this should better be "...we ligated"

Response: Thanks for this good suggestion.

Indeed, the mentioned "ligatured" should be "ligated". Also, we carefully checked the manuscript to avoid misuse of this word.

I propose to accept the MS for publication and congratulate the authors for this study which has, in this referee's opinion a high translational potential.

Response: We thank the reviewer again for the very supportive comments.

Reviewer #2 (Remarks to the Author):

The authors should be commended for their attempts to amend the manuscript according to the initial reviewer comments. This includes an impressive amount of new data which I feel allows firmer conclusions to be made regarding this paper. I have no further comments to make.

Response: We really appreciate the reviewer's positive comment on our revised manuscript.

Reviewer #3 (Remarks to the Author):

The authors have satisfactorily addressed the previous round of comments.

Response: Again, we sincerely appreciate the reviewer's positive comment on our revised manuscript.

Reviewer #4 (Remarks to the Author):

Thank you for adding in the N = 2 for the DVT assays.

1. Given your stenosis model, how did not find some rats without any thrombus, especially with your compounds that seem to inhibit all major thrombogenic mechanisms (inflammatory, PAI-1, vWF)?

Response: Thanks for these good questions.

As mentioned in the manuscript, we used Doppler ultrasound to confirm the thrombus formation and reduced blood flow in the iliac vein after modeling in pregnant rats. Only rats with the confirmed thrombus formation were used for following experiments, while pregnant rats without thrombus were excluded. We mentioned this point in the revised methods (see page 18).

In addition, our compounds (i.e., nanotherapies) can inhibit almost all major thrombogenic mechanisms because the presence of relevant bioactive components, such as Tempol, LA, and LMWH^{1,2}. Of note, Tempol functions as an antioxidative component³, while LA is an anti-inflammatory agent⁴. In addition, LMWH can inhibit PAI-I and protect the endothelium by suppressing vWF^{5,6}. Accordingly, multiple bioactivities of our nanotherapies are afforded by the integrated different components with various pharmacological effects.

2. I reviewed your source data -- how can you get a thrombus measurement with any accuracy to the microgram range??

Response: Thanks for this very good question. We are sorry for unclear description that leads to misunderstanding.

We agree with the reviewer that it is impossible to get a thrombus measurement with accuracy to the microgram range when general laboratory analytical balances are used.

All the data of thrombus measurement were recorded for pregnant rats. The difference in the number of litters may lead to a notable difference in the body weight of pregnant rats, which affects the formed thrombus, thereby resulting in considerable differences in the same group. To reduce this difference, we normalized the weight of thrombi in each group, using the formula:

The normalized thrombus weight = (Weight of thrombus/Weight of pregnant rat) × Mean weight of pregnant rats

After data normalizing, two recorders reported data with differences in the number of digits reserved, i.e., some data with 2 decimal places and some with 4 decimal places. To keep the same number of decimal places and give more reasonable data, 2 decimal places are kept for all thrombus weight data (please see the revised source data). It is worth noting that this issue does not affect all the statistical results. The related contents were added in the revised methods (see page 19). Also, the y-axis labels were revised in the related figures.

References

1. Mulloy B, Hogwood J, Gray E, Lever R, Page CP. Pharmacology of heparin and related drugs. *Pharmacol Rev* **68**, 76-141 (2016).
2. Li X, *et al.* Sensitive activatable nanoprobes for real-time ratiometric magnetic resonance imaging of reactive oxygen species and ameliorating inflammation in vivo. *Adv Mater* **34**,

e2109004 (2022).

3. Zhang Q, *et al.* A superoxide dismutase/catalase mimetic nanomedicine for targeted therapy of inflammatory bowel disease. *Biomaterials* **105**, 206-221 (2016).
4. Belkind-Gerson J, Carreón-Rodríguez A, Contreras-Ochoa CO, Estrada-Mondaca S, Parra-Cabrera MS. Fatty acids and neurodevelopment. *J Pediatr Gastroenterol Nutr* **47**, S7-9 (2008).
5. Wang D, *et al.* Embelin ameliorated sepsis-induced disseminated intravascular coagulation intensities by simultaneously suppressing inflammation and thrombosis. *Biomed Pharmacother* **130**, 110528 (2020).
6. Diaz JA, *et al.* P-selectin inhibition therapeutically promotes thrombus resolution and prevents vein wall fibrosis better than enoxaparin and an inhibitor to von willebrand factor. *J Vasc Surg Venous Lymphat Disord* **2**, 114 (2014).

REVIEWERS' COMMENTS

Reviewer #4 (Remarks to the Author):

Thanks for the clarifications.

Although there are still some concerns with duplex accuracy of thrombus measurement, i am comfortable with the explained methods of how thrombogenesis was determined and also how the various pathways were inhibited.

No further comments to address.

Response to Reviewers

REVIEWER COMMENTS

Reviewer #4 (Remarks to the Author): Thanks for the clarifications.

Although there are still some concerns with duplex accuracy of thrombus measurement, i am comfortable with the explained methods of how thrombogenesis was determined and also how the various pathways were inhibited.

Response: We really appreciate the reviewer's valuable comments during the whole peer review process, which can notably improve the quality and scientific rigor of our manuscript. Many thanks for your positive comments on our revised manuscript.

No further comments to address.

Response: Thanks again for your positive comments on our revised manuscript.